EMBO
Molecular Medicine

# Embryological cellular origins and hypoxia-mediated mechanisms in *PIK3CA*-driven refractory vascular malformations

Sota Torii[1], Keiki Nagaharu [2], Nanako Nakanishi[1], Hidehito Usui[3], Yumiko Hori[4,5], Katsutoshi Hirose [6], Satoru Toyosawa[6], Eiichi Morii [4], Mitsunaga Narushima[7], Yoshiaki Kubota[8], Osamu Nakagawa[9], Kyoko Imanaka-Yoshida[1] & Kazuaki Maruyama [1]✉

## Abstract

**Congenital vascular malformations, affecting 0.5% of the population, often occur in the head and neck, complicating treatment due to the critical functions in these regions. Our previous research identified distinct developmental origins for blood and lymphatic vessels in these areas, tracing them to the cardiopharyngeal mesoderm (CPM), which contributes to the development of the head, neck, and cardiovascular system in both mouse and human embryos. In this study, we investigated the pathogenesis of these malformations by expressing Pik3ca^{H1047R} in the CPM. Mice expressing Pik3ca^{H1047R} in the CPM developed vascular abnormalities restricted to the head and neck. Single-cell RNA sequencing revealed that Pik3ca^{H1047R} upregulates *Vegf-a* expression in endothelial cells through HIF-mediated hypoxia signaling. Human samples supported these findings, showing elevated HIF-1α and VEGF-A in malformed vessels. Notably, inhibition of HIF-1α and VEGF-A in the mouse model significantly reduced abnormal vasculature. These results highlight the role of embryonic origins and hypoxia-driven mechanisms in vascular malformations, providing a foundation for the development of therapies targeting these difficult-to-treat conditions.**

**Keywords** Vascular Malformations; Endothelial Cellular Origin; Cardiopharyngeal Mesoderm; PIK3CA Mutation; Hypoxia
**Subject Categories** Cardiovascular System; Vascular Biology & Angiogenesis

## Introduction

Vascular malformations, including lymphatic (LMs) and venous malformations (VMs), are chronic, often debilitating conditions that arise during embryonic development (Pang et al, 2020; Castillo et al, 2019; Limaye et al, 2009). They frequently manifest in the head and neck region—here defined as the orofacial and cervical areas, excluding the brain. In many reports, up to ~80% of lymphatic malformations, ~90% of capillary malformations, and ~40% of venous malformations occur there, although these estimates vary depending on the patient population and classification system (Zenner et al, 2019; Lee and Chung, 2018; Nair, 2018; Alsuwailem et al, 2020). In severe cases, such lesions compromise vital functions like breathing and feeding and can significantly affect appearance. Consequently, head and neck vascular malformations are particularly difficult to treat and have been classified as intractable diseases, distinguishing them from those occurring elsewhere in the body (Damme et al, 2020; Mimura et al, 2020; Nair, 2018; Sadick et al, 2017; Alsuwailem et al, 2020). Standard treatments, including surgical excision and sclerotherapy, are often employed but not always effective. Thus, there is an urgent need for novel therapies that target the underlying disease mechanisms. Vascular malformations are generally categorized into low-flow (venous, lymphatic, and capillary) and fast-flow (arteriovenous) lesions, with low-flow malformations being the most common (Mimura et al, 2020; Kobialka et al, 2022).

*PIK3CA* mutations are found in ~85% of LMs (Boscolo et al, 2015a; Luks et al, 2015; Mäkinen et al, 2021; Zenner et al, 2019) and 20% of VMs (Limaye et al, 2015; Hirose et al, 2024; Castel et al, 2016; Castillo et al, 2016), though this varies between studies. PIK3CA encodes the p110α lipid kinase, responsible for converting Phosphatidylinositol 4,5-bisphosphate (PIP2) to Phosphatidylinositol 3,4,5-trisphosphate (PIP3), which activates key downstream signaling pathways. This kinase is crucial for the development of

[1]Department of Pathology and Matrix Biology, Graduate School of Medicine, Mie University, 2-174 Edobashi, Tsu, Mie 514-8507, Japan. [2]Department of Hematology and Oncology, Mie University Graduate School of Medicine, 2-174 Edobashi, Tsu 514-8507, Japan. [3]Department of Surgery, Kanagawa Children's Medical Center, 2-138-4Mutsukawa, Minami-ku, Yokohama, Kanagawa, Japan. [4]Department of Pathology, Osaka University Graduate School of Medicine, 2-2 Yamadaoka, Suita, Osaka 565-0871, Japan. [5]Department of Central Laboratory and Surgical Pathology, NHO Osaka National Hospital, 2-1-14 Hoenzaka, Chuo-ku, Osaka 540-0006, Japan. [6]Department of Oral and Maxillofacial Pathology, Osaka University Graduate School of Dentistry, 1-8 Yamadaoka, Suita, Osaka 565-0871, Japan. [7]Department of Plastic and Reconstructive Surgery, Mie University Graduate School of Medicine, 2-174 Edobashi, Tsu, Mie 514-8507, Japan. [8]Department of Anatomy, Keio University School of Medicine, 35 Shinanomachi, Shinjuku-ku, Tokyo 160-8582, Japan. [9]Department of Molecular Physiology, National Cerebral and Cardiovascular Center Research Institute, 6-1 Kishibe-shimmachi, Suita, Osaka 564-8565, Japan. ✉E-mail: k-maruyama0608@med.mie-u.ac.jp

blood and lymphatic vessels (Graupera et al, 2008; Gupta et al, 2007; Stanczuk et al, 2015), and is activated by receptor tyrosine kinases, including vascular endothelial growth factor (VEGF) receptors and the TIE2/TEK receptor. Studies in mice have shown that p110α is essential for normal blood and lymphatic vessel development (Graupera et al, 2008; Gupta et al, 2007; Stanczuk et al, 2015). Conversely, expression of the *PIK3CA*[H1047R] mutation in endothelial cells (ECs) has been shown to cause vascular malformations (Blasio et al, 2018; Mäkinen et al, 2021).

VMs and LMs are frequently driven by mutations in the helical (E542K, E545K) and kinase (H1047R, H1047L) domains of p110α, identical to those observed in cancer and genetic syndromes marked by tissue overgrowth, such as *PIK3CA*-related overgrowth spectrum (PROS) (Madsen et al, 2018; Keppler-Noreuil et al, 2015). These somatic *PIK3CA* mutations activate key downstream pathways, most notably the AKT-mTOR pathway, which is thought to play a significant role in disease progression. However, the exact mechanisms remain unclear, with additional pathways, such as ERK and transforming growth factor-α (TGF-α), also being implicated (Gomes et al, 2022; Blesinger et al, 2018; Jauhiainen et al, 2023; Aw et al, 2023; Broek et al, 2019). The effects of these mutations may not be limited to ECs because abnormal ECs can interact with surrounding tissues, potentially driving fibrosis (Jauhiainen et al, 2023; Blesinger et al, 2018). While mTOR inhibitors like rapamycin and its analogs (sirolimus, everolimus) have been effective in reducing the symptoms of vascular malformations, complete lesion regression is rare. This suggests the involvement of additional pathways beyond mTOR, underscoring the need for more potent therapies (Seront et al, 2019; Blasio et al, 2018; Rodriguez-Laguna et al, 2019; Boscolo et al, 2015b; Maruani et al, 2021).

Our research highlights the unique pathophysiology of vascular malformations, particularly their development in the head and neck, where ECs arise from the *Islet1* (*Isl1*)[+] cardiopharyngeal mesoderm (CPM) (Maruyama et al, 2019, 2022; Yamaguchi et al, 2024; Lioux et al, 2020).

Herein, we demonstrated that the expression of Pik3ca[H1047R] in the CPM generates a model mouse that closely mimics human pathologies. Notably, although the CPM differentiates into a broad range of cell types, the effects of Pik3ca[H1047R] were specifically confined to veins and lymphatic vessels. Single-cell RNA sequencing (scRNA-seq) of FACS-sorted eYFP[+] cells from *Isl1-Cre; Pik3ca*[H1047R]; *R26R-eYFP* embryos and Bulk RNA-seq of PIK3CA[H1047R]-expressing human umbilical vein ECs (HUVECs) and patient-derived ECs revealed hypoxia-driven metabolic shifts and enhanced angiogenesis, including elevated Vegf-A production, contributing to vascular malformation pathogenesis. Immunohistochemical analysis of human vascular malformation samples confirmed the elevated HIF-1α and VEGF-A levels in abnormal vessels, reinforcing this hypothesis. Additionally, inhibiting HIF-1α and VEGF-A signaling in our mouse model significantly suppressed lesion formation, suggesting a potential therapeutic strategy for these intractable conditions.

This study offers new insights into the anatomical and molecular mechanisms underlying vascular malformations, highlighting the significance of developmental origins and hypoxia-mediated signaling pathways. These findings provide a promising foundation for developing novel therapeutic strategies aimed at targeting these complex and currently untreatable vascular anomalies.

# Results

## Expression of Pik3ca[H1047R] in endothelial cells recapitulates human low-flow vascular malformations

To assess whether Pik3ca[H1047R] expression can recapitulate the VMs and LMs observed in humans, we utilized a transgenic mouse model, LoxP-STOP-LoxP–Pik3ca[H1047R] (*R26R-Pik3ca*[H1047R]). This model enables tissue-specific activation of Pik3ca[H1047R] through Cre-loxP technology. By inducing Pik3ca[H1047R] during embryonic development using various Cre-expressing strains, we observed that crossing with *Tie2-Cre*, which drives Cre expression in ECs, resulted in widespread dilated, blood-filled vasculature by embryonic day (E)13.5 (Fig. 1A). Sagittal section co-immunostaining at E13.5 with EC markers, namely platelet endothelial cell adhesion molecule (PECAM) and VEGF receptor 3 (VEGFR3), as well as the lymphatic marker Prospero homeobox protein 1 (Prox1), revealed a dilated cardinal vein surrounded by an increased number of blood-filled PECAM[+]/Prox1[+]/VEGFR3[+] lymphatic vessels in *Tie2-Cre; R26R-Pik3ca*[H1047R] mutant embryos (Fig. 1B–F). Although less prominent than in the cardinal vein, dilated PECAM[+]/Prox1[−]/partially VEGFR3[+] veins were also observed extending from the pharyngeal arches to the mandible in mutant embryos (Fig. 1B,G–I). The overall number of PECAM[+] vasculatures did not significantly differ between control and mutant embryos (Fig. 1J). In the liver, we observed an increase in spongy, dilated PECAM[+]/Prox1[−]/partially VEGFR3[+] veins connected to the inferior vena cava (Fig. 1B,K–N). Additionally, in the brain, there was an increase in tortuous and slightly dilated PECAM[+]/Prox1[−]/VEGFR3[+] blood vessels (Fig. 1B,O–R).

At E12.5 in *Tie2-Cre; R26R-Pik3ca*[H1047R]; *R26R-eYFP* mutant embryos, as observed at E13.5, macroscopic examination revealed blood-filled patchy lesions (Fig. EV1A). The widespread expression of eYFP across the body in mutant mice indicates a high recombination efficiency of *Tie2-Cre* (Fig. EV1A). In sagittal tissue sections, phospho-S6 (pS6) staining, an indicator of PI3K activity, revealed enhanced signaling in the PECAM[+] cardinal vein and surrounding Prox1[+] lymphatic ECs (LECs) in mutant embryos compared with controls (Fig. EV1B–B''',C–C'''). Analysis of cellular proliferation using the Ki67 antibody showed a higher number of positive cells in the mutant cardinal vein than in the control cardinal vein (Fig. EV1B'''',C'''',L). Furthermore, most ECs in the mutant cardinal vein were eYFP[+], indicating high recombination efficiency by Cre recombinase (Fig. EV1B''''',C''''').

Next, we performed morphological analyses of the pulmonary artery, aorta, and heart. No significant differences were observed in the luminal diameters of the PECAM[+] pulmonary artery, descending aorta, or cardiac morphology, including the PECAM[+] endocardium, when comparing mutant and control embryos (Fig. EV1B,C,D–E'',M,N). Both control and mutant ECs in the pulmonary artery, descending aorta, and endocardium exhibited pS6 expression, with no significant differences between them (Fig. EV1F–G''). Similarly, the number of Ki67[+] ECs in the pulmonary artery, descending aorta, and endocardium did not differ between the control and mutant embryos (Fig. EV1H–I'',O–Q). The widespread eYFP expression in ECs of the pulmonary artery, descending aorta, and endocardium confirmed the high recombination efficiency of Cre recombinase, indicating that the lack of arterial phenotypes was not due to poor recombination (Fig. EV1J–K''). Whereas clear phenotypes were evident at E12.5 and E13.5, no pronounced external abnormalities were observed at E9.5 or

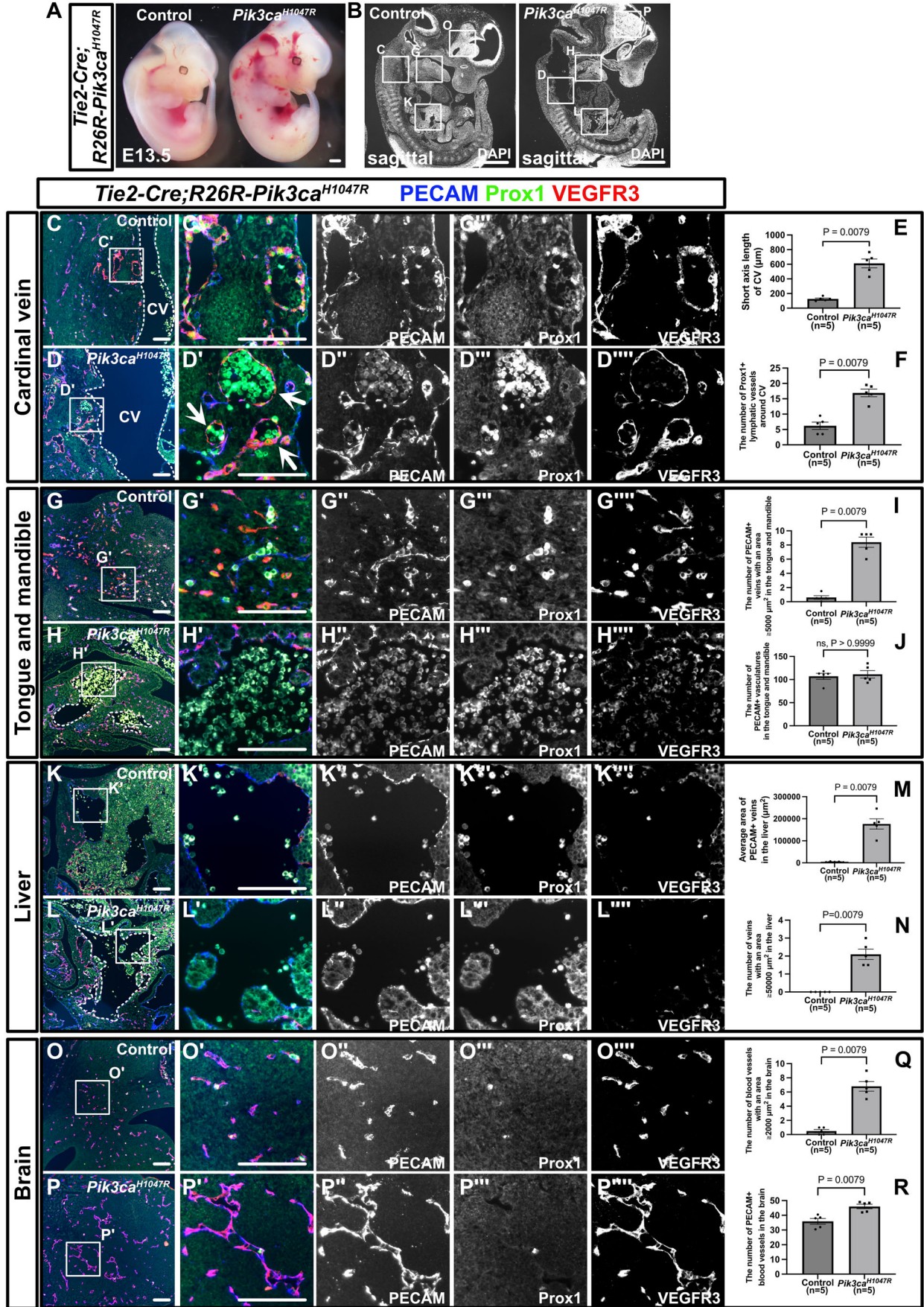

◀ **Figure 1.** *Tie2-Cre; R26R-Pik3ca^H1047R^ embryos mimic human venous and lymphatic malformations.*

(A) Gross morphology of control and *Tie2-Cre; R26R-Pik3ca^H1047R^* mutant embryos at E13.5. Comparisons were made between control and mutant embryos ($n = 5$ per group). (B, C–D'''', G–H'''', K–L'''', O–P'''') Immunostaining of sagittal sections with the indicated antibodies. (B–F) In the mutants, the cardinal vein (CV) is dilated (E), and there is an increase in blood-filled PECAM$^+$/ Prox1$^+$/VEGFR3$^+$ lymphatic vessels surrounding the CV (white arrows) (D). (B, G–J) In the mandible-tongue region, there is an expansion of blood-filled PECAM$^+$/Prox1$^-$/partially VEGFR3$^+$ blood vessels with an area greater than 5000 μm$^2$ (white dotted lines) (H). No significant increase in the total number of PECAM$^+$ vessels was observed (J). (B, K–N) In the liver, irregularly shaped, dilated PECAM$^+$/Prox1$^-$/partially VEGFR3$^+$ blood vessels with an area greater than 5000 μm$^2$ (white dotted lines) were observed (M, N). (O, P) Similarly, in the brain, irregularly shaped and dilated PECAM$^+$/Prox1$^-$/VEGFR3$^+$ blood vessels with an area greater than 2000 μm$^2$ were observed (Q, R). The region outlined by a box in the original image is displayed in an enlarged view in a corresponding panel. Each dot represents a value obtained from one sample. CV cardinal vein. Scale bars: 100 μm (C–D'''', G–H'''', K–L'''', O–P'''') and 1 mm (A, B). The nonparametric Mann–Whitney $U$ test was used for statistical analysis, with exact $P$ values indicated. ns ≥0.05. Error bars represent the mean ± standard error of the mean (SEM). Source data are available online for this figure.

E11.5 (Fig. EV2A,B). Similarly, histological examination revealed no significant differences in the short-axis diameter of the PECAM$^+$ CV or in the number of Prox1$^+$ LECs surrounding the CV between control and mutant embryos at E11.5 (Fig. EV2C–F). We also assessed *Tie2-Cre; R26R-Pik3ca^H1047R^* mutant embryos at E14.0 from five pregnant mice. Only two embryos were alive at this stage, and both showed severe edema and hemorrhaging, indicating they were nearly moribund. These observations suggest that the critical point for survival of these mutant embryos lies between E13.5 and E14.0 (Fig. EV2G).

## Pik3ca^H1047R^ activation timing influences vascular malformation subtypes but does not fully replicate human disease

To investigate whether the resulting human disease subtype (e.g., lesions confined to the head and neck region) is determined by the specific embryonic stage at which Pik3ca$^{H1047R}$ is expressed, we crossed tamoxifen-inducible, pan-endothelial *CDH5-CreERT2* mice with *R26R-Pik3ca^H1047R^* mice and analyzed the embryos at E16.5 or E17.5. At these stages, lymphatic vessels are considered fully developed throughout the body (Srinivasan et al, 2007; Maruyama et al, 2022). We focused our analysis on the tongue, neck, liver, skin, and mesentery—regions frequently affected by vascular malformations in humans. Since the standard tamoxifen dose (125 mg/kg body weight) leads to miscarriage or embryonic death within 1–2 days, we diluted it to one-fifth of the original concentration. When tamoxifen was administered to pregnant mice at E9.5 and analyzed at E16.5, *CDH5-CreERT2; R26R-Pik3ca^H1047R^* mutant embryos exhibited widespread, abnormal blood-filled vasculatures throughout the body (Fig. 2A,B,I). In sagittal sections, the mutant mice showed an increase in slightly dilated, blood-filled PECAM$^+$/Prox1$^+$/VEGFR3$^+$ lymphatic vessels in the tongue (Fig. 2C–D',J–K'). In mutant mice, the number of blood-filled PECAM$^+$/Prox1$^+$/VEGFR3$^+$ lymphatic vessels surrounding the dilated jugular vein increased (Fig. 2E,E',L,L',AD), and dilated PECAM$^+$/ Prox1$^-$/partially VEGFR3$^+$ blood vessels were observed in the liver (Fig. 2F,F',M,M'). Additionally, in the dorsal skin and mesentery, mutant mice exhibited an increase in dilated PECAM$^+$/Prox1$^-$/ VEGFR3$^-$ blood vessels and blood-filled PECAM$^+$/Prox1$^+$/VEGFR3$^+$ lymphatic vessels (Fig. 2G–H',N–O',AD). Following tamoxifen administration at E12.5, blood-filled vessels were less prominent, though notable edema was observed in mutant embryos (Fig. 2P,W). In mutant embryos, PECAM$^+$/Prox1$^-$/partially VEGFR3$^+$ enlarged blood vessels were increased in the tongue (Fig. 2Q–R',X–Y',AE). No significant differences were found in the PECAM$^+$/Prox1$^+$/VEGFR3$^+$ lymphatic vessels in the neck, but mild dilation was observed in the PECAM$^+$/Prox1$^-$/VEGFR3$^-$ jugular vein in mutant embryos compared with controls (Fig. 2S,S',Z,Z',AE), and although the dilation was

less extensive than at E9.5, dilated PECAM$^+$/Prox1$^-$/partially VEGFR3$^+$ blood vessels were observed in the liver (Fig. 2T,T',AA,AA'). Slightly dilated, blood-filled PECAM$^+$/Prox1$^+$/VEGFR3$^+$ lymphatic vessels were seen in the mesentery and skin of mutant mice, though the dilation was less pronounced than that observed at E9.5 (Fig. 2U–V',AB–AC',AE). Mutants showed less pronounced edema when tamoxifen was administered to pregnant mice at E15.5 and analyzed at E17.5 than administration at E9.5 or E12.5, with only a few blood-filled vessels observed (Fig. EV3A). In mutant embryos, dilated PECAM$^+$/Prox1$^-$/VEGFR3$^+$ blood vessels and PECAM$^-$/Prox1$^+$/ VEGFR3$^+$ lymphatic vessels proliferated at the tip of the tongue (Fig. EV3B–B'',G–G''). Blood-filled, dilated PECAM$^+$/partially Prox1$^+$/ partially VEGFR3$^+$ lymphatic vessels were observed in the skin and mesentery (Fig. EV3C–C'',H–H'',E–E',J–J''). Additionally, dilated PECAM$^+$/Prox1$^-$/VEGFR3$^-$ blood vessels were observed in the liver of mutant mice (Fig. EV3F–F'',K–K''), but no significant differences were found in the neck region compared with controls (Fig. EV3D–D'',I–I'').

These results indicate that even though the timing of Pik3ca$^{H1047R}$ activation in ECs partially influences the location and extent of vascular malformations, it does not fully explain their localized nature in humans. In general, later induction of Pik3ca$^{H1047R}$ leads to a more restricted anatomical distribution and fewer affected vessels than earlier induction.

## Expression of Pik3ca^H1047R^ in the CPM replicates the anatomical features of human vascular malformations

The distribution of lymphatic and blood vessels derived from the CPM aligns with the common sites of refractory vascular malformations in humans (Maruyama et al, 2022). Therefore, we hypothesized that expressing Pik3ca$^{H1047R}$ in the CPM could generate a mouse model that mimics human vascular malformations. To test this hypothesis, we crossed *Isl1-Cre;R26R-eYFP* mice, which express Cre recombinase under the control of the *Isl1* promoter and label CPM derivatives, with *R26R-Pik3ca^H1047R^* mice. Embryos of the resulting *Isl1-Cre;R26R-Pik3ca^H1047R^;R26R-eYFP* mice were analyzed at E11.5. In the mutant embryos, blood-filled vessels were observed in the first and second branchial arches, but no other significant external differences from the controls were noted (Fig. 3A). eYFP expression was observed in the pharyngeal mesodermal region (Fig. 3A). Sagittal sections of mutant embryos revealed enlarged PECAM$^+$/Prox1$^-$/VEGFR3$^+$ blood vessels, predominantly in the first and second branchial arches (Fig. 3B–B',C–C',K). pS6 expression was detected in both mesenchymal cells and ECs within the branchial arches of control and mutant embryos (Fig. 3B'',C''). Additionally, eYFP was present

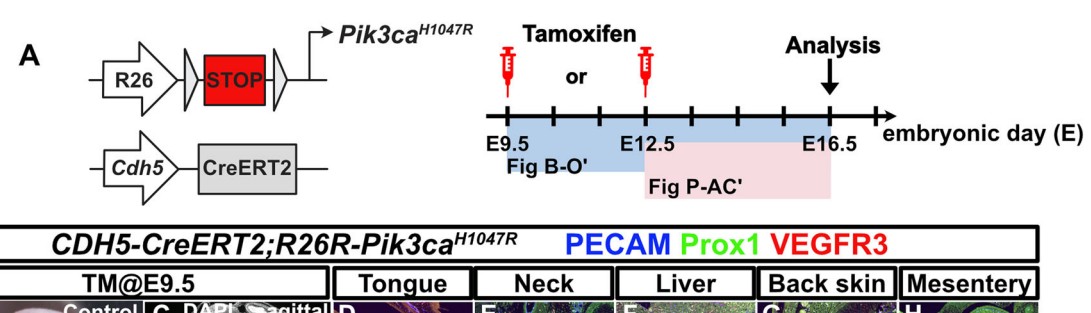

◄ **Figure 2. Pik3ca^H1047R expression timing influences the location of vascular malformations but does not fully recapitulate human pathology.**

(A) Overview of the experiments and the tamoxifen administration schedule. (B, I) Gross morphology of the control and *CDH5-CreERT2; R26R-Pik3ca^H1047R* mutant embryos at E16.5, after tamoxifen administration at E9.5. Comparisons were made between control and mutant embryos (*n* = 5, 3 for each group). (C–H', J–O') Immunostaining of sagittal sections with the indicated antibodies. (C, C', J, J') DAPI-stained sections of the same specimen. (E, E', L, L') In the neck region of mutant embryos, dilated jugular veins are visible (L, white dotted lines). Compared with controls, smaller blood-filled PECAM^+/Prox1^+/VEGFR3^+ lymphatic vessels (L', white arrows) and PECAM^+/ Prox1^-/VEGFR3^- blood vessels (L', red arrows) are also seen. (F, F', M, M') In the liver, dilated PECAM^+/Prox1^-/partially VEGFR3^+ blood vessels are observed (M, white dotted lines). The number of PECAM^+/Prox1^-/partially VEGFR3^+ blood vessels with an area ≥20,000 μm² in the liver is 0 ± 0 (median ± SEM) (*n* = 5) in controls and 3 ± 0.33 (median ± SEM) (*n* = 3) in mutants (*P* = 0.0179). (G–H', N–O') Similar observations are made in the skin and mesentery, where blood-filled PECAM^+/Prox1^+/ VEGFR3^+ lymphatic vessels (N', O', white arrows) and dilated PECAM^+/Prox1^-/VEGFR3^- blood vessels (N', O', red arrows) are seen. (P, W) Gross morphology of the control and *CDH5-CreERT2; R26R-Pik3ca^H1047R* mutant embryos at E16.5, after tamoxifen administration at E12.5. Comparisons were made between control and mutant embryos (*n* = 5, 3 for each group). Compared with embryos treated at E9.5, blood-filled dilated vessels are less prominent, but generalized edema is more severe. (Q, Q', X, X') DAPI-stained sections of the same specimen. (R–V', Y–AC') Immunostaining of sagittal sections with the indicated antibodies. (T, T', AA, AA') In the liver, dilated PECAM^+/Prox1^-/partially VEGFR3^+ blood vessels are observed (AA, AA', white dotted lines). The number of PECAM^+ vessels with an area ≥20,000 μm² in the liver is 0 ± 0 (median ± SEM) (*n* = 5) in controls and 3 ± 0.29 (median ± SEM) (*n* = 3) in mutants (*P* = 0.0179). (U–V', AB–AC') Dilated, blood-filled PECAM^+/weak Prox1^+/ VEGFR3^+ lymphatic vessels (AB', AC', white arrows) are observed in the skin and mesentery. (AD, AE) Statistical analysis of the tongue, neck, skin, and mesentery. Scale bars: 100 μm (D–H', K–O', R–V', Y–AC'), 1 mm (C, C', J, J', Q, Q', X, X'), and 2 mm (B, I, P, W). The nonparametric Mann–Whitney *U* test was used for statistical analysis, with exact *P* values indicated. ns ≥0.05. Panels C and C', J and J', Q and Q', and X and X' depict identical images; the duplicate panels are intentionally included for clarity and comparative purposes. The region outlined by a box in the original image is displayed in an enlarged view in a corresponding panel. Error bars represent the mean ± standard error of the mean (SEM). Source data are available online for this figure.

not only in ECs but also in mesenchymal cells within the branchial arches (Fig. 3B''',C'''). Similar expanded PECAM^+/Prox1^-/partially VEGFR3^+ blood vessels were observed in the PA1-PA2 region of mutant embryos when *Mef2c-AHF-Cre* mice, another line that labels CPM, were crossed with *R26R-Pik3ca^H1047R* and analyzed at E11.5 (Fig. EV4A–E). In *Isl1-Cre; R26R-Pik3ca^H1047R* embryos at E13.5, expanded vasculature was observed from the head to the neck (Fig. 3D,D'). Sagittal sections revealed dilated vessels extending from the mandible to the tongue, along with expanded lymphatic vessels and large, blood-filled lymph sacs in mutant embryos (Fig. 3E–J'',L,M). These *Isl1-Cre; R26R-Pik3ca^H1047R* mutant embryos likely died from facial hemorrhaging between E13.5 and E14.0 (Fig. 3N). Next, *Isl1-CreERT2; R26R-Pik3ca^H1047R* embryos were analyzed at E13.5 after tamoxifen administration to pregnant mice at E8.5. No significant external differences were observed compared with controls (Fig. EV4F). eYFP expression was detected along the lower jaw, neck, and genital regions (Fig. EV4F'). Histological sections revealed irregular, dilated PECAM^+/Prox1^+/ VEGFR3^+ lymphatic vessels at the junction of the tongue and lower jaw, while the number of Prox1^+ cells remained unchanged between control and mutant embryos, suggesting that lymphatic vessel dilation may underlie the observed phenotype (Fig. EV4G–G''',H–H''',I,J). eYFP^+ cells were confirmed to contribute to ECs of these dilated lymphatic vessels (Fig. EV4G'''',H'''').

The head and neck mesoderm arises primarily from the cardiopharyngeal mesoderm and the cranial paraxial mesoderm. In *Pax3-CreERT2; R26R-Pik3ca^H1047R* embryos, *Pax3^+* paraxial mesoderm (including cranial paraxial mesoderm) is labeled; this lineage reportedly contributes to the common cardinal vein and subsequently forms trunk lymphatics (Lupu et al, 2025), tamoxifen was administered to pregnant mice at E9.0. Upon analysis at E14.0, no vascular malformations were observed in the head or neck regions (Fig. EV4K–M,O–Q). However, mutant embryos exhibited an increased number of dilated, blood-filled PECAM^+/partially Prox1^+/VEGFR3^+ abnormal vessels around the spine (Fig. EV4N–N'',R–R'',S,T). In *Myf5-CreERT2; R26R-tdTomato* mice—which label the cranial paraxial mesoderm, particularly muscle satellite cells—crossed with *R26R-Pik3ca^H1047R*, tamoxifen was administered to pregnant mice at E9.5. Upon analysis at E14.5, no external differences were observed between the control and

mutant embryos (Fig. EV4U). tdTomato expression was observed in the face and limbs (Fig. EV4U'). Histological analysis showed no major changes in sagittal sections, and tdTomato^+ cells contributed to tongue skeletal muscle without hypertrophy or hyperplasia in mutants. Additionally, PECAM^+ vasculature showed no significant differences between the controls and mutants (Fig. EV4V–X).

These findings suggest that ECs are susceptible to the effects of the *Pik3ca^H1047R* mutation and that their cellular origin dictates the anatomical site where vascular malformations develop.

## Single-cell RNA sequencing analysis elucidates the endothelial cell differentiation process from the CPM

To clarify the causes of diverse vascular malformations in the head and neck and establish a foundation for understanding how ECs differentiate from CPM, we re-analyzed previously published scRNA-seq data (Nomaru et al, 2021). Specifically, we analyzed the differentiation process of *Mesp1^+* CPM cells into ECs. UMAPs were generated from *Mesp1^+* CPM populations at E8.0, E8.25, E9.5, and E10.5 (Fig. EV5A,B), identifying EC clusters expressing various markers, such as *Pecam1*, *Kdr*, *Cdh5*, *Tek*, and *Etv2* (Clusters 7 and 15), whereas *Isl1* expression was not highly detected (Fig. EV5A–D; Dataset EV1). Further subclustering of these two clusters revealed eight subclusters, labeled 0 through 7 (Fig. EV5E–H; Dataset EV1). Additionally, RNA velocity and trajectory inference analyses were performed to predict the differentiation pathways of ECs (Fig. EV5I). The results suggested a lineage progression from *Isl1^+* undifferentiated mesodermal cells to *Npr3^+* endocardial cells (Zhang et al, 2023) or *Etv2^+* EC progenitor cells (PCs) (Fig. EV5I,J). *Prox1^+* LEC PCs appear to arise directly from *Etv2^+* EC progenitors, bypassing the venous stage, whereas *Aplnr^+* venous ECs and *Efnb2^+* arterial ECs likely develop from shared arterial/venous PCs (A/V PCs) (Fig. EV5J).

To further investigate the gene expression changes that drive the differentiation of immature ECs from undifferentiated mesodermal cells, we conducted a more detailed analysis using only the data from E8.0 and E8.25. When projected onto UMAP, the data revealed 15 distinct clusters (Fig. EV6A–C; Dataset EV2). Widespread expression of the CPM markers *Isl1* and

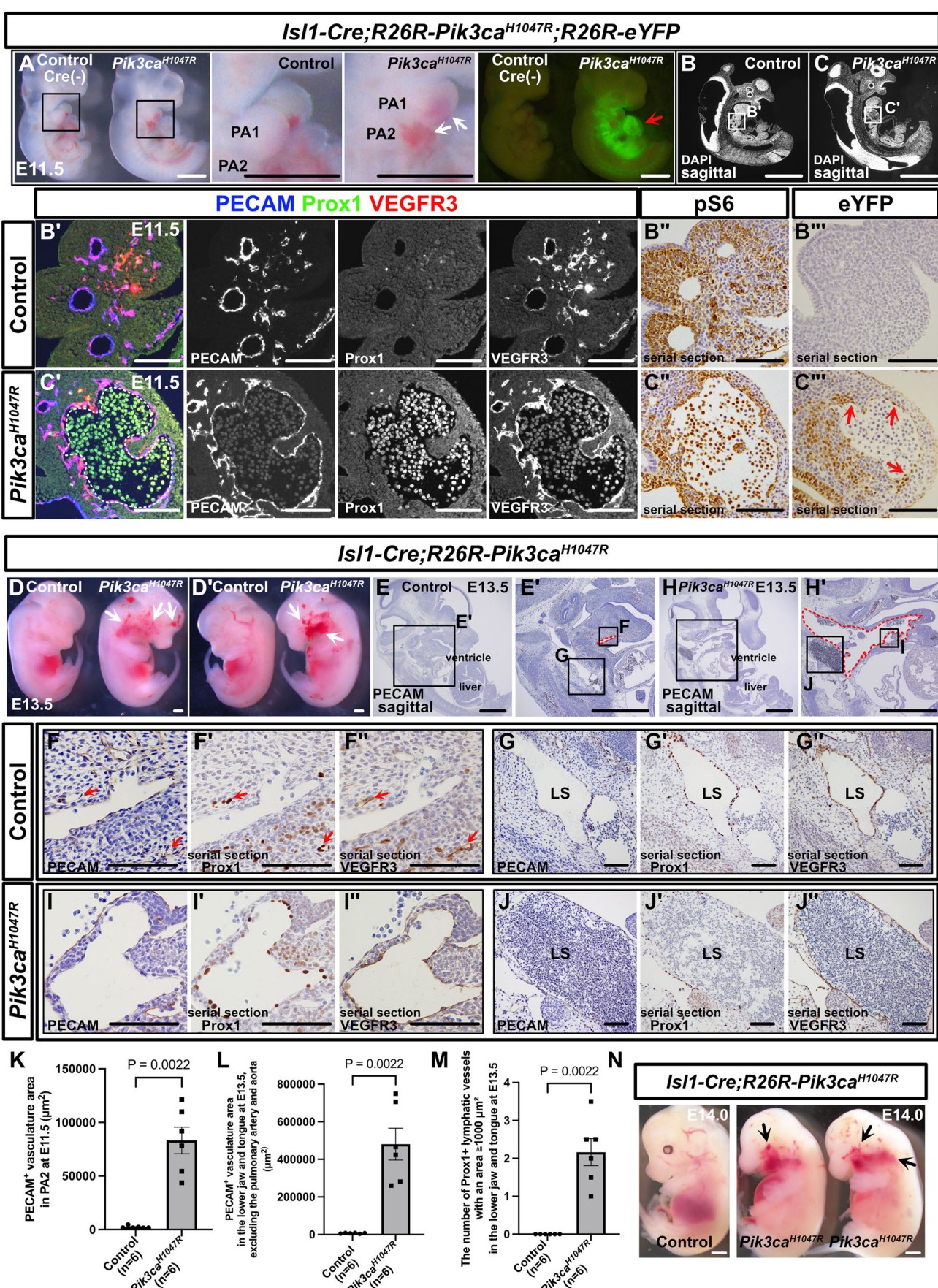

◀

**Figure 3. Pik3ca^H1047R^ expression in *Isl1*^+^ CPM leads to vascular malformations confined to the head and neck region.**

(A) Gross morphology of the control and *Isl1-Cre;R26R-Pik3ca^H1047R^;R26R-eYFP* mutant embryos at E11.5. Comparisons were made between control and mutant embryos (n = 6, 6 for each group). Mutant mice show widespread blood-filled, dilated vessels in the first and second pharyngeal arches, along with eYFP expression in the pharyngeal mesoderm (red arrows). (B–C‴) Immunostaining of sagittal sections with the indicated antibodies. Mutant mice exhibit dilated, blood-filled PECAM^+^/Prox1^-^/ partially VEGFR3^+^ blood vessels in the second pharyngeal arch (B–B′, C–C′, white dotted lines). (B″, B‴, C″, C‴) Enzyme-antibody staining reveals pS6 expression in both endothelial and mesenchymal cells. eYFP, detected by GFP antibody, is expressed in ECs of dilated vessels (red arrows). (D, D′) Gross morphology of the control and *Isl1-Cre;R26R-Pik3ca^H1047R^* mutant embryos at E13.5. Comparisons were made between control and mutant embryos (n = 6, 6 for each group). Mutant embryos display extensive blood-filled, dilated vessels on both the left and right sides of the head and neck region (white arrows). (E–J″) Enzyme-antibody staining of sagittal sections. In mutants, large PECAM^+^ vessels extend from the lower jaw to the jugular vein (E–H′, red dotted lines). Controls show small partially PECAM^+^/Prox1^+^/VEGFR3^+^ lymphatic vessels in the lower jaw, and mutants exhibit enlarged PECAM^+^/Prox1^+^/VEGFR3^+^ lymphatic vessels and blood-filled lymph sacs (F–J″, red arrows). (K–M) Statistical analysis. Each dot represents a value from one sample. (N) Gross morphology of control and *Isl1-Cre; R26R-Pik3ca^H1047R^* mutant embryos at E14.0. The mutant embryos exhibit hemorrhaging in the head and neck region (black arrows), and all mutants had died by this stage. The study was performed using five pregnant mothers. PA pharyngeal arch, LS lymph sac. The region outlined by a box in the original image is displayed in an enlarged view in a corresponding panel. Scale bars: 100 μm (B′–C‴, F–J″) and 1 mm (A–C, D, D′, E, E′, H, H′, N). The nonparametric Mann–Whitney *U* test was used for statistical analysis. Error bars represent the mean ± standard error of the mean (SEM). Source data are available online for this figure.

*Wnt5a* was observed (Fig. EV6D). We focused on clusters 4, 11, 12, and 3, which contained immature ECs, as indicated by markers such as *Etv2, Kdr* (Fig. EV6C,D; Dataset EV2). Subclustering analysis of these clusters revealed seven subclusters, labeled 0 through 6 (Fig. EV6E–H; Dataset EV2). RNA velocity and trajectory analyses revealed early expression of *Isl1* and *Wnt5a*, which decreased as EC differentiation progressed. This was followed by the expression of key endothelial differentiation markers, *Etv2* and *Kdr*, and subsequently by markers characteristic of a later stage of endothelial differentiation (Val and Black, 2009; Morita et al, 2015), such as *Flt4* and *Pecam1* (Fig. EV6E–J).

### *Pik3ca*-driven vascular malformations are associated with hypoxia-mediated metabolic changes and Vegf-a expression

To investigate how Pik3ca^H1047R^ expression in the CPM affects ECs and drives gene expression changes contributing to vascular malformations, we conducted scRNA-seq analysis on *Isl1-Cre; R26R-Pik3ca^H1047R^; R26R-eYFP* mutant embryos and *Isl1-Cre; R26R-eYFP* controls at E13.5, the stage when lymphatic vessels form lumens in the head and neck. After FACS sorting of eYFP^+^ cells, the sorted cells were analyzed (Figs. 4A and EV7A). UMAP analysis identified 15 clusters, with cluster 1 defined as ECs (Fig. 4B,C; Dataset EV3). In mutant mice, the proportion of ECs increased compared with controls, while the proportion of mesenchymal cells decreased (Fig. 4B,D; Dataset EV3). Enrichment analysis highlighted significant changes in hypoxia and glycolysis pathways, which were observed not only in ECs but also in cardiomyocytes, pharyngeal arch muscles, neurons, epithelium, and mesenchyme (Figs. 4E and EV7B; Datasets EV3 and 4). Bulk RNA-seq analysis of HUVECs expressing PIK3CA^H1047R^ and ECs from patients with *PIK3CA* gain-of-function mutations also showed elevated hypoxia and glycolysis pathways (Figs. 4E and EV7C; Datasets EV3 and 4). The distribution of cells in the G1, G2/M, and S phases was similar across cell types, except for neurons, where over 80% were in the G1 phase (Fig. EV7D; Dataset EV4). Volcano plots for cluster 1 ECs highlighted genes associated with hypoxia and glycolysis (Fig. 4F,G; Dataset EV3). Several genes involved in hypoxia and glycolysis, including *Vegf-a, Flt1, Adm, Slc2a1, Pgk1, Tip1*, and *Pkm*, were upregulated in mutant embryo ECs. Correspondingly, hypoxia-inducible factor (*Hif*) stabilizing genes, such as *Nfe2l2, Atf4, Ddit*,

and *Pdk1*, were also upregulated in mutant EC clusters. Additionally, Hsp family members (*Hsp90b1, Hsp90aa1, Hspa1a, Hspa1b, Hspa5*), possibly associated with Hif stabilization, also showed increased expression (Fig. 4F; Dataset EV3). However, the expression levels of *Vegf-b, Vegf-c, Vegf-d, Kdr*, and *Flt4* were not significantly different between the control and mutant mice (Figs. 4G and EV7E; Datasets EV3 and 4).

In summary, mutant ECs showed increased expression of genes involved in hypoxia and glycolysis, including Hif target genes, such as *Vegf-a* and *Adm*, which stimulate angiogenesis and lymphangiogenesis. Glycolytic enzymes were also upregulated. Notably, *Pdk1*, which inhibits pyruvate conversion to acetyl-CoA, and lactate dehydrogenase A (*Ldha*) were elevated, indicating a shift toward anaerobic metabolism, similar to that seen in malignant tumors and actively proliferating cells (Fig. 4H; Dataset EV3).

### Elevated HIF-1α and VEGF-A expression is observed in *PIK3CA*-mutated human vascular malformations

To determine whether the signals identified by scRNA-seq were altered in human vascular malformation samples, we focused on HIF-1α and VEGF-A and confirmed protein expression by immunostaining. In control human epicardial tissue, PECAM^-^/ podoplanin^+^ LECs and PECAM^+^/podoplanin^-^ venous ECs did not express VEGF-A or HIF-1α. However, in PECAM^+^/podoplanin^-^ arterial ECs, mild expression of VEGF-A was observed in both the vessel wall and ECs, and HIF-1α was detected in these locations (Fig. 5A–C‴″). HIF-1α and VEGF-A were not detected in negative controls (Fig. 5A‴″,B‴″,C‴″). In LM samples with the *PIK3CA^H1047R^* mutation, particularly in clinically microcystic cases with extensive fibrosis and chronic inflammatory cell infiltration, irregularly structured PECAM^+^/partially podoplanin^+^ abnormal lymphatic vessels within fibrotic tissue showed partial expression of VEGF-A and HIF-1α (Fig. 5D–I). Nuclear expression of HIF-1α was observed (Fig. 5I). pS6 expression was detected in a subset of malformed ECs (Fig. 5J). No signal was detected in the negative control (Fig. 5K). A similar expression pattern was observed in macrocystic lymphatic vessels (Fig. EV8A–H). Comparable findings of HIF-1α, VEGF-A, and pS6 expression were noted in venous malformation patients with the same mutation (Fig. 5L–S) and in LM patients with *PIK3CA^E542K^* and *PIK3CA^E545K^* mutations (Fig. EV8I–X). Furthermore, to verify whether VEGF-A can act via VEGFR2, we performed VEGFR2 immunostaining on several

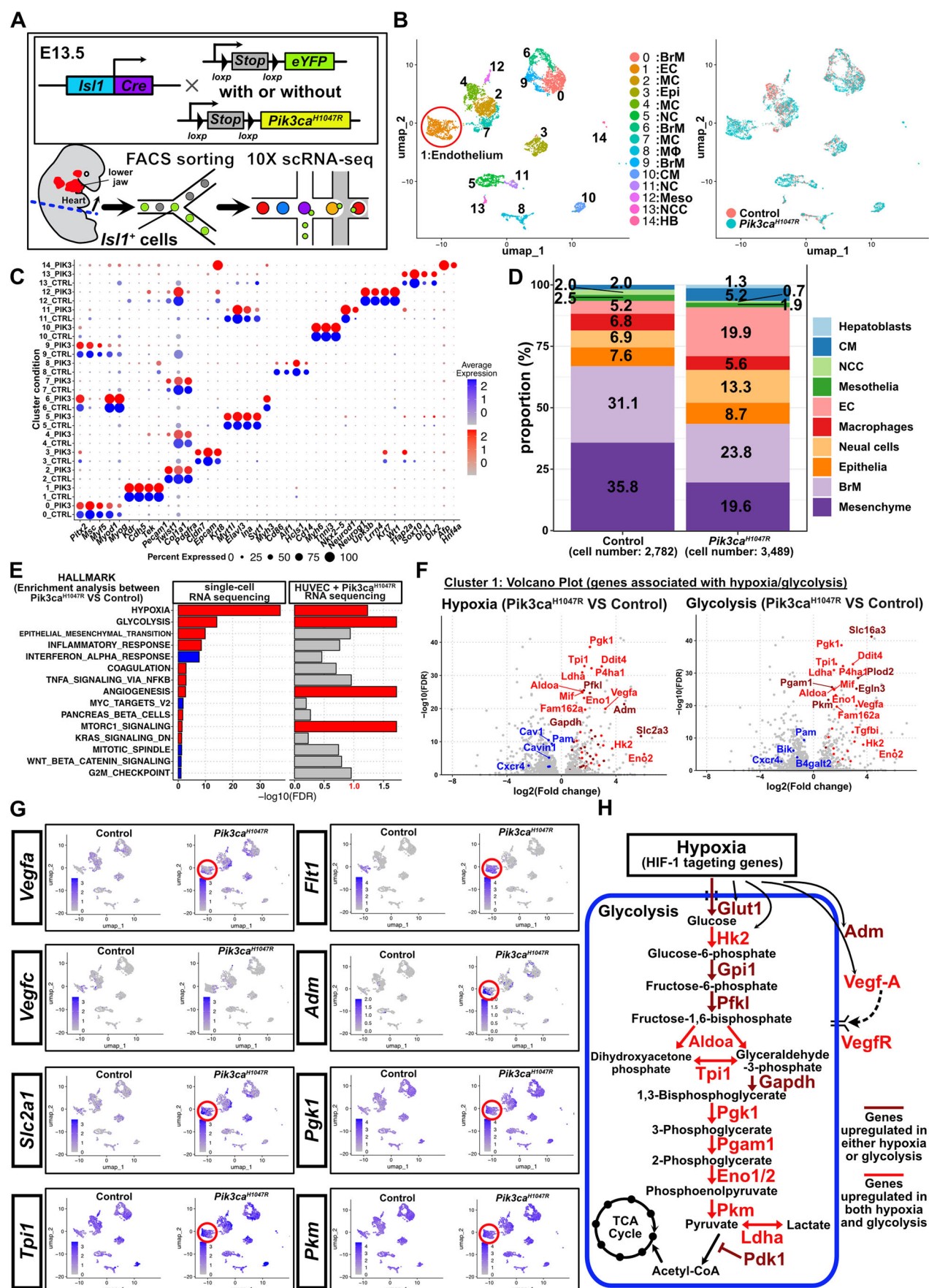

**Figure 4. Pik3ca$^{H1047R}$ induces hypoxia-mediated metabolic changes in endothelial cells.**

(A) Schematic overview of the cell isolation conducted for scRNA-seq. For each genotype (*Isl1-Cre; R26R-eYFP and Isl1-Cre; R26R-eYFP; R26R-Pik3ca$^{H1047R}$*), E13.5 embryos from three different embryos were pooled for single-cell library preparation. Metrics showed that 65.2% of reads were mapped to cells in the *Isl1-Cre; R26R-eYFP* group (mean 52,351 reads/cell, median 3850 genes, 4505 cells estimated) and 67.9% of reads were mapped to cells in the *Isl1-Cre; R26R-eYFP; R26R-Pik3ca$^{H1047R}$* group (mean 43,227 reads/cell, median 3529 genes, 5357 cells). Filtering based on gene count (2000–9000) and mitochondrial content (<10%) resulted in 2782 and 3489 cells for *Isl1-Cre; R26R-eYFP* and *Isl1-Cre; R26R-eYFP; R26R-Pik3ca$^{H1047R}$*, respectively, with 6271 cells included in downstream analyses. (B) Left: UMAP plot showing color-coded clusters (0–14). 0: branchial muscles (BrM); 1: endothelial cells (EC); 2: mesenchymal cells (MC); 3: epithelia (Epi); 4: MC; 5: neural cells (NC); 6: BrM; 7: MC; 8: macrophages (Mφ); 9: BrM; 10: cardiomyocytes (CM); 11: NC; 12: mesothelia (Meso); 13: neural crest cells (NCC); and 14: hepatoblasts (HB). Right: UMAP plot color-coded by condition. Pink represents the control, and light blue represents mutant cells. (C) Heatmap showing the average gene expression of marker genes for each cluster by condition. (D) Cell type proportions. (E) Comparison of enrichment analysis between EC clusters from scRNA-seq and bulk RNA-seq of HUVECs. The left bar graph shows the top 15 significantly altered Hallmark gene sets in EC clusters from scRNA-seq using ssGSEA (escape R package). The right bar graph shows the re-analysis of public bulk RNA-seq from PIK3CA$^{H1047R}$-transfected HUVECs. Red bars represent significantly upregulated Hallmark gene sets in mutants (FDR < 0.1), and gray bars indicate non-significant sets. (F) Left: Volcano plot showing hypoxia-related genes (red: mutant upregulated, blue: control upregulated, dark red: hypoxia only, bright red: hypoxia and glycolysis). Right: Glycolysis-related genes with the same color scheme. Differential expression is defined as FDR < 0.05 and fold change >1.5. (G) UMAP plot showing expression levels of genes related to hypoxia or glycolysis. Genes that were upregulated in the *Pik3ca$^{H1047R}$* mutant endothelial cell clusters are circled in red. (H) Schematic showing upregulated genes in scRNA-seq. Bright red indicates genes involved in both hypoxia and glycolysis, while dark red represents genes specifically upregulated in either the hypoxia or glycolysis pathway.

mouse models: *Tie2-Cre; R26R-Pik3ca$^{H1047R}$* embryos (E12.5, corresponding to Fig. EV1), *CDH5-CreERT2; R26R-Pik3ca$^{H1047R}$* embryos (tamoxifen administered at E9.5 and analyzed at E16.5, corresponding to Fig. 2), and *Isl1-Cre; R26R-Pik3ca$^{H1047R}$* embryos (E11.5 and E13.5, corresponding to Fig. 3). In all cases, both control and mutant embryos exhibited widespread VEGFR2 expression in blood and lymphatic vessels at early and late developmental stages (Fig. EV9A–R'). These findings suggest that Pik3ca$^{H1047R}$ may act in an autocrine manner, at least in part via the VEGF-A/VEGFR2 axis in endothelial cells, potentially explaining the observed phenotype.

## Hif-1α and Vegf-A inhibitors suppress the progression of vascular malformations

We next examined whether administering Hif-1α and Vegf-A inhibitors could effectively treat vascular malformations. Tamoxifen was administered to 3–4-week-old *CDH5-CreERT2; R26R-Pik3ca$^{H1047R}$* mice to induce the Pik3ca$^{H1047R}$ mutation in the dorsal skin. Anti-VEGF-A (a Vegf-A–neutralizing antibody), LW6 (a Hif-1α inhibitor), or rapamycin (an mTOR inhibitor) were then topically applied, and their effects were assessed (Fig. 6A). Both anti-VEGF-A and LW6 reduced the visible swelling in the dorsal skin, whereas rapamycin showed less difference compared with tamoxifen-treated but drug-free controls at day 7 (Fig. 6B). No changes were observed in skin regions that did not receive tamoxifen (Fig. 6B). At day 4 (before drug administration), we compared tamoxifen-untreated (Tam(−)) and tamoxifen-treated (Tam(+)) dorsal skin in *CDH5-CreERT2; R26R-Pik3ca$^{H1047R}$* mice by staining histological sections for PECAM and VEGFR3. In the Tam(+) group, both the overall number of PECAM$^+$ and VEGFR3$^+$ vessels and the number of enlarged vessels were higher compared with the Tam(−) group (Fig. 6C–H). By day 7, the Tam(−) group still did not exhibit any increase in PECAM$^+$ vessels (Fig. 6I,I'). To determine whether oil droplets from tamoxifen injections could induce vessel proliferation, we examined littermate *R26R-Pik3ca$^{H1047R}$* mice treated with tamoxifen but lacking *Cre* (Cre(-)/Tam(+)). No increase in PECAM$^+$ vessels was observed in these controls (Fig. 6J,J'). Next, we compared tamoxifen-treated *CDH5-CreERT2; R26R-Pik3ca$^{H1047R}$* mice that received one of the indicated drugs (Drug(+)) with those that did not (Drug(−)). Anti-VEGF-A and LW6 treatment both reduced the number of PECAM$^+$

vessels, whereas rapamycin did not cause a significant decrease compared with the untreated group (Fig. 6K–R). A similar pattern was seen for VEGFR3$^+$ lymphatic vessels: anti-VEGF-A and LW6 each led to a reduction, whereas rapamycin had no statistically significant effect (Fig. 6S–AB).

## Discussion

The common *PIK3CA*-activating mutation H1047R, which is linked to both VMs and LMs, as well as its association with cancer and PROS, has been well established (Yu et al, 2015b; Kobialka et al, 2022; Limaye et al, 2015; Luks et al, 2015; Castel et al, 2016; Castillo et al, 2016). However, the anatomical factors contributing to the complexity of treating vascular malformations and the changes in downstream signaling pathways remain insufficiently understood. Our analysis showed that Pik3ca$^{H1047R}$ expression in the CPM recapitulates the head and neck refractory vascular malformations. From scRNA-seq, we identified the significant upregulation of genes related to hypoxia and glycolysis in Pik3ca$^{H1047R}$-expressing ECs. Consistent with these findings, elevated HIF-1α and VEGF-A expression was observed in human samples of both LMs and VMs with various *PIK3CA* mutations. Additionally, topical application of Hif-1α and Vegf-A inhibitors in a *CDH5-CreERT2; R26R-Pik3ca$^{H1047R}$* skin model led to a significant reduction in lesion formation. In summary, our findings reveal hypoxia-driven molecular signaling changes in ECs harboring this mutation.

Our findings suggest that embryological cellular origins may play a crucial role in anatomically characterizing refractory vascular malformations. Although the CPM contributes to connective tissues, skeletal muscles, and cardiomyocytes, our study only reproduced vascular malformations, in contrast to the overgrowth observed in conditions like PROS. Similarly, Castillo et al found that early expression of Pik3ca$^{H1047R}$ in mesodermal cells resulted in VMs without organomegaly (Castillo et al, 2016). In contrast, human PROS often presents with organomegaly during embryonic development (Mirzaa et al, 2016). These discrepancies may stem from differences in the types of *PIK3CA* mutations (Brouillard et al, 2021) or tissue-specific sensitivities. For instance, *Pik3ca$^{H1047R}$* mutations replicate brain organomegaly in mouse models of PROS(Roy et al, 2015). Moreover, *PIK3CA* mutations have not been associated with arterial malformations (Peyre et al, 2021; Ren et al, 2021). Given that PI3K/AKT signaling is critical for arterial

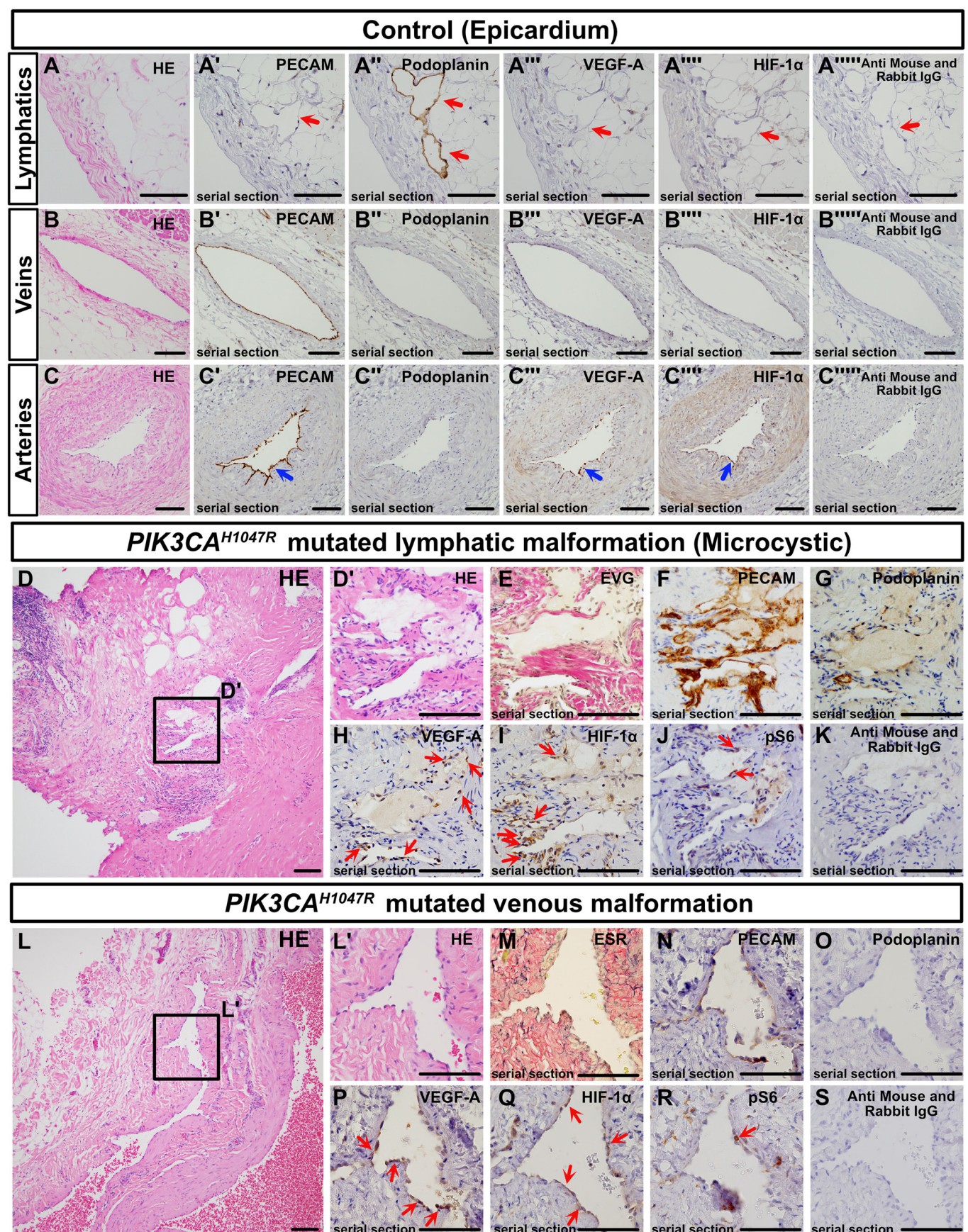

**Figure 5.  HIF-1α and VEGF-A expression in endothelial cells of malformed lymphatic vessels and veins in human patient samples.**

(A–S) Hematoxylin and eosin (HE) staining, special staining [elastica van Gieson (EVG), elastica Sirius red (ESR)—where collagen fibers appear red, elastic fibers black, and muscle tissue yellow], and immunohistochemistry using the indicated antibodies. No signal was detected in negative controls using only secondary antibodies (A''''', B''''', C''''', K, S). (A–A''''') Lymphatic vessels (red arrows) on the epicardium, characterized as PECAM⁻/podoplanin⁺, do not express HIF-1α or VEGF-A. (B–B''''') Similarly, PECAM⁺/podoplanin⁻ veins show no expression of these markers. (C–C''''') In PECAM⁺/podoplanon⁻ muscular arteries, both VEGF-A and HIF-1α are expressed in the arterial walls and endothelium (blue arrows). (D–K) Microcystic-type lymphatic malformations with the *PIK3CA^H1047R* mutation. Among the fibrotic tissue background (D), small, irregular PECAM-overexpressing vessels (E), partially podoplanin⁺ (G) lymphatic vessels are found. These vessels express VEGF-A and HIF-1α, and partially express pS6 (H–J, red arrows). (L–S) Venous malformation samples with the *PIK3CA^H1047R* mutation. These vessels have blood-filled lumens and thicker walls than those in lymphatic malformations. Similar to the lymphatic vessels, the malformed venous endothelium expresses VEGF-A and HIF-1α (P–Q, red arrows). The region outlined by a box in the original image is displayed in an enlarged view in a corresponding panel. Scale bars: 100 μm (A–C''''', D'–K, L'–S) and 1 mm (D, L). Source data are available online for this figure.

specification (Orsenigo et al, 2020; Luo et al, 2021), arteries, which already have sufficient PI3K/AKT signaling, may be less affected by these mutations (Hong et al, 2006). Consistent with this, no significant differences in pS6 expression were observed between control and mutant aortic, pulmonary arterial, and endocardial ECs in *Tie2-Cre;R26R-Pik3ca^H1047R* embryos (Fig. EV2).

By analyzing the differentiation of *Mesp1⁺* CPM cells (Nomaru et al, 2021), we explored how ECs in the head and neck region change their gene expression and at what stage their fate is determined. Our analysis showed that *Isl1* expression decreases at the mRNA level during early embryogenesis (E8.0–E8.25), followed by the expression of *Etv2*, *Kdr*, and eventually *Pecam*, marking the maturation into ECs. Additionally, our findings suggest that lymphatic vessels may arise directly from a common progenitor shared with veins, rather than through venous endothelium (Figs. EV4 and 5). Given that *Isl1* expression disappears at a very early stage and contributes to endothelial differentiation, experiments using *Isl1-Cre* or *Isl1-CreERT2* mice cannot clearly distinguish between LMs, VMs, and capillary malformations, In other words, *Isl1⁺* cells likely label a common progenitor population for multiple endothelial subtypes. Consequently, the diverse vascular malformations in the head and neck—including mixed venous-lymphatic and capillary malformations, as well as the macro- and microcystic subtypes of LMs—cannot be fully accounted for by this study alone. Additionally, the timeline of development significantly differs between humans and mice. In humans, lymphatic vessels in the trunk are formed within approximately one week, whereas it takes approximately four weeks for lymphatic vessels in the head and neck to establish a lumen structure (Yamaguchi et al, 2024). A deeper understanding of Pik3ca^H1047R's impact on cell differentiation requires further investigation.

Our scRNA-seq analysis revealed a notable increase in EC numbers and significant changes in hypoxia- and glycolysis-associated gene expression. Similar shifts were observed in other cell types, but they did not result in phenotypic changes, suggesting that ECs are particularly susceptible to these molecular alterations. Additionally, we performed endothelial subclustering to explore potential differences in gene expression among arterial, venous, capillary, and lymphatic endothelium. However, in the control embryos, the number of endothelial cells was too low to yield reliable data. Furthermore, future studies using both earlier and later embryonic stages will be essential to fully elucidate how Pik3ca^H1047R influences cellular differentiation and the progression of vascular malformations. In our data, the mesenchymal cell population was decreased, and within this cluster, pathways typically promoting epithelial mesenchymal tansition (EMT) (e.g., TGF-β, Wnt, and MYC target genes) were downregulated

(Fig. EV7B). Although PI3K activation is generally thought to enhance EMT, several studies in undifferentiated cells have reported that PI3K can suppress these signals via SMAD2/3 (Singh et al, 2012; Yu et al, 2015a). Elucidating how these changes in the mesenchyme contribute to vascular malformation pathogenesis remains an important avenue for future research.

We focused on Vegf-A, a key regulator of ECs proliferation and a downstream target of Hif-1α. Vegf-A likely drives both cell-autonomous and non-cell-autonomous effects on blood ECs, as well as LECs (Hong et al, 2004; Dellinger and Brekken, 2011). It is well known that overactivation of PI3K enhances glycolysis (Hu et al, 2016). In our study, the elevated expression of glycolytic enzymes, including *Ldha*, suggests a shift toward aerobic glycolysis, consistent with the Warburg effect. Previous studies have also shown that the stabilization of HIF-1α in cancerous tumors and pulmonary arterial smooth muscle cells is regulated by the PI3K/AKT signaling pathway (Zhong et al, 2000; Xiao et al, 2017). Additionally, the Warburg effect, driven by HIF-1α, promotes EC proliferation and migration (Fitzgerald et al, 2018; Yu et al, 2017). Our findings indicate that vascular malformations may share mechanistic similarities with cancer. Notably, mTOR inhibitors (Zou et al, 2020), commonly used in cancer therapies, have shown efficacy in reducing symptoms of vascular malformations. Likewise, HIF and VEGF inhibitors, employed in the treatment of malignant tumors, induce a significant reduction in malformed vasculature.

Our study reveals previously unrecognized mechanisms behind *PIK3CA*-driven vascular malformations and highlights the role of hypoxia-mediated signaling pathways in disease pathogenesis. Furthermore, we propose novel therapeutic strategies targeting these mechanisms. This research supports the development of new treatment approaches for intractable vascular malformations that afflict patients from childhood into adulthood.

## Methods

**Reagents and tools table**

| Reagent/resource | Reference or source | Identifier or catalog number |
| --- | --- | --- |
| **Experimental models** | | |
| Mice | | |
| Tie2-Cre | Kisanuki et al, 2001 | Not provided/N/A |
| CDH5-CreERT2 | Okabe et al, 2014 | Not provided/N/A |
| Isl1-Cre | Cai et al, 2003 | Not provided/N/A |

| Reagent/resource | Reference or source | Identifier or catalog number |
|---|---|---|
| Mef2c-AHF-Cre | Verzi et al, 2005 | Not provided/N/A |
| Isl1-CreERT2 | Laugwitz et al, 2005 | RRID:IMSR_JAX:029566 |
| Myf5-CreERT2 | Biressi et al, 2013 | RRID:IMSR_JAX:023342 |
| Pax3-CreERT | Southard et al, 2014 | RRID:IMSR_JAX:025663 |
| R26R-eYFP | Srinivas et al, 2001 | Not provided/N/A |
| R26R-tdTomato | Madisen et al, 2010 | RRID:IMSR_JAX:007914 |
| R26R-Pik3caH1047R | Adams et al, 2011 | RRID:IMSR_JAX:016977 |
| **Recombinant DNA** | | |
| No recombinant DNA was explicitly used | | |
| **Antibodies** | | |
| Anti-PECAM | Dianova, DIA-310 | RRID:AB_2631039 |
| Anti-PECAM | DAKO, M0823 | RRID:AB_2114471 |
| Anti-Prox1 | AngioBio, 11-002 | RRID:AB_10013720 |
| Anti-Prox1 | R&D Systems, AF2727 | RRID:AB_2170716 |
| Anti-VEGFR3 | R&D Systems, AF743 | RRID:AB_355563 |
| Anti-GFP | Abcam, ab290 | RRID:AB_303395 |
| Anti-phospho-S6 | Cell Signaling Technology (CST), #2215 | RRID:AB_331682 |
| Anti-Ki67 | Abcam, ab15580 | RRID:AB_443209 |
| Anti-PDPN (D2-40) | Nichirei Biosciences, 413151 (ready to use) | Not provided/N/A |
| Anti-HIF-1α | Abcam, ab114977 | RRID:AB_10900336 |
| Anti-VEGF-A | Abcam, ab52917 | RRID:AB_883427 |
| Anti-VEGF-A | Abcam, ab51745 | RRID:AB_2256948 |
| Anti-VEGFR2 | CST, #2479 | RRID:AB_2212507 |
| **Oligonucleotides and other sequence-based reagents** | | |
| Genotyping primers | This study (Table EV1) | Not provided/N/A |
| Custom NGS panel for PIK3CA exons | SureDesign (Agilent Technologies) | https://earray.chem.agilent.com/suredesign (accessed Dec 10, 2023) |
| Sanger sequencing primers | Hirose et al, 2024; Hori et al, 2022 | Not provided/N/A |
| **Chemicals, enzymes and other reagents** | | |
| QIAamp DNA FFPE Tissue Kit | Qiagen (Valencia, CA, USA) | ID. 56404 |
| Tamoxifen | Sigma Aldrich | T5648 |
| Rapamycin | Selleck Chemicals | S1039 |
| LW6 (HIF-1α inhibitor) | Selleck Chemicals | S8441 |
| 2G11-2A05 (VEGF-A inhibitor) | Biolegend | 512810 |

| Reagent/resource | Reference or source | Identifier or catalog number |
|---|---|---|
| Rat IgG2a isotype control | Biolegend | 400544 |
| Type II collagenase | Worthington | CLS-2 |
| DNase I | Roche | EN0521 |
| DMEM | Fujifilm | 041-30081 |
| FBS | Gibco | 10270106 |
| Antibiotic-antimycotic (AB/AM) solution | Gibco | 15240062 |
| Elastica van Gieson (EVG) / ESR staining dyes | Muto Pure Chemicals Co. (Tokyo) | 40321 |
| DMSO | Sigma Aldrich | C992J30 |
| **Software** | | |
| SureDesign | Agilent Technologies | https://earray.chem.agilent.com/suredesign |
| SureCall | Agilent Technologies | v4.0, https://www.agilent.com/en/download-software-surecall |
| ImageJ | NIH | https://imagej.nih.gov/ij/ |
| GraphPad Prism | GraphPad Software | Version 10 |
| R | CRAN | Version 4.3.3 |
| Seurat | CRAN (Seurat package) | Version 5.0.3 |
| Harmony | CRAN (harmony package) | Version 1.2.0 |
| Enrichr | Enrichr web tool | https://maayanlab.cloud/Enrichr/ |
| escape | CRAN (escape package) | Version 1.12.0 |
| iDEP | iDEP web tool (Ge et al, 2018) | v2.01 |
| GSEA software | Broad Institute | v4.3.3 |
| DESeq2 | Bioconductor | v1.38.3 |
| Trim Galore | Babraham Bioinformatics | v0.6.10 |
| STAR | Dobin et al, 2012 | v2.7.10a |
| featureCounts (Subread) | Bioconductor (Liao et al, 2014) | v2.0.1 |
| Cell Ranger | 10x Genomics | v6.1.2 / v7.1.0 |
| velocyto | Python (La Manno et al, 2018) | v0.17.17 |
| scVelo | Python (Bergen et al, 2020) | v0.3.3 |
| **Other** | | |
| Illumina MiSeq instrument | Illumina (San Diego, CA) | |
| FACS Aria III | BD Biosciences | |
| Keyence BZ-X700 microscope | Keyence | |
| BD FACSDiva | BD Biosciences | Not provided/N/A |

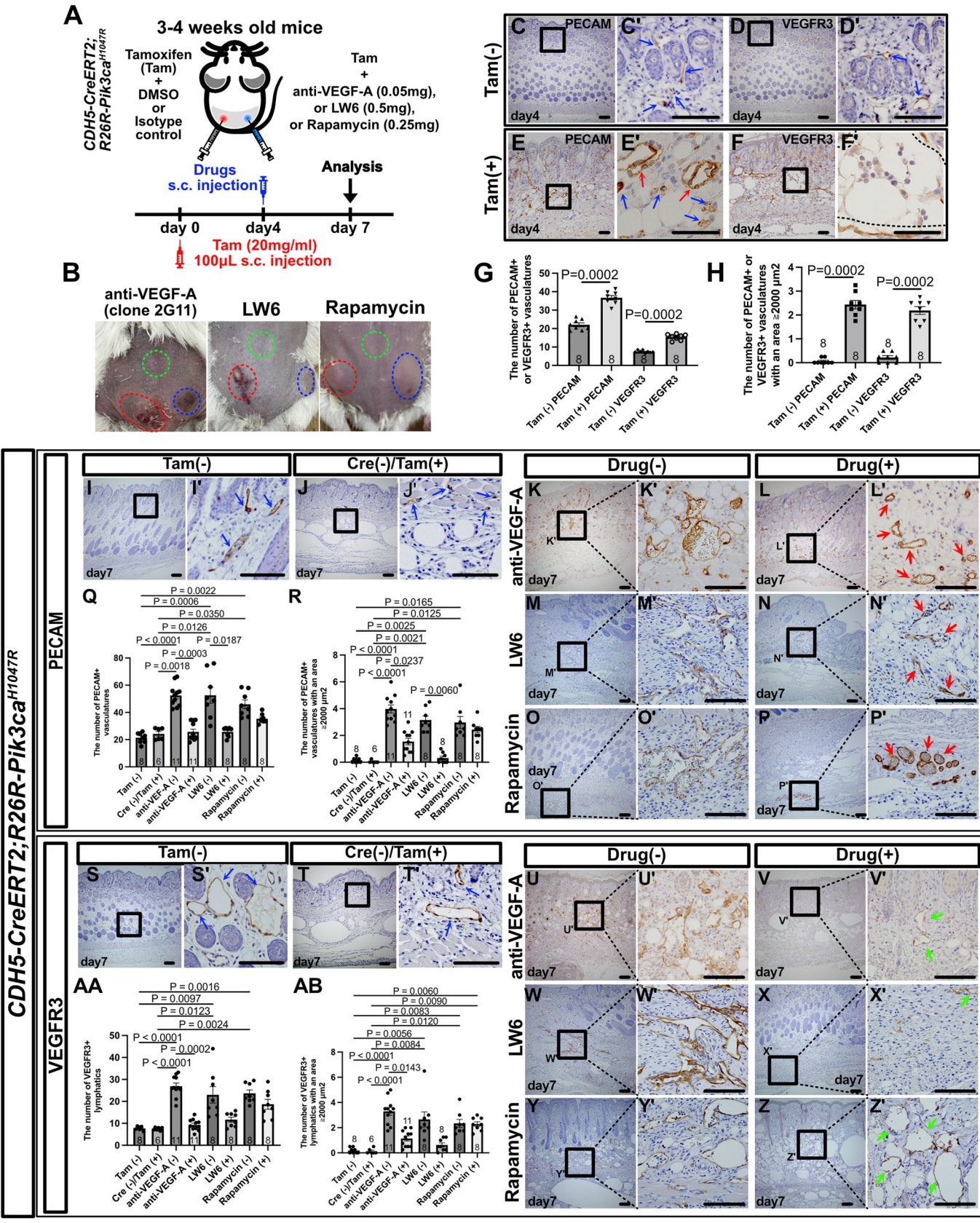

**Figure 6.   HIF-1α inhibitor and VEGF-A neutralizing antibody significantly suppress vascular malformations in the dorsal skin of mice.**

(A) Experimental design for inducing and treating progressive vascular malformations using anti-VEGF-A (clone 2G11), LW6, rapamycin, or respective controls (DMSO for LW6 and rapamycin; rat IgG2a isotype control for anti-VEGF-A). (B) Macroscopic effects of each treatment. The left panel (red dotted line) represents the control group (*CDH5-CreERT2; R26R-Pik3ca^{H1047R}* administered tamoxifen but no drug, whereas the right panel (blue dotted line) shows the treatment group (*CDH5-CreERT2; R26R-Pik3ca^{H1047R}* administered both tamoxifen and the indicated drug). In the same mouse, a region that did not receive tamoxifen served as the Tam(−) control (green dotted line). (C–F′) Histological sections from day 4 (before drug administration) of normal skin without tamoxifen (Tam(−)) and tamoxifen-treated (Tam(+)) skin in *CDH5-CreERT2; R26R-Pik3ca^{H1047R}* mice. Sections were cut vertically from the epidermis through the dermis, revealing PECAM⁺ blood vessels and VEGFR3⁺ lymphatic vessels (blue arrows). On day 4, enlarged PECAM⁺ vasculatures (red arrows) and VEGFR3⁺ lymphatic vessels (black dotted lines) were observed. (G, H) Statistical analysis of the same cohorts shown in (C–F′). (I, J) Histological sections of tamoxifen-untreated *CDH5-CreERT2; R26R-Pik3ca^{H1047R}* mutant skin at day 7 (Tam(−)) and *R26R-Pik3ca^{H1047R}* mice lacking *Cre* but receiving tamoxifen (Cre(−)/Tam(+)) at day 7, stained for PECAM. Neither group displayed increased PECAM⁺ vasculature (blue arrows). (K–P′) Histological sections of tamoxifen-treated *CDH5-CreERT2; R26R-Pik3ca^{H1047R}* mutant skin at day 7, stained for PECAM, comparing groups with (Drug(+)) or without (Drug(−)) the indicated drug. Anti-VEGF-A and LW6 each reduced PECAM⁺ vasculature (red arrows), whereas rapamycin exerted a milder effect (red arrows). (Q, R) Statistical analysis of the same cohorts. (S, T) Histological sections of tamoxifen-untreated *CDH5-CreERT2; R26R-Pik3ca^{H1047R}* mutant skin at day 7 (Tam(−)) and *R26R-Pik3ca^{H1047R}* mice lacking *Cre* but receiving tamoxifen (Cre(−)/Tam(+)) at day 7, stained for VEGFR3. Neither group showed increased VEGFR3⁺ lymphatic vessels (blue arrows). (U–Z′) Histological sections of tamoxifen-treated *CDH5-CreERT2; R26R-Pik3ca^{H1047R}* mutant skin at day 7, stained for VEGFR3, comparing Drug(+) and Drug(−) groups. Anti-VEGF-A and LW6 markedly reduced VEGFR3⁺ lymphatic vessels (green arrows), whereas rapamycin treatment left more large lymphatic vessels intact (green arrows). (AA, AB) Statistical analysis of the same cohorts. Each dot represents data from an individual sample. The white, vacuole-like structures in the images are oil droplets from tamoxifen injection. The region outlined by a box in the original image is displayed in an enlarged view in a corresponding panel. Scale bars: 100 μm (C–F′, I–J′, K–P′, S–T′, U–Z′). Statistical analyses were performed using the Kruskal–Wallis test followed by Dunn's multiple comparison test (Q, R, AA, AB) and the nonparametric Mann–Whitney *U* test. ns ≥0.05 (G, H). Error bars represent the mean ± standard error of the mean (SEM). Source data are available online for this figure.

## Human samples

Formalin-fixed paraffin-embedded (FFPE) tissues of residual specimens used for diagnostic purposes were employed. These surgical specimens were classified according to the classification system proposed by the International Society for the Study of Vascular Anomalies (Kunimoto et al, 2022). This study was approved by the Ethical Review Board of the Graduate School of Medicine, Mie University (approved number: H2024-078) and the Graduate School of Medicine, Osaka University (approved number: 17214 and K24132). Informed consent was supplemented by an opt-out provision, ensuring the participants' autonomy to withdraw from the study at any point. The human study was conducted in compliance with Japanese regulations, following the Ethical Guidelines for Medical and Health Research Involving Human Subjects, and conformed to the principles set out in the WMA Declaration of Helsinki and the Department of Health and Human Services Belmont Report. Mutation analysis was conducted by YH and HK following the protocols (Hirose et al, 2024). Next-generation sequencing (NGS) was performed using a custom panel, as previously described (Hori et al, 2022). Genomic DNA was extracted from FFPE tissues using the QIAamp DNA FFPE Tissue Kit (Qiagen, Valencia, CA, USA). The gene panel was designed using SureDesign (https://earray.chem.agilent.com/suredesign, accessed Dec 10, 2023) to cover all the exons of the *PIK3CA* genes. Sequence libraries were prepared using the custom SureSelect Low-Input Target Enrichment System (Agilent Technologies Inc., Santa Clara, CA, USA) and sequenced using an Illumina MiSeq instrument (Illumina, San Diego, CA, USA). Variant calling was performed using SureCall version 4.0 (https://www.agilent.com/en/download-software-surecall, accessed Dec 10, 2023). Intron DNA, non-coding DNA, and variants with an allele frequency of less than 1% were excluded. Variants obtained by panel sequencing were confirmed by Sanger sequencing (Hirose et al, 2024; Hori et al, 2022). The patient demographics—including age, sex, ethnicity, variant allele frequency, and lesion location—are provided in Table EV2.

## Immunohistochemistry and special staining

For histological analyses, samples were collected and fixed in 2% paraformaldehyde overnight at 4 °C, followed by storage in 70% ethanol at 4 °C. HE staining and IHC were performed using 3 μm-

thick paraffin-embedded sections. Sections for IHC were deparaffinized and rehydrated through a series of xylene and ethanol. For the enzyme-antibody method, endogenous peroxidase activity was blocked using 0.3% hydrogen peroxide ($H_2O_2$) in methanol for 20 min. For fluorescent antibody staining, to suppress autofluorescence, samples were incubated in 0.1% sodium borohydride in 0.1 M phosphate-buffered saline (PBS) for 30 min, then rinsed with water, and subsequently incubated for 5 min in 0.2 M glycine in 0.1 M PBS. Antigen retrieval was carried out using a pressure chamber with Tris-EDTA buffer (7.4 mM Tris, 1 mM EDTA-2Na, pH 9.0). The sections were immunostained using primary antibodies against PECAM (DIA-310, Dianova, 1:100, RRID:AB_2631039; M0823, DAKO, 1:100, RRID:AB_2114471), Prox1 (11-002, AngioBio, 1:100, RRID:AB_10013720; AF2727, R&D Systems, 1:100, RRID:AB_2170716), VEGFR3 (AF743, R&D Systems, 1:100, RRID:AB_355563), GFP (ab290, Abcam, 1:250, RRID:AB_303395), phospho-S6 (#2215, CST, 1:100, RRID:AB_331682), Ki67 (ab15580, Abcam, 1:250, RRID:AB_443209), D2-40 (anti-PDPN) (413151, Nichirei Biosciences, ready to use), HIF-1α (ab114977, Abcam, 1:100, RRID:AB_10900336), VEGF-A (ab52917, Abcam, 1:100, RRID:AB_883427; ab51745, Abcam, 1:100, RRID:AB_2256948), and VEGFR2 (#2479, CST, 1:100, RRID:AB_2212507). For the enzyme-antibody method, the secondary antibody from the Histofine Simple Stain System (Nichirei Biosciences) was incubated with the slides for 1 h. Peroxidase activity was visualized using DAB-$H_2O_2$. For fluorescent immunostaining, Alexa Fluor-conjugated secondary antibodies (Abcam, 1:400) were subsequently applied. Imaging was carried out using a Keyence BZ-X700 microscope. All images were processed using ImageJ software.

For Elastica van Gieson (EVG) staining, tissue sections were deparaffinized in xylene, rehydrated through graded ethanol, and rinsed in Milli-Q water. Sections were then treated with 1% hydrochloric acid in 70% ethanol, stained with Weigert's resorcin-fuchsin for 40–50 min to stain elastic fibers dark purple-black, and counterstained with Weigert's iron hematoxylin for 3–5 min. After additional rinsing, sections were stained in Van Gieson's solution for 15 min, which stains collagen fibers red, muscle tissue yellow, and elastic fibers black. For Elastica Picrosirius Red (ESR) staining,

tissue sections were deparaffinized, rehydrated through graded ethanol, and rinsed in Milli-Q water. Sections were treated with 1% hydrochloric acid in 70% ethanol, stained with Weigert's resorcin-fuchsin for 40–50 min to visualize elastic fibers (stained dark purple-black), and counterstained with Weigert's iron hematoxylin for 3–5 min. After additional rinsing, sections were stained in Picrosirius Red solution for 15 min, which stains collagen fibers red. In this protocol, muscle tissue is also stained yellow, and elastic fibers appear black. Reagents for special staining were purchased from Muto Pure Chemicals Co., Tokyo.

## Quantification of section immunostaining

Veins and arteries were classified based on anatomical criteria. Vessels demonstrating continuity with a clearly identifiable vein (e.g., the common cardinal vein) in serial sections were defined as veins. In contrast, the aorta and pulmonary artery, each exhibiting a distinct wall structure indicative of a direct connection to the heart, were designated as arteries. Lymphatic vessels were identified based on the combined expression of Prox1, VEGFR3, and PECAM, along with the developmental stage, morphology, and anatomical location as described in our previous studies (Maruyama et al, 2019, 2022, 2021). PECAM$^+$ vessels that lacked a definitive wall structure, did not express lymphatic markers, or did not exhibit clearly identifiable continuity necessary for classification as veins or capillaries were collectively designated as blood vessels or vasculatures. For quantifying immunostained sections, an average of at least two 3 µm-thick sections from 20× power fields (0.55 mm$^2$/field) for each anatomical region were analyzed. For sagittal sections, one HE-stained section and nine unstained sections were prepared per embryo, typically with a total of 100 sections made. The sections were centered around the position containing the ascending aorta and heart. In the case of transverse sections, the samples included all sections from the head down to where the heart was no longer visible.

## Mouse lines and treatments

The following mouse strains were used: *Tie2-Cre* (Kisanuki et al, 2001), *CDH5-CreERT2* (Okabe et al, 2014), *Isl1-Cre* (Cai et al, 2003), *Mef2c-AHF-Cre* (Verzi et al, 2005), *Isl1-CreERT2* (RRID:IMSR_JAX:029566) (Laugwitz et al, 2005), *Myf5-CreERT2* (RRID:IMSR_JAX:023342) (Biressi et al, 2013), *Pax3-CreERT* (RRID:IMSR_JAX:025663) (Southard et al, 2014), *R26R-eYFP* (Srinivas et al, 2001), *R26R-tdTomato* (RRID:IMSR_JAX:007914) (Madisen et al, 2010), and *R26R-Pik3ca$^{H1047R}$* (RRID:IMSR_JAX: 016977) (Adams et al, 2011). Mice with the indicated RRID were purchased from The Jackson Laboratory, USA. All mice were maintained on a mixed genetic background (C57BL/6J× Crl:CD1(ICR)), both female and male mice were used for analyses, and no differences in phenotype were observed between them. The genotypes of the mice were determined by polymerase chain reaction using tail-tip or amnion DNA and the primers listed in Table EV1. The mice were housed in an environmentally controlled room at 23 ± 2 °C, with a relative humidity level of 50%–60%, under a 12-h light:12-h dark cycle. Embryonic stages were determined by timed mating, with the day of the appearance of a vaginal plug designated as embryonic day (E)0.5. All animal experiments were approved by the Mie University Animal Care and Use Committee

and performed in accordance with institutional guidelines (approved number:728).

For embryo experiments, tamoxifen (20 mg/mL; Sigma Aldrich, T5648) was dissolved in corn oil, and pregnant mice were intraperitoneally administered (125 mg/kg body weight) at the indicated time points. In experiments using *CDH5-CreERT2*, tamoxifen was diluted to one-quarter strength (25 mg/kg body weight) in corn oil and intraperitoneally injected into pregnant mice.

## Mouse model for skin vascular malformations

For postnatal induction, 3 to 4-week-old *CDH5-CreERT2; PIk3ca$^{H1047R}$* mice and controls were used. The dorsal region was shaved using clippers and depilatory cream (Reckitt Benckiser). Tamoxifen (20 mg/mL) was intradermally administered (100 µL) into the skin above both thighs, ensuring a localized circular swelling to confine the injection area. Four days later, the right side received one of the following treatments applied directly to the tamoxifen-treated area to target malformed vasculature: LW6 (Hif-1α inhibitor, S8441, Selleck), dissolved in DMSO (5 mg/mL), with 100 µL administered; 2G11-2A05 (a VEGF-A inhibitor, 512810, Biolegend) was prepared in PBS at a concentration of 1 mg/mL and administered as a single 50 µL injection. Rapamycin (an mTOR inhibitor, S1039, Selleck) was dissolved in DMSO at 5 mg/mL and administered at 50 µL. As a control, the left side was treated with the corresponding solvent (DMSO) or, for 2G11-2A05, with a Rat IgG2a isotype control (400544, Biolegend) prepared at 1 mg/mL and administered as a 50 µL injection on the contralateral side. The dosage of the drugs was determined based on previously published literature (Lee et al, 2021; Martinez-Corral et al, 2020; Basu et al, 2008; Reinders et al, 2003; Surve et al, 2024; Churchill et al, 2022) and information provided on the Selleck Chemicals and Biolegend website. Three days after treatment, the skin was excised down to the muscle using scissors, processed for paraffin embedding, and subjected to immunohistochemistry analysis. Sections were stained for PECAM and VEGFR3. Additionally, untreated skin from the upper dorsal region was harvested and used as a normal control.

## Embryo preparation for single-cell RNA sequencing and FACS sorting

E13.5 embryos were collected by cesarean section from *Isl1-Cre;R26R-eYFP* or *Isl1-Cre;R26R-eYFP;Pik3ca$^{H1047R}$* mice. The embryos were washed twice with 0.1 M PBS. Using a stereomicroscope, three eYFP$^+$ embryos were selected per group. The upper body regions, including tissues above the liver, were dissected and finely minced using scissors in 10 ml of DMEM (Fujifilm, 041-30081) containing 10% FBS (Gibco, 10270106), 1% antibiotic-antimycotic (AB/AM) solution (Gibco, 15240062), 0.5 mg/ml DNase I (Roche, EN0521), and 1 mg/ml type II collagenase (Worthington, CLS-2). The tissue suspension was incubated at 37 °C in a water bath for 30 min, then filtered through a 70-µm cell strainer (Falcon, 352350), followed by two passages through a 40-µm cell strainer (Corning, 431750). After staining, cells were washed with FACS buffer (PBS, 0.5% FBS, 2 mM EDTA), and the cell pellet was resuspended in 1 ml of 2 mM EDTA/PBS/0.1% BSA for FACS sorting. Cell sorting was performed using FACS Aria III

(BD Biosciences). Data were analyzed using BD FACSDiva software.

## Analysis of the single-cell RNA-seq results

Libraries were prepared using the Next GEM Single Cell 5′ Library and Gel Bead Kit v2 (10x Genomics), following the manufacturer's protocol. Raw reads were processed using 10x Genomics Cell Ranger software (v7.1.0) for demultiplexing, genome alignment (GRCm38/mm10), and feature-barcode matrix generation. Data analysis was performed using R (version 4.3.3) and the Seurat package (version 5.0.3). Quality control was conducted based on unique molecular identifiers, gene counts, and mitochondrial gene expression. Cell cycle gene expression was regressed during the integration step to avoid clustering bias. Data from multiple samples were integrated using both Anchor-based CCA (Seurat version 5.0.3) and Harmony integration (harmony package version 1.2.0), and no significant differences were found between these methods. Therefore, Anchor-based CCA was used for downstream analyses. Principal component analysis (PCA) was performed for dimensionality reduction, followed by UMAP for cluster visualization. To address cell cycle heterogeneity, cell cycle phase scores were calculated using known marker genes (Kowalczyk et al, 2015), and the cell cycle effects were regressed using the *CellCycleScoring* and *ScaleData* functions.

To annotate each cluster, conserved marker genes between conditions were detected using the *FindConservedMarkers* function from the Seurat package (FDR < 0.05, log2-fold change >1).

For differentially expressed gene analysis, the *FindMarkers* function from the Seurat package was used (FDR < 0.05, fold change >1.5). For enrichment analysis, we utilized Hallmark gene sets from the MSigDB Collections, applying two methods: over-representation analysis (Enrichr web tool[69]) and single-sample Gene Set Enrichment Analysis (ssGSEA) (escape version 1.12.0). Significant Hallmark gene sets were defined as those with an FDR < 0.1.

## Re-analysis of bulk RNA sequencing

We also include publicly available bulk RNA-seq data (GEO: GSE196311, GSE130807) (Jauhiainen et al, 2023). Raw FASTQ files were processed using Trim Galore (v0.6.10) and STAR (v2.7.10a). Read counts were generated using featureCounts (Subread v2.0.1). The R package DESeq2 (v1.38.3), and iDEP web tool (v2.01) (Ge et al, 2018) were used to normalize the gene expression matrix. Gene Set Enrichment Analysis (GSEA) was performed using GSEA software (v4.3.3), focusing on Hallmark gene sets from the MSigDB collection. Significant Hallmark gene sets were defined as those with an FDR < 0.1.

## Re-analysis of single-cell RNA sequencing data from *Mesp1*+ CPM cells

We re-analyzed scRNA-seq data from *Mesp1*+ CPM cells between E8.0 and E10.5 (GEO: GSE167493) (Nomaru et al, 2021). Raw reads were processed using 10x Genomics Cell Ranger software (v6.1.2), mapped to GRCm38 (mm10), and feature-barcode matrices were generated on the SHIROKANE SC. Data analysis was conducted using R (v4.2.0) and the Seurat package (v4.3.0). Quality control

### The paper explained

**Problem**

Vascular malformations—abnormal development of blood or lymphatic vessels—often occur in the head and neck region, where they can interfere with essential functions such as breathing and swallowing. These lesions, frequently linked to mutations in the PIK3CA gene, remain challenging to treat, highlighting the urgent need for more effective therapies.

**Results**

By introducing a common PIK3CA mutation (Pik3ca[H1047R]) into specific embryonic cells in mice, we generated a model closely resembling human head and neck vascular malformations. Single-cell RNA sequencing revealed that the mutation activates hypoxia-related signaling and increases the production of VEGF-A, a growth factor that drives vessel expansion. In patient samples, malformed vessels also showed elevated HIF-1α and VEGF-A. Importantly, blocking HIF-1α or VEGF-A in the mouse model substantially reduced the vascular malformations.

**Impact**

These findings highlight embryonic origins and hypoxia-driven mechanisms as key contributors to vascular malformations. By pinpointing HIF-1α and VEGF-A as central drivers, the study suggests new therapeutic strategies that target these pathways. Such approaches may offer more effective treatments for patients with currently intractable head and neck vascular malformations.

yielded the following cell counts: E8.0 (8,276 cells), E8.25 (3255 cells), E9.5 (4288 cells), and E10.5 (9664 cells). Cells with 1000–7500 feature counts and <5% mitochondrial genes were retained, leaving 22,899 cells for analysis. Normalization was performed using *sctransform*, selecting the top 3000 variable genes. Cell cycle effects were regressed using phase scores based on markers from Kowalczyk et al (2015). Two approaches were used: one across all stages (E8.0–E10.5) and one focusing on early stages (E8.0–E8.25). Batch correction and integration were conducted using Anchor-based CCA, followed by dimensionality reduction employing PCA and clustering with k-nearest neighbor (k-NN) graphs (E8.0–E10.5: 20 clusters, E8.0–E8.25: 15 clusters). Clustering was visualized using UMAP. Clusters were annotated based on marker genes identified using Seurat's *FindAllMarkers* function (FDR < 0.05, log2-fold change >1). Subclustering of EC clusters was performed in both the E8.0–E10.5 (C7 and 15) and E8.0–E8.25 (C3, 4, 11, and 12) approaches, following the outline reported by Piper et al (Piper et al, 2022). ECs were extracted using Seurat's *subset* function, and the downstream analysis (PCA, clustering, and visualization) was performed. The E8.0–E10.5 approach identified eight clusters (C0–7), while the E8.0–E8.25 approach identified seven clusters (C0–6). The following parameters were used: for E8.0–E10.5, PCs 1 to 20, k = 20, resolution = 0.6; for E8.0–E8.25, PCs 1 to 5, k = 20, resolution = 0.5. Clusters were visualized using UMAP and annotated based on marker genes identified using Seurat's *FindAllMarkers* (FDR < 0.05, log2-fold change >1).

RNA velocity analysis was performed following the velocyto and scVelo tutorials (Manno et al, 2018; Bergen et al, 2020). We used velocyto (v0.17.17) to separate spliced and unspliced mRNA reads from the scRNA-seq mapping data, generating spliced and

unspliced count matrices on the SHIROKANE SC. Subsequent RNA velocity estimation, pseudotime calculation, and trajectory inference were performed using scVelo (v0.3.3) in Python (v3.11.7), based on the spliced/unspliced matrices and Seurat analysis results. The following parameters were used for both the E8.0–E10.5 and E8.0–E8.25 approaches: PCs 1 to 30, k = 30 NN graph. Pseudotime was calculated from the directed velocity graph, and the inferred differentiation trajectories were represented as a PAGA graph, extended with velocity-derived directionality.

## Statistics and reproducibility

All statistical analyses and data visualization were performed using GraphPad Prism 10. Data are presented as the mean ± standard error of the mean (SEM). Comparisons between two groups were made using the Mann–Whitney *U* test. For three groups, the Kruskal–Wallis test followed by Dunn's multiple comparison test was conducted. *P* values below 0.05 were considered significant. The experiments were not randomized, and no blinding procedures were implemented during analysis or quantification. No statistical methods were used to predetermine sample sizes.

## Data availability

The single-cell RNA-seq data generated in this study have been deposited in the NCBI Gene Expression Omnibus (GEO) under accession number GSE279129. Because we obtained consent under the condition that the patient-derived data would not be reused (i.e., no secondary usage), these data cannot be made publicly available.

The source data of this paper are collected in the following database record: biostudies:S-SCDT-10_1038-S44321-025-00235-1.

## Peer review information

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

## Acknowledgements

We extend our sincere gratitude to all laboratory members for their insightful discussions and encouragement throughout this study. This work was supported in part by Grants-in-Aid for Scientific Research from the Ministry of Education, Culture, Sports, Science, and Technology of Japan (20K17072 and 23K15949 to KM); the Japan Foundation for Applied Enzymology (VBIC to KM); the SENSHIN Medical Research Foundation (KM); the Takeda Science Foundation (KM); The Ichiro Kanehara Foundation for the Promotion of Medical Sciences and Medical Care (KM); the Mie University Life Relay Foundation (KM); the HITACHI Group Foundation (Kurata Research Grant to KM); the Japan Intractable Diseases Research Foundation (KM); the Mochida Memorial Foundation for Medical and Pharmaceutical Research (KM); the Japan Agency for Medical Research and Development (AMED) under Grant Number 22jm0610079h0001 (KM); the Mie University Research Promotion and Graduate School Reform-Related Research Grant Project (KM); and the TERUMO Life Science Foundation (KM); The Sumitomo Foundation (KM). We also acknowledge the NGS Core Facility at the Research Institute for Microbial Diseases of Osaka University for their support with sequencing.

## Author contributions

**Sota Torii**: Validation; Investigation; Visualization. **Keiki Nagaharu**: Data curation; Investigation. **Nanako Nakanishi**: Investigation. **Hidehito Usui**: Data curation; Investigation. **Yumiko Hori**: Resources; Investigation. **Katsutoshi Hirose**: Resources; Data curation. **Satoru Toyosawa**: Resources. **Eiichi Morii**: Resources. **Mitsunaga Narushima**: Resources; Supervision. **Yoshiaki Kubota**: Resources; Supervision. **Osamu Nakagawa**: Resources; Supervision. **Kyoko Imanaka-Yoshida**: Supervision. **Kazuaki Maruyama**: Conceptualization; Funding acquisition; Visualization; Writing—original draft; Writing—review and editing.

Source data underlying figure panels in this paper may have individual authorship assigned. Where available, figure panel/source data authorship is listed in the following database record: biostudies:S-SCDT-10_1038-S44321-025-00235-1.

## Disclosure and competing interests statement

The authors declare no competing interests.

# Expanded View Figures

**Figure EV1.   Pik3ca^{H1047R} expression in *Tie2-Cre* embryos does not significantly affect the endothelium of the aorta, pulmonary artery, or endocardium.**

(**A**) Gross morphology of the control and *Tie2-Cre; R26R-Pik3ca^{H1047R}* mutant embryos at E12.5. Comparisons were made between control and mutant embryos ($n = 5, 5$ for each group). In the control, eYFP is not expressed, but in the mutant, eYFP is expressed throughout the body. (**B–K"**) Immunostaining of sagittal sections with the indicated antibodies. An increase in Ki67$^+$ cells in the cardinal vein ECs is observed (**B""**, **C""**, red arrows). In mutant mice, eYFP$^+$ cells are observed in both cardinal vein ECs and LECs (**B"""**, **C"""**, red arrows). (**D–K"**) Similar analysis in the pulmonary artery, descending aorta, and endocardium. The region outlined by a box in the original image is displayed in an enlarged view in a corresponding panel. (**L–Q**) Statistical analysis of the indicated parameters. CV, cardinal vein; Ao, aorta; PA, pulmonary artery. Each dot represents a value obtained from one sample. Scale bars, 100 μm (**B'–B"""**, **C'–E"""**, **D–K"**) and 1 mm (**A**). The nonparametric Mann–Whitney *U* test was used for statistical analysis, with exact p-values indicated. ns ≥0.05. Error bars represent the mean ± standard error of the mean (SEM).

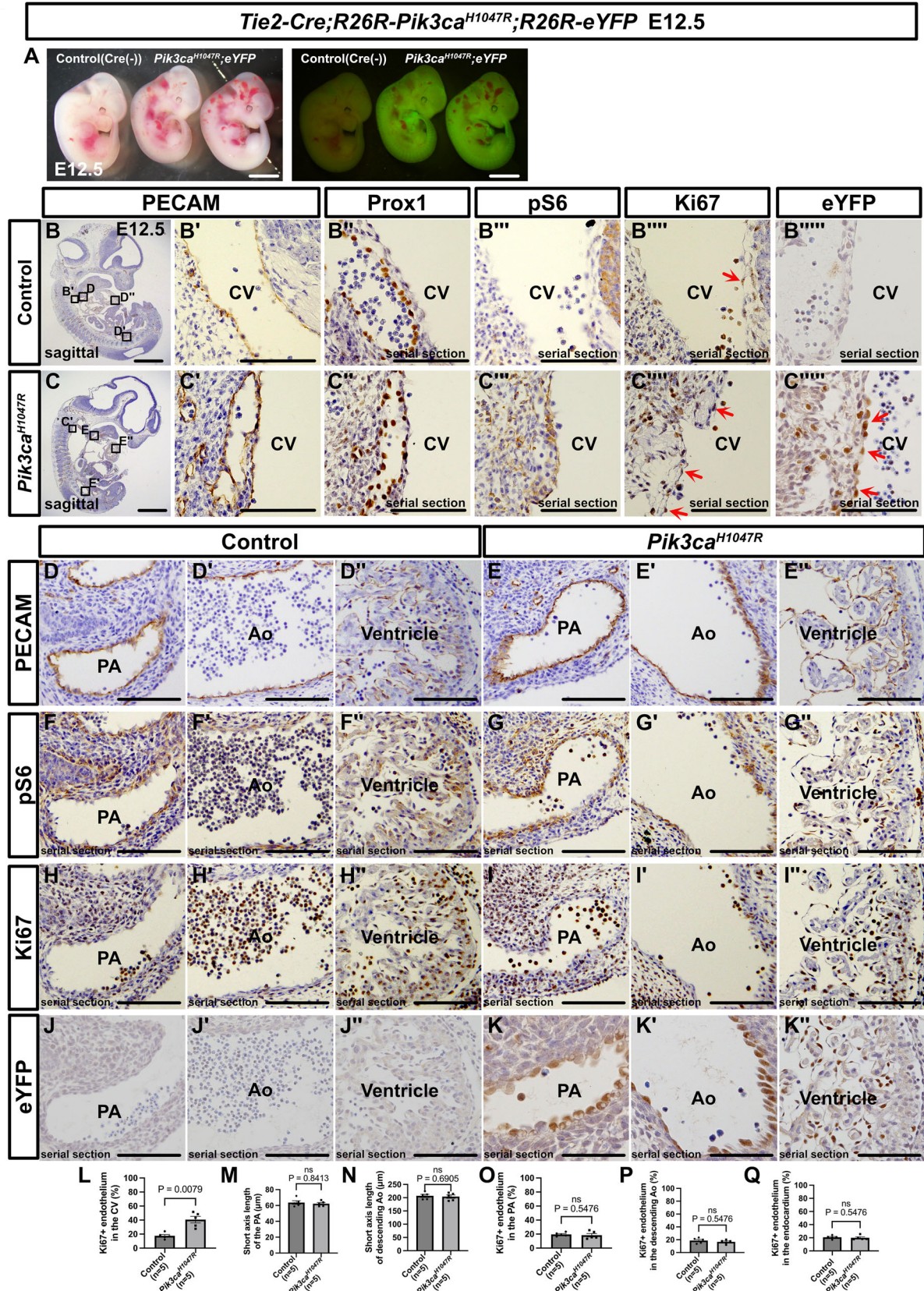

Tie2-Cre;R26R-Pik3ca^{H1047R};R26R-eYFP E12.5

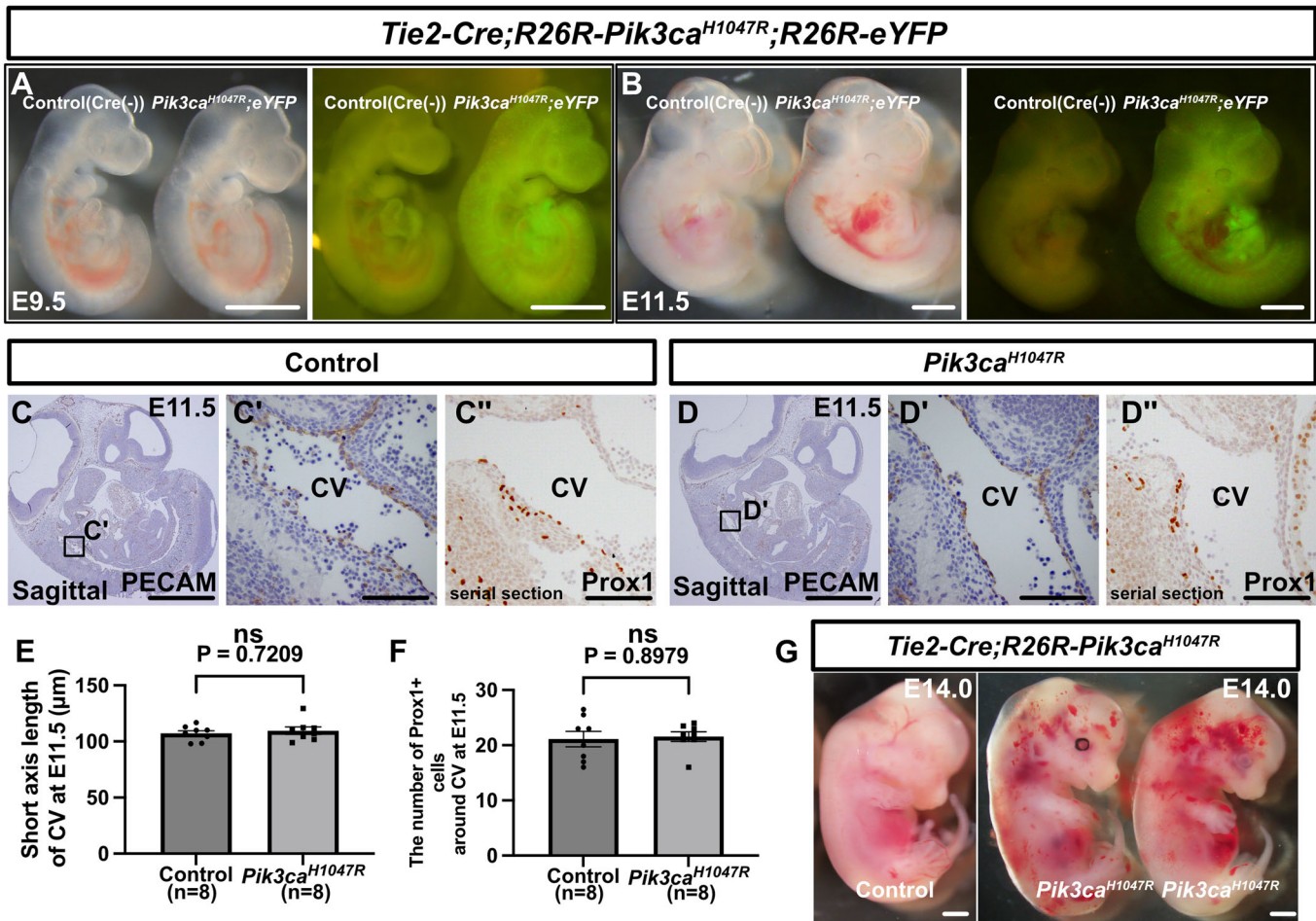

**Figure EV2. Pik3ca^H1047R^ expression in *Tie2-Cre* embryos does not produce a discernible phenotype at E11.5.**

(A, B) Gross morphology of control and *Tie2-Cre; R26R-Pik3ca^H1047R^* mutant embryos at E9.5 (*n* = 4 each for control and mutant) and at E11.5 (*n* = 8 each for control and mutant). (C–D″) Immunostaining of sagittal sections with the indicated antibodies at E11.5. (E, F) Neither overall phenotype, short-axis length of the cardinal vein (CV), nor the number of Prox1+ cells around the CV differed between control and mutant embryos. (G) Gross morphology of control and *Tie2-Cre; R26R-Pik3ca^H1047R^* mutant embryos at E14.0. The region outlined by a box in the original image is displayed in an enlarged view in a corresponding panel. Each dot represents a value from a single sample. CV cardinal vein. Scale bars, 100 μm (C′–C″, D′–D″) and 1 mm (A, B, C, D). Statistical analysis was performed using the nonparametric Mann–Whitney *U* test, with exact *P* values indicated. ns ≥ 0.05. Error bars represent the mean ± standard error of the mean (SEM).

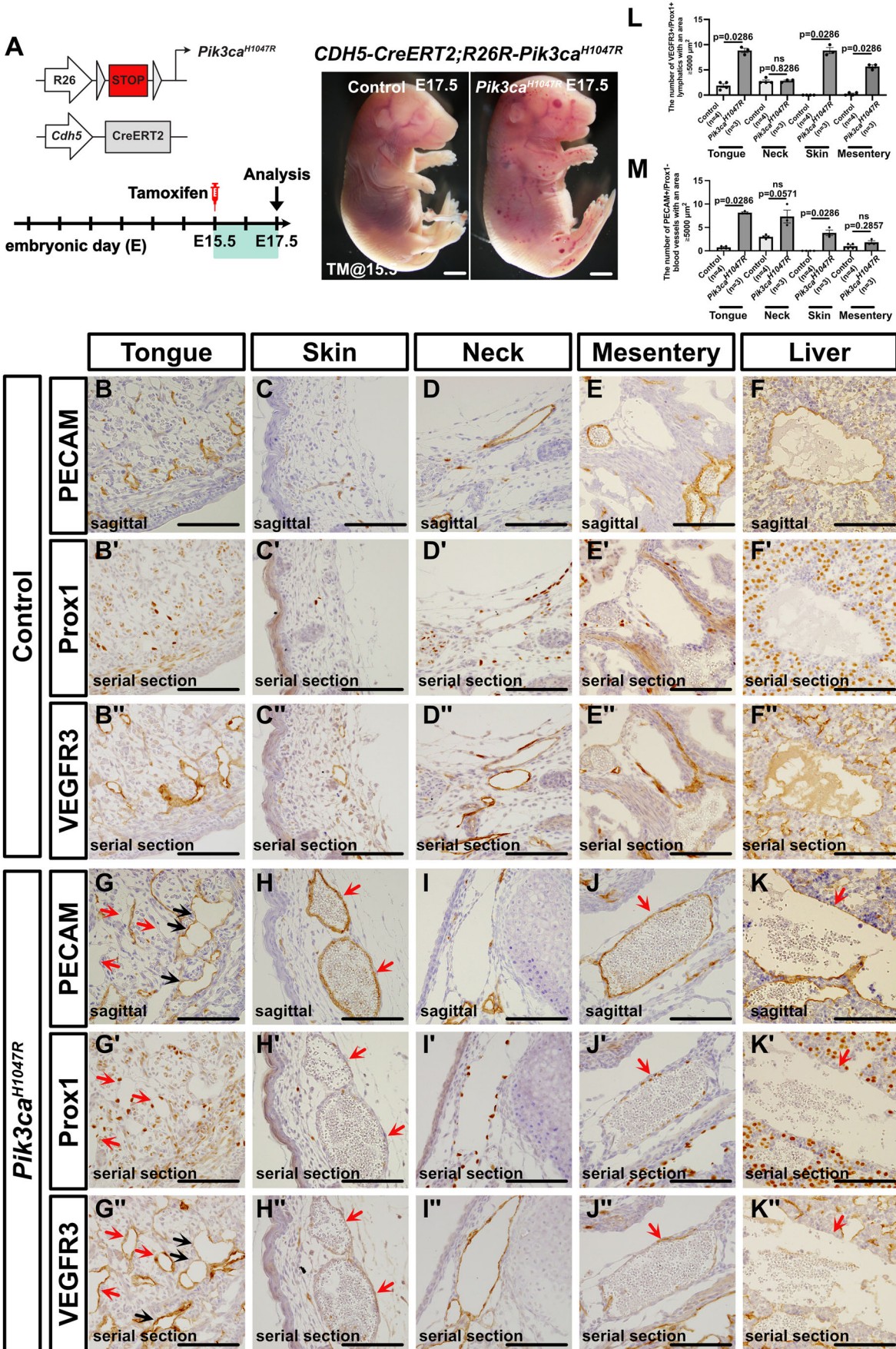

◀ **Figure EV3. Pik3ca^H1047R expression in endothelial cells at E15.5 leads to systemic vascular malformations.**

(A) Schematic of the experimental schedule and gross morphology of control and *CDH5-CreERT2; R26R-Pik3ca^H1047R* mutant embryos at E17.5 following tamoxifen administration to pregnant mice at E15.5. Comparisons were made between control and mutant embryos ($n = 4$, 3 for each group). (B–K") Sagittal section immunohistochemistry using the indicated antibodies. (B–B", G–G") In mutant embryos, mildly dilated PECAM^-/Prox1^+/VEGFR3^+ lymphatic vessels (red arrows) and an increased number of PECAM^+/Prox1^-/VEGFR3^+ blood vessels (black arrows) are observed in the tongue. (C–C", H–H") Aberrant, dilated PECAM^+/partially Prox1^+/partially VEGFR3^+ vasculatures are seen in the skin (red arrows). (D–D", I–I") No significant differences in the neck (larynx) are observed between the control and mutant embryos. (E–E", J–J") In the mesentery, aberrant, dilated PECAM^+/partially Prox1^+/partially VEGFR3^+ vasculatures similar to those in the skin are seen (red arrows). (F–F", K–K") In the liver, dilated PECAM^+/Prox1^-/partially VEGFR3^+ blood vessels are observed. The number of PECAM+ vessels with an area ≥20,000 μm$^2$ in the liver is $0 \pm 0$ (median ± SEM) ($n = 4$) in controls and $1.5 \pm 0.33$ (median ± SEM) ($n = 3$) in mutants ($P = 0.0286$). (L, M) Statistical analysis in the tongue, neck, skin, and mesentery. Each dot represents a value obtained from one sample. Scale bars: 100 μm (B–K") and 2 mm (A). The nonparametric Mann–Whitney $U$ test was used for statistical analysis, with exact $P$ values indicated. ns ≥ 0.05. Error bars represent the mean ± standard error of the mean (SEM).

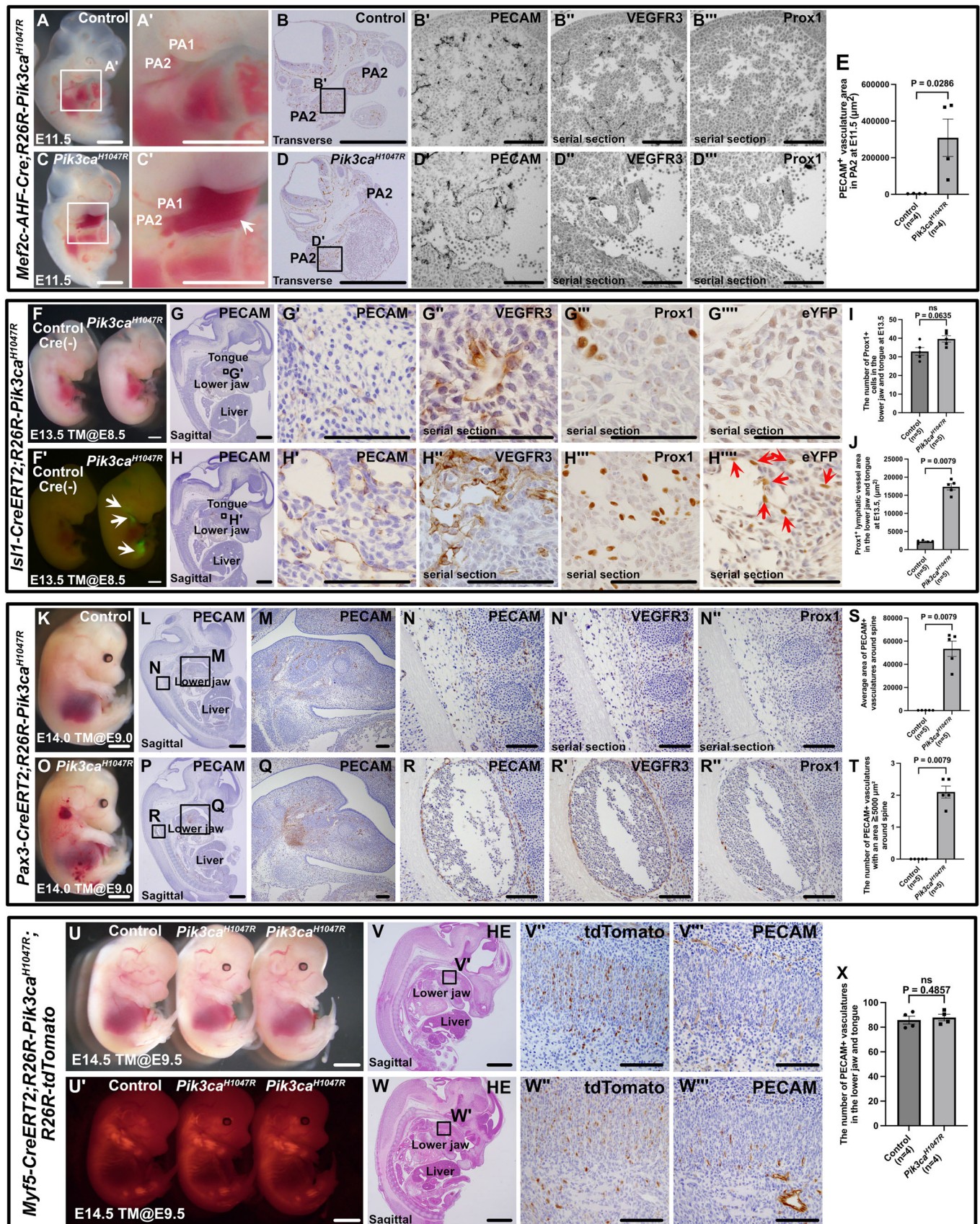

◀ **Figure EV4. Pik3ca^H1047R expression in CPM induces vascular malformations in the head and neck.**

(A, A', C, C') Gross morphology of the control and *Mef2c-AHF-Cre; R26R-Pik3ca^H1047R* mutant embryos at E11.5. In mutant embryos, enlarged blood-filled vasculatures were evident in PA1 and PA2 (white arrow). Comparisons were made between control and mutant embryos ($n = 4$, 4 for each group). (B–B''', D–D''') Transverse section immunostaining with the indicated antibodies. (E) Mutant embryos show dilated PECAM^+/Prox1^-/partially VEGFR3^+ blood vessels extending from the first to the second pharyngeal arches. (F, F') Gross morphology of the control and *Isl1-CreERT2; R26R-Pik3ca^H1047R; R26R-eYFP* mutant embryos at E13.5, after tamoxifen administration at E8.5. Comparisons were made between control and mutant embryos ($n = 5$, 5 for each group). eYFP expression is observed from the lower jaw to the neck and outflow tracts, and in the genital region (white arrows). (G–G'''', H–H'''', I, J) Mutant embryos show dilated PECAM^+/Prox1^+/VEGFR3^+ lymphatic vessels between the lower jaw and tongue. The number of Prox1^+ cells in these lymphatic vessels did not differ between the control and mutant groups. These vessels are also eYFP^+ (red arrows). (K, O) Gross morphology of the control and *Pax3-CreERT2; R26R-Pik3ca^H1047R* mutant embryos at E14.0, after tamoxifen administration at E9.0. Comparisons were made between control and mutant embryos ($n = 5$, 5 for each group). (L–N'', P–R'', S, T) No vascular malformations are seen in the head and neck of mutant embryos, but dilated PECAM^+/partially Prox1^+/VEGFR3^+ blood-filled vessels are observed around the spine. (U, U') Gross morphology of the control and *Myf5-CreERT2; R26R-Pik3ca^H1047R; R26R-tdTomato* mutant embryos at E14.5, after tamoxifen administration at E9.5. Comparisons were made between control and mutant embryos ($n = 4$, 4 for each group). (V, W) Sagittal section H&E staining. (V'', V''', W'', W''') Sagittal section immunostaining with the indicated antibodies. (X) No differences were observed in the number of PECAM^+ vasculatures in the lower jaw and tongue. PA pharyngeal arch. The region outlined by a box in the original image is displayed in an enlarged view in a corresponding panel. Scale bars: 100 μm (B'–B''', D'–D''', G'–G'''', H'–H'''', M–N'', Q–R'', V'', V''', W'', W'''), 1 mm (A, B, C, D, G, H, K, L, O, P, V, W), and 2 mm (F, F', U, U'). The nonparametric Mann–Whitney *U* test was used for statistical analysis, with exact *P* values indicated. ns ≥ 0.05. Error bars represent the mean ± standard error of the mean (SEM).

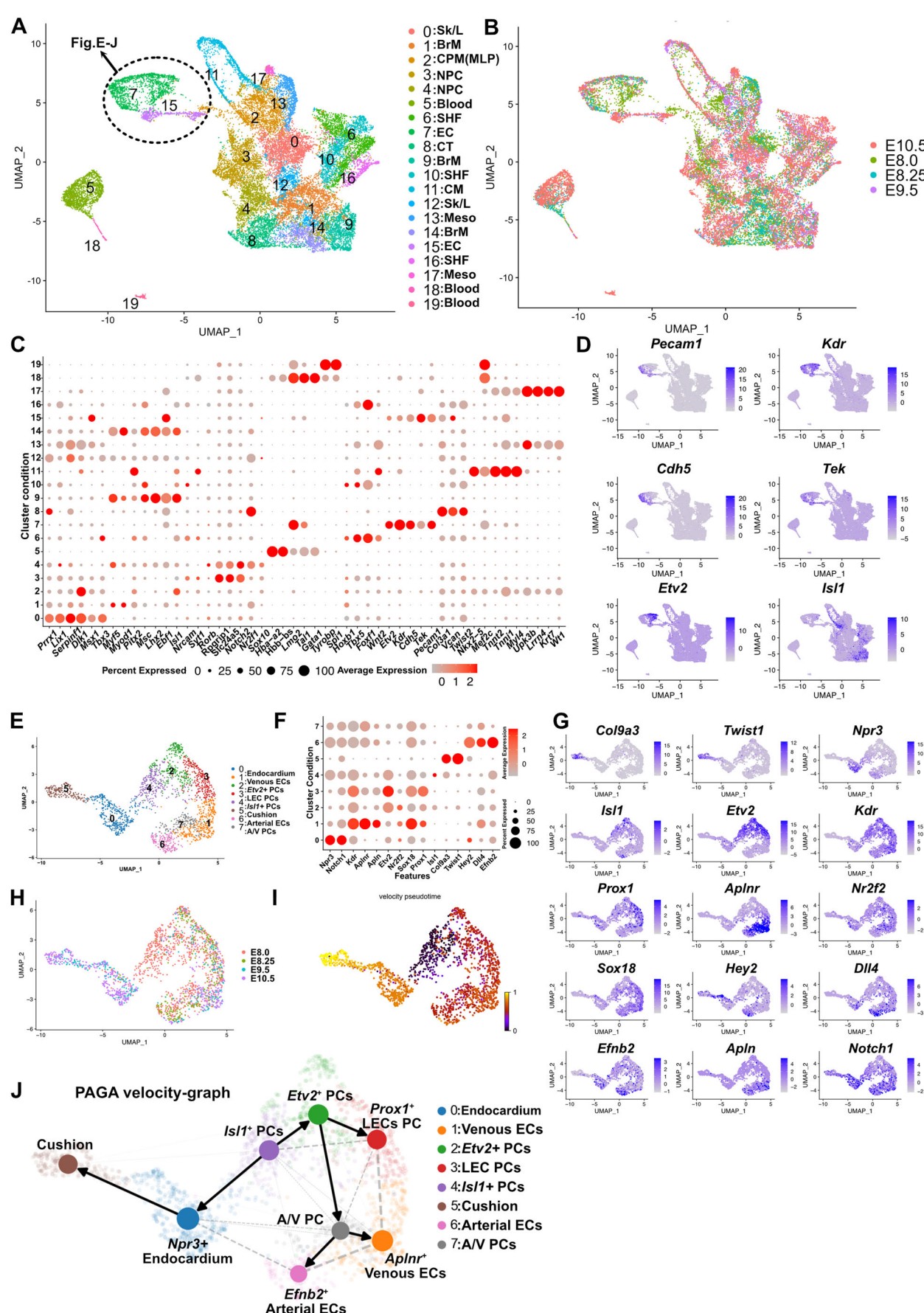

◀ **Figure EV5.  Predicted differentiation lineage of endothelial cells from the CPM.**

(A–G) Re-analysis of scRNA-seq data (Nomaru et al, 2021) of Mesp1$^+$ CPM at E8.0-E10.5. (**A**) UMAP plot with clusters color-coded (0–19): 0: Sk/L, skeleton/limb progenitor cells; 1: BrM, branchial muscle; 2: CPM (MLP), cardiopharyngeal mesoderm (multilineage progenitor cells); 3, 4: NPC, neural progenitor cells; 5: blood, blood cells; 6: SHF, second heart field; 7: EC, endothelial cells; 8: CT, Connective tissue; 9: BrM, branchial muscle; 10: SHF, second heart field; 11: CM, cardiomyocyte; 12: Sk/L, skeleton/limb progenitor cells; 13: Meso, mesothelium; 14: BrM, branchial muscle; 15: EC, endothelial cell; 16: SHF, second heart field; 17: Meso, mesothelium; 18,19: blood, blood cells. Black dotted circle indicates endothelial clusters, which underwent sub-clustering analysis. (**B**) UMAP plot color-coded by embryonic day: green (E8.0), light blue (E8.25), purple (E9.5), and red (E10.5). (**C**) Heatmap displaying the average expression levels of marker genes per cluster. (**D**) UMAP plot showing the expression levels of endothelial markers. (**E**) UMAP plot with clusters color-coded (0-7): 0: Endocardium; 1: Venous EC, venous endothelial cells; 2: Etv2$^+$ PCs, Etv2$^+$ endothelial progenitor cells; 3: LEC PCs, lymphatic endothelial progenitor cells; 4: Isl1$^+$ PCs, Isl1$^+$ endothelial progenitor cells; 5: Cushion, cushion tissue; 6: Arterial ECs, arterial endothelial cells; 7: A/V PCs, arterial and venous endothelial progenitor cells. (**F**) Heatmap displaying the average expression levels of marker genes per cluster. (**G**) Heatmap displaying the average expression levels of marker genes per cluster. (**H**) UMAP plot, color-coded by embryonic day, with red (E8.0), green (E8.25), light blue (E9.5), and purple (E10.5). (**I**) UMAP plot representing pseudotime calculated from RNA velocity analysis. (**J**) PAGA graph illustrating the predicted differentiation lineage of CPM-derived endothelial cells based on RNA velocity.

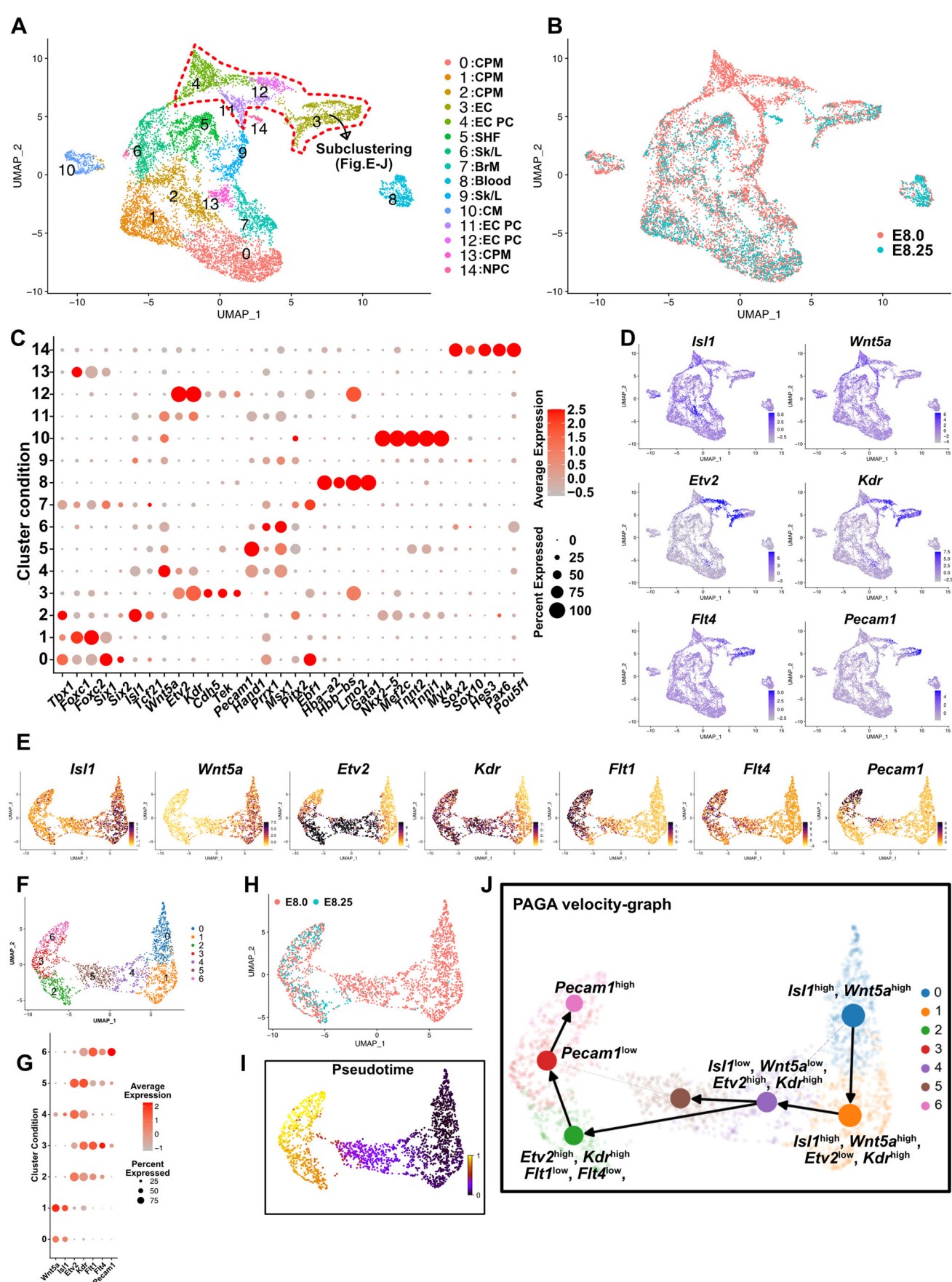

◀ **Figure EV6. The downregulation of *Isl1* and the expression of *Etv2* drive the differentiation of endothelial cells from the CPM.**

(A) UMAP plot color-coded by cluster (0-14): 0, 1, 2: CPM, cardiopharyngeal mesoderm; 3: EC, endothelial cells; 4: EC PC, endothelial cell progenitors; 5: SHF, second heart field; 6, 9: Sk/L, skeleton/limb progenitor cells; 7: BrM, branchial muscle; 8: blood, blood cell; 9: BrM, branchial muscle; 10: CM, cardiomyocyte; 11, 12: EC PC, endothelial cell progenitors; 13: CPM, cardiopharyngeal mesoderm; 14: NPC, neural progenitor cells. (B) UMAP plot color-coded by embryonic day, with red representing E8.0 and light blue representing E8.25. (C) Heatmap showing the average expression levels of marker genes for each cluster. (D) UMAP plot displaying the expression levels of CPM or endothelial cell marker genes. (E) Sub-clustering analysis of clusters 3, 4, 11, and 12, showing the endothelial cell marker genes. (F) UMAP plot with clusters color-coded (0-6). (G) Heatmap showing the average expression levels of marker genes for each cluster. (H) UMAP plot, color-coded by embryonic day (red: E8.0, light blue: E8.25). (I) UMAP plot showing pseudotime calculated from RNA velocity analysis. (J) PAGA graph illustrating the predicted differentiation trajectory from CPM to endothelial cells based on RNA velocity.

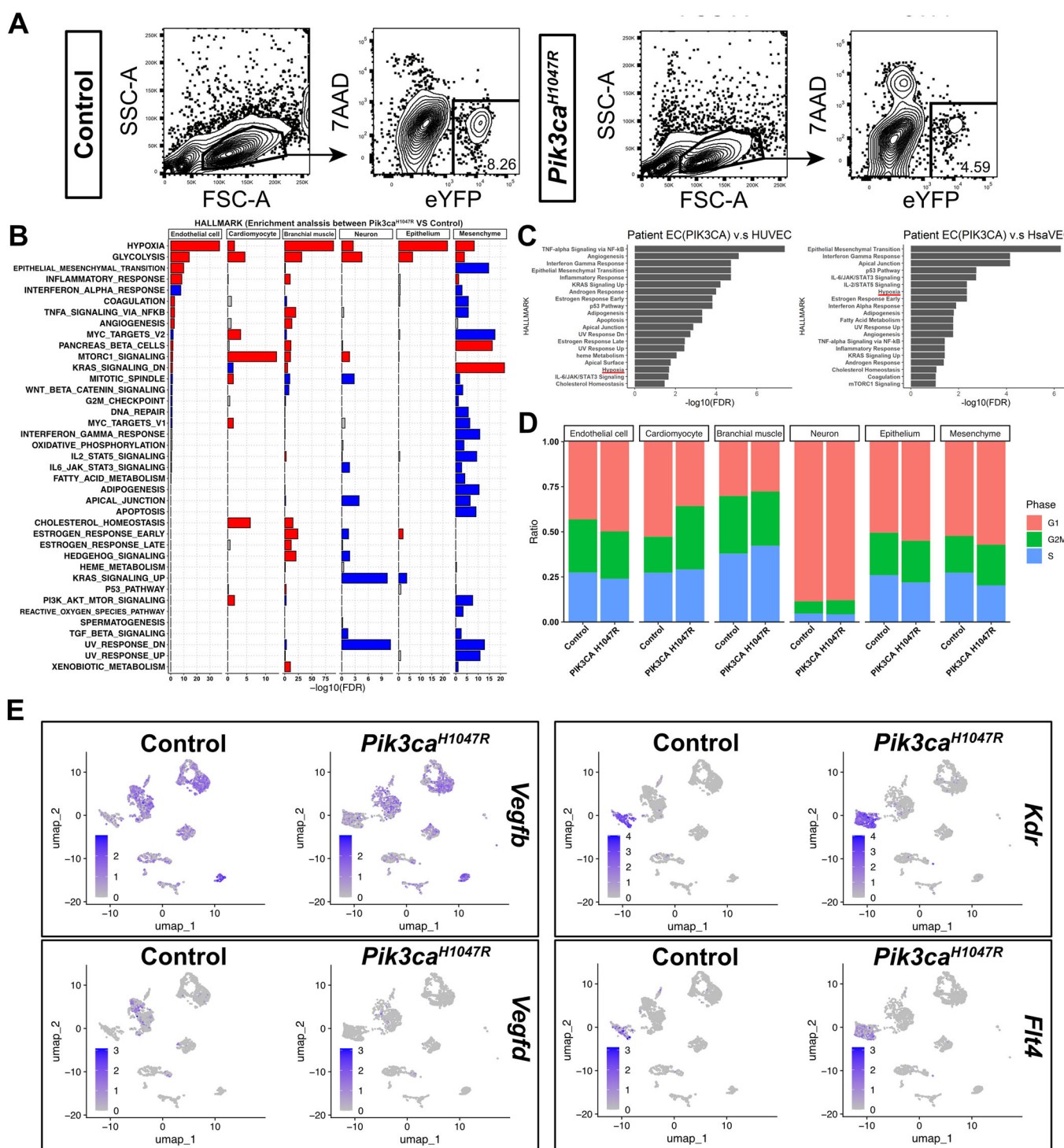

**Figure EV7. Pik3ca^H1047R expression enhances hypoxia signaling across multiple cell types.**

(A, B, D, E) scRNA-seq analysis of eYFP+ cells sorted by FACS from *Isl1-cre;R26R-eYFP* and *Isl1-cre;R26R-Pik3ca^H1047R;R26R-eYFP* embryos. (A) FACS sorting of eYFP+ cells. (B) Enrichment analysis from scRNA-seq comparing different cell types. Each bar chart shows Hallmark gene sets that exhibited significant changes in different cell types when comparing control and mutant groups. Red bars indicate Hallmark gene sets with higher enrichment scores in the mutant group, blue bars indicate higher scores in the control group, and gray bars indicate non-significant gene sets. The x axis represents -log10(FDR), and significant Hallmark gene sets were defined as FDR < 0.1. (C) Re-analysis of bulk-RNA-seq data (Jauhiainen et al, 2023) from endothelial cells derived from PIK3CA mutated venous malformations (VM) (Patient EC (PIK3CA)) compared to control endothelial cells (Human umbilical venous endothelial cells: HUVEC: or Human saphenous vein endothelial cells: HsaVEC). The left bar chart shows the top 20 significantly enriched Hallmark gene sets in *PIK3CA* mutated Patient ECs compared to HUVEC, while the right chart shows enrichment in *PIK3CA* mutated Patient ECs compared to HsaVEC. Differentially expressed genes were defined as having a fold change >1.5 and FDR < 0.05. The x axis represents -log10(FDR), and significant Hallmark gene sets were defined as FDR < 0.1. (D) Proportions of cells in each phase of the cell cycle (G1/S/G2-M) across cell types. (E) UMAP plots showing expression levels of *Vegfb, Vegfd, Kdr,* and *Flt4* by condition.

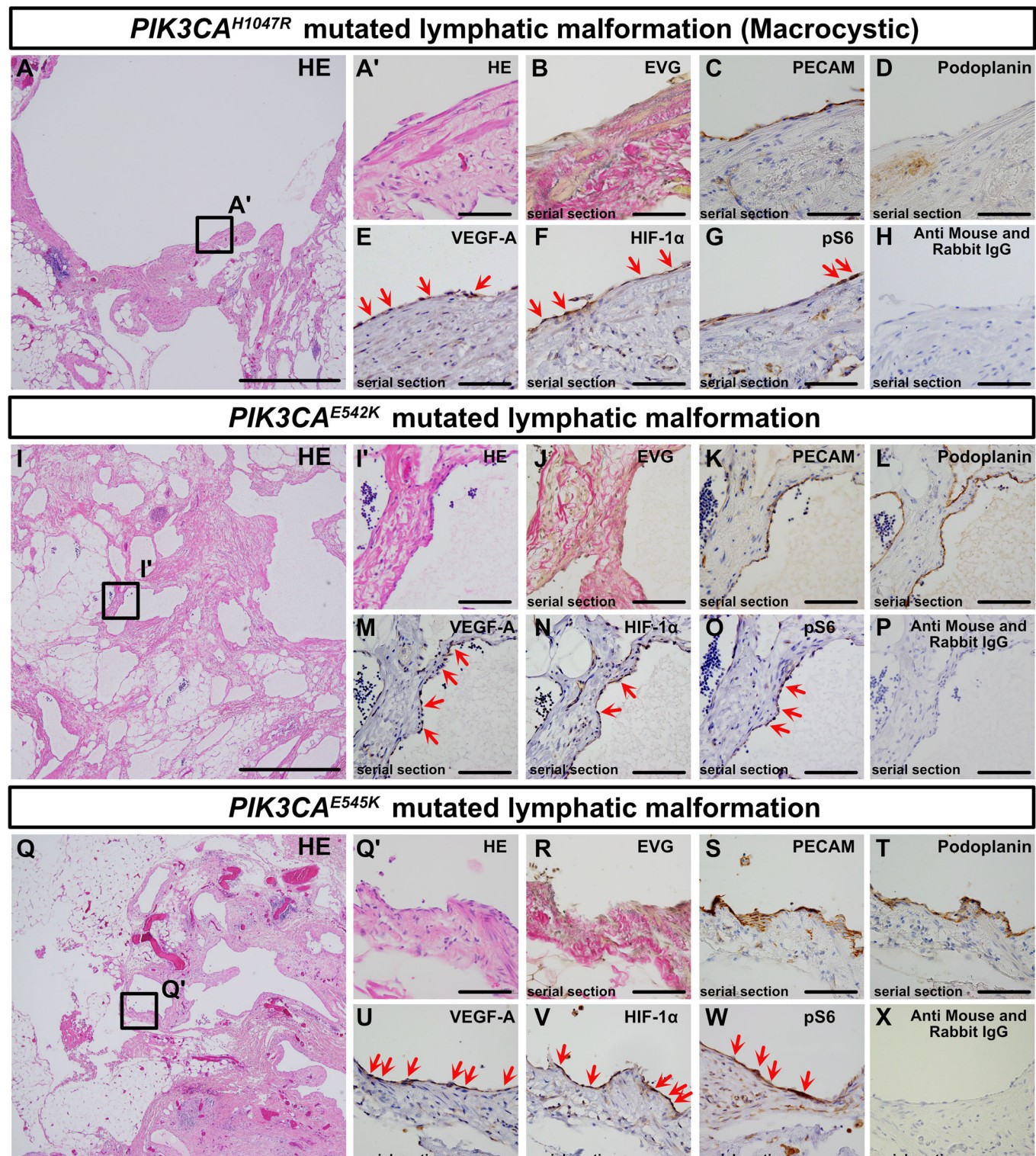

**Figure EV8. HIF-1α and VEGF-A expression in malformed lymphatic endothelial cells with *PIK3CA^E542K* and *PIK3CA^E545K* mutations.**

(A–X) Hematoxylin and eosin (HE) staining, special stains [elastica van Gieson (EVG), elastica Sirius red (ESR)—collagen fibers appear red, elastic fibers black, muscle tissue yellow], and immunohistochemistry using the indicated antibodies. No signal was detected in the negative controls using only secondary antibodies (H, P, X). (A–H) Macrocystic type lymphatic malformations. VEGF-A, HIF-1α, and pS6 are expressed in the ECs of malformed lymphatic vessels (E–G, red arrows). (I–P''''') Similar expression patterns are observed in lymphatic malformations with the *PIK3CA^E542K* mutation, where VEGF-A, HIF-1α, and pS6 are detected in malformed LECs (M–O, red arrows). (Q–X) The same findings are present in lymphatic malformations with the *PIK3CA^E545K* mutation, showing VEGF-A, HIF-1α, and pS6 expression in the malformed endothelial cells (red arrows, U–X). The region outlined by a box in the original image is displayed in an enlarged view in a corresponding panel. Scale bars: 100 μm (A'–H, I'–P, Q'–X), 1 mm (A, I, Q).

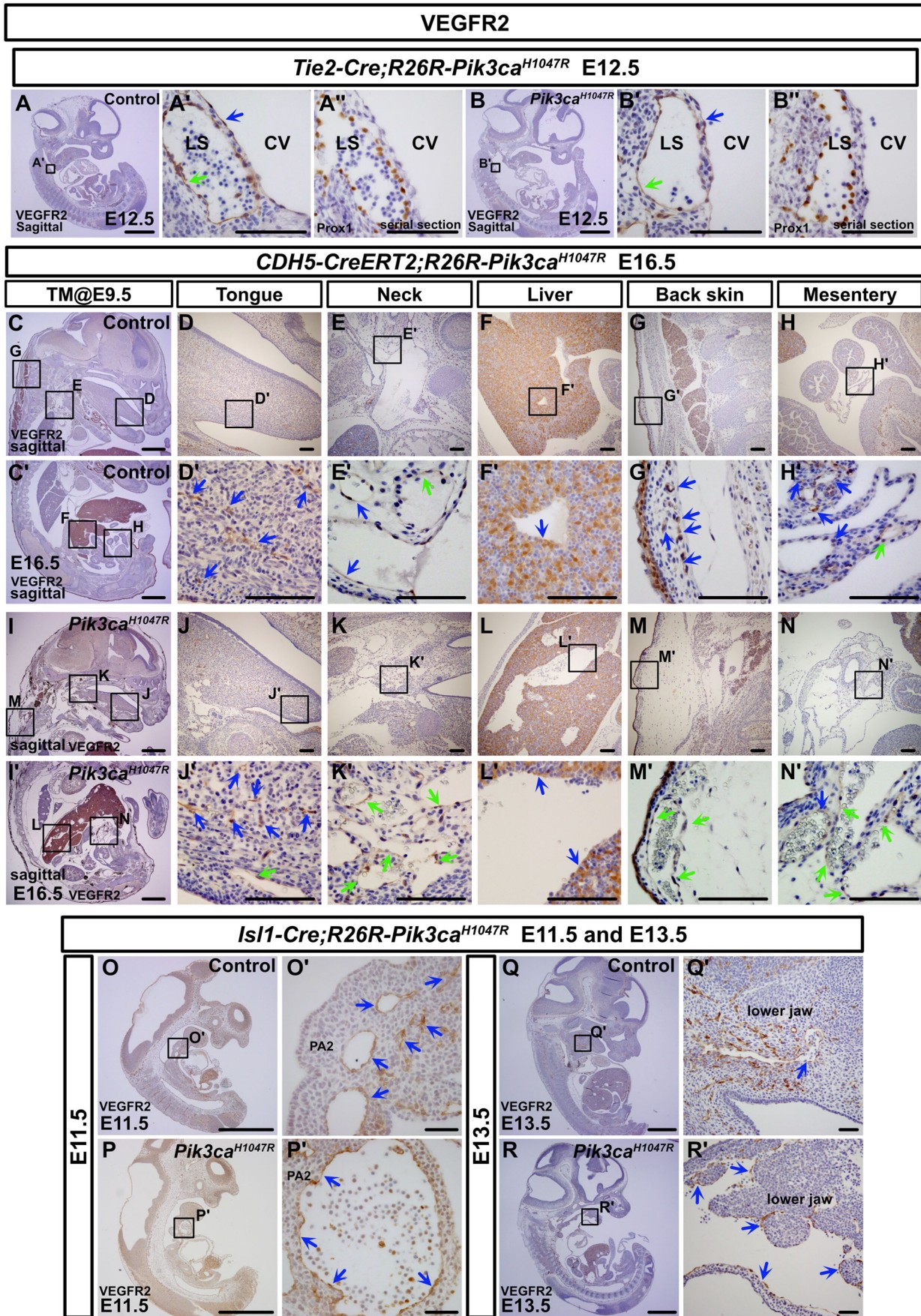

**Figure EV9. VEGFR2 is broadly expressed in blood and lymphatic vessels during embryogenesis.**

(A–R′) Sagittal sections were immunostained with the indicated antibodies. In the serial sections, lymphatic vessels can be distinguished from blood vessels based on their anatomical characteristics (in panels **A–B″**, Prox1 staining delineates the position of lymphatic vessels; in (**C–R′**), lymphatic vessels can be identified by their location in the serial sections in Figs. 2 and 3). Lymphatic vessels are marked with green arrows, and blood vessels with blue arrows. (**A–B″**) *Tie2-Cre; R26R-Pik3ca^{H1047R}* control and mutant embryos at E12.5, serial sections from Fig. 1B. (**C–N′**) *CDH5-CreERT2; R26R-Pik3ca^{H1047R}* control and mutant embryos treated with tamoxifen at E9.5 and analyzed at E16.5, serial sections from Fig. 2C, C′, J, J′. (**O–R′**) *Isl1-Cre; R26R-Pik3ca^{H1047R}* control and mutant embryos at E11.5 and E13.5, serial sections from Fig. 3B, C, E, H. CV cardinal vein, LS lymph sac, PA2 second pharyngeal arch. The region outlined by a box in the original image is displayed in an enlarged view in a corresponding panel. Figure EV9″ is the same sample image as Fig. EV1B″, but it has been reused to demonstrate the Prox1⁺ lymphatic vessels. Scale bars: 100 μm (**A′, A″, B′, B″, D–H, D′–H′, J–N, J′–N′, O′, P′, Q′, R′**), and 1 mm (**A, B, C, C′, I, I′, O, P, Q, R**).

