## [Peer Review File · EMBO Molecular Medicine]

Embryological cellular origins and hypoxia-mediated mechanisms in PIK3CA-Driven refractory vascular malformations

Sota Torii, Keiki Nagaharu, Nanako Nakanishi, Hidehito Usui, Yumiko Hori, Katsutoshi Hirose, Satoru Toyosawa, Eiichi Morii, Misyounaga Narushima, Yoshiaki Kubota, Osamu Nakagawa, Kyoko Imanaka-Yoshida, and Kazuaki Maruyama

Corresponding author: Kazuaki Maruyama (k-maruyama0608@med.mie-u.ac.jp)

Review Timeline:

Transferred from Review Commons:	13th Feb 25
Editorial Decision:	11th Mar 25
Revision Received:	19th Mar 25
Editorial Decision:	21st Mar 25
Revision Received:	23rd Mar 25
Accepted:	27th Mar 25

Editor: Lise Roth

Transaction Report:

This manuscript was transferred to EMBO Molecular Medicine following peer review at Review Commons.

Review #1**1. Evidence, reproducibility and clarity:****Evidence, reproducibility and clarity (Required)**

The authors investigate the pathogenesis of congenital vascular malformations by overexpressing the *Pik3caH1047R* mutation under the R26 locus in different cell populations and developmental stages using various Cre and CreERT2 lines, including endothelial-specific and different mesoderm precursor lines. The authors provide a thorough characterization of the vascular malformation phenotypes across models. Specifically, they claim that expressing *Pik3caH1047R* in the cardiopharyngeal mesoderm (CPM) precursors results in vascular abnormalities localized to the head and neck region of the embryo. The study also includes scRNAseq data analyses, including from previously published data and new data generated by the authors. Trajectory inference analysis of a previous scRNA-seq dataset revealed that *Isl1*⁺ mesodermal cells can differentiate into *ETV2*⁺ cells, directly giving rise to *Prox1*⁺ lymphatic endothelial cell progenitors, bypassing the venous stage. Single-cell RNA sequencing of their CPM model and other in vitro datasets show that *Pik3caH1047R* upregulates VEGF-A via HIF-1 α -mediated hypoxia signaling, findings further corroborated in human samples. Finally, preclinical studies in adult mice confirm that pharmacological inhibition of HIF-1 α and VEGF-A reduces the number and size of mutant vessels.

****Major comments****

While the study provides a nice characterization of *Pik3caH1047R*-derived vascular phenotypes induced by expressing this mutation in different cells, the main message of the study is unclear. What is the main question that the authors want to address with this manuscript? The precursor type from where these lesions appear, that venous and lymphatic malformations emerge independently, when and where this phenotype appears? The manuscript needs some work to make the sections more cohesive and to structure better the main findings and the rationale for choosing the models.

Authors should explain better when and where the pathogenic phenotypes refer to blood and/or lymphatic malformations. From the quantifications provided in Figure 1, *Pik3caH1047R* leads to different phenotypes in blood and lymphatic vessels. These are larger diameters with no difference in the number of blood vessels (are you quantifying all *pecam1* positive? Vein, arteries, capillaries?), and an increase in the number of lymphatic vessels. Please clarify and discuss.

Which vessel types are considered for the quantifications shown in Fig. 1I, M, Q? All

Pecam1+ vessels, including lymphatic, vein, capillaries and arteries or which ones? Provide clarifications.

The authors propose that the CPM model results in localized head and neck vascular malformations. However, I am not convinced. The images supporting the neck defects are evident, but it is unclear whether there are phenotypes in the head.

Why are half of the experiments with the Tie2-Cre model conducted at E12.5 (e.g., validation of recombination, signaling, proliferation) and the others at E13.5? It becomes confusing for the reader why the authors start the results section with E13.5 and then study E12.5.

The quantifications provided do not clarify what the "n" represents or how many embryos or litters were analyzed.

Blasio et al. (2018), Hare et al (2015) reported that Pik3caH1047R with Tie2-Cre embryos die before E10.5. How do the authors explain the increase in survival here? Were embryos at E13.5 alive? What was the Mendelian ratio observed by the authors? Please provide this information and discuss this point.

Please explain the rationale for using the Cdh5-CreERT2. It is likely due to the lethality observed with Tie2Cre, but this was not mentioned. Including this information will help readers who may need to become more familiar with the vasculature or the different Cre lines. Why were tamoxifen injections done at various time points (E9.5, E12.5, E15.5)? Please clarify the reasoning behind administering tamoxifen at these specific times. Explaining the rationale will help the reader follow the experimental design more easily. Additionally, including an initial diagram summarizing all the strategies to guide the reader from the beginning would be helpful.

Why do you use the Isl1-Cre constitutive line (instead of the CreERT2)? The former does not allow control of the timing of recombination (targeting specifically your population of interest) and loses the ability to trace the mutant cell behaviors over time. Is the constitutive expression of Pik3caH1047R in Isl1+ cells lethal at any embryonic time, or do the animals survive into adulthood? When you later use the Isl1-CreERT2 line, why do you induce recombination specifically at E8.5? It would be helpful for the reader to have an explanation for this choice, along with a reference to your previous paper. What is the purpose of using this battery of CreERT2 lines (for example, the Myf5-CreERT2)?

I find the scRNAseq data in Fig S4 and S5 results very interesting, although I am unsure how they fit with the rest of the story. In principle, a subset of Isl1+ cardiopharyngeal mesoderm (CPM) derivatives into lymphatic endothelial cells was already demonstrated in a previous publication from the group. What is the novelty and purpose here?

Why in Fig. 4 ECs were not subclustered for further analysis (as in Fig. S4,5)? This is a missed opportunity to understand the pathogenic phenotypes.

Hypoxia and glycolysis signatures are not specific to mutant ECs. Do the authors have an

explanation for this? It is well known that PI3K overactivation increases glycolysis; please acknowledge this. Do you have an explanation for the expression of VEGFA by lymphatic mutant cells?

Likewise, why mesenchymal cells traced from the *Isl1-Cre* decreased upon expression of *Pik3caH1047R*?

Authors need to characterize the preclinical model before conducting any preclinical study. No controls are provided, including wild-type mice and phenotypes, before starting the treatment (day 4).

Why did the authors not use their developmental model of head and neck malformation model for preclinical studies? This would be much more coherent with the first part of the manuscript. Also, how many animals were treated and quantified for the different conditions?

****Minor Comments****

References in the introduction need to be revised. Specifically, how authors reached the stats on head and neck vascular malformations needs to be clarified. For instance, one of the cited papers refers to all types of vascular malformation, while the other focuses exclusively on lymphatic malformations with PIK3CA mutations. Moreover, in the latter, the groups are divided into orofacial and neck and body categories. How do authors substrate the information from the neck and head here? Also, in line 79, I need clarification on ref 24 about fibrosis. Include references: Studies in mice have shown that p110 α is essential for normal blood and lymphatic vessel development.

Please clarify and correct.

Please define PIP2 and PIP3

Why is Prox1 showing positivity in erythrocytes in Figure 1?

Regarding Figure 1, I suggest organizing the quantifications in the same order to facilitate phenotype comparisons. For example, I, J vs. Q, R. What is the difference between M and N?

Add the reference of the Bulk RNseq data.

Mark in the Fig. 4F that the volcano plots are from cluster one of the scRNASeq (this is explained in text and legend, but when you go to the figure, it isn't very clear).

Please label Figure 6D/E with the proper labels.

In Fig. 6, it is mentioned that vacuoles are from the tamoxifen injection, how do you know? Do you also see them if you add oil alone (without tamoxifen) or tamoxifen in a WT background?

****Referees cross-commenting****

I completely agree with referee #2 regarding the preclinical studies. Bevacizumab, does not neutralize murine VEGFA. This is a major issue.

2. Significance:

Significance (Required)

This study addresses a timely and relevant question: the origins, onset and progression of congenital vascular malformations, a field with limited understanding. The work is novel in its approach, employing complex embryonic models that aim to mimic the disease in its native context. By focusing on the effects of Pik3caH1047R mutations in cardiopharyngeal mesoderm-derived endothelial cells, it sheds light on how these mutations drive phenotypic outcomes through specific pathways, such as HIF-1 α and VEGF-A signaling, while also identifying potential therapeutic targets.

A strong aspect of the study is the use of embryonic models, which enables the investigation of disease onset in a context that closely resembles the in vivo environment. This is particularly valuable for congenital disorders, where native developmental cues are an integral aspect of disease progression. The study also integrates advanced techniques, including single-cell RNA sequencing, to dissect the cellular and molecular responses induced by the Pik3caH1047R mutation. Moreover, from a translational perspective, it provides novel therapeutic strategies for these diseases.

Limitations of the study are (1) unclarity of the main question authors try to address, and main conclusions derived thereof; (2) the different parts of the manuscripts are not well connected, not clear the rationale; (3) scRNAseq analysis is underdeveloped; (4) characterization of the preclinical model is not provided.

Audience:

The findings presented here interest specialized audiences within developmental biology, vascular biology, and congenital disease research fields, and clinicians by providing new therapies to treat vascular anomalies. Moreover, the study's integration of single-cell and in vivo models could inspire further research in other contexts where understanding clonal behavior and signaling pathways is critical.

3. How much time do you estimate the authors will need to complete the suggested revisions:

Estimated time to Complete Revisions (Required)

(Decision Recommendation)

Between 3 and 6 months

4. Review Commons values the work of reviewers and encourages them to get credit for their work. Select 'Yes' below to register your reviewing activity at Web of Science Reviewer Recognition Service (formerly Publons); note that the content of your review will not be visible on Web of Science.

Yes

Review #2

1. Evidence, reproducibility and clarity:

Evidence, reproducibility and clarity (Required)

This paper focuses on vascular malformations driven by PI3K mutation, with particular interest on the vascular defects localized at head and neck anatomical sites. The authors exploit the H1047R mutant which has been largely demonstrated to induce both vascular and lymphatic malformation. To limit the effect of H1047R to tissues originated from cardiopharinegal mesoderm, PI3caH1047R mice were crossed with mice expressing Cre under the control of the promoter of *Ils1*, a transcription factor that contributes to the development of cardiopharinegal mesoderm-derived tissues. By comparing the embryo phenotype of this model with that observed by inducing at different times of development the expression of PI3caH1047R, the authors conclude that *Isl-Cre; PI3caH1047R; R26R-eYFP* model recapitulates better the anatomical features of human vascular malformations and in particular those localized at head and neck. In my opinion the new proposed model represents a significant progress to study human vascular malformations. Furthermore, scRNA seq analysis has allowed to propose a mechanism focused on the role of HIF and VEGFA. The authors provides partial evidences that HIF and VEGFA inhibitors halt the development of vascular malformation in *VeCAAdCre; Pik3caH1047* mice. This experiment is characterized by a conceptual mistake because bevacizumab does not recognize murine VEGFA (see for instance 10.1073/pnas.0611492104; 10.1167/iovs.07-1175. This error dampens my enthusiasm

****Criticism****

Fig 1A. E13.5 corresponds to the early phase of vascular remodelling. Which is the phenotype at earliest stages (e.g. 9.5 or 10.5)

Fig 1,2,3. The analysis of VEGFR2 expression is required. This request is important for the paradigmatic and non-overlapping role of this receptor in early and late vascular development. Furthermore, these data better clarify the mechanism suggested by the experiments reported in fig 5 (VEGFA and HIF expression)

As done in Fig 1,2 and 3, data quantification by morphometric analysis is also required for results reported in supplemental figure 3

Lines 166-174. I suppose that the reported observations were done at E16.5. What happens later? It's crucial to sustain the statement at lines 187-190

scRNAseq was performed at E13.5 (Fig 4). It's mandatory to perform the same analysis at E16.5, which corresponds to the phenotypic analysis shown in fig 3. This experiment is required to understand how hypoxia and glycolysis genes changes along the development of the vascular malformation.

Lines 326-343. In this section the authors provide pharmacological evidences that HIF and VEGFA are involved in vascular malformation caused by H1047R. However, I'm surprised of efficacy of bevacizumab, which neutralizes human but not murine VEGFA. Genetech has developed B20 mAb that specifically neutralizes murine VEGFA. So the data shown require a. clarification by the authors and the experiments must be done with the appropriate reagent.

Furthermore, which is the pharmacokinetics of these compounds topically applied?

****Referees cross-commenting****

The issues raised by referee #1 related to the phenotype analysis are right. In my opinion the Isl model here proposed well mimic human pathology evenf the vascular damage at head is not so evident

2. Significance:

Significance (Required)

General assessment

Strength: a new mouse model seems to well recapitulate human vascular malformation.
Possible key molecules have been identified

Weakness: The pharmacological approach to support the role of VEGFA e HIF is not appropriate

3. How much time do you estimate the authors will need to complete the suggested revisions:

Estimated time to Complete Revisions (Required)

(Decision Recommendation)

Between 3 and 6 months

Yes

Full Revision

Manuscript number: RC-2024-02767

Corresponding author(s): Kazuaki Maruyama

1. General Statements

Response to Reviewer #1:

We sincerely appreciate your thoughtful review of our manuscript. Our primary objective is to elucidate the pathogenic mechanisms underlying congenital low-flow vascular malformations, thereby informing the development of novel therapeutic strategies. We recognize that, given the dual nature of our study encompassing both fundamental and clinical science, the presentation may have appeared somewhat convoluted. In response, we have revised the manuscript to clarify these points and have reformatted the text corresponding to your comments—originally presented as a single continuous block—into defined, numbered sections to enhance readability.

Response to Reviewer #2:

We are deeply grateful for the time and effort you have dedicated to reviewing our manuscript despite your busy schedule. Your comments have been particularly insightful, especially regarding the section on the preclinical mouse model. In light of your suggestions, we have conducted additional experiments and revised the manuscript accordingly. We trust that these modifications address your concerns and contribute to the overall improvement of our work.

The revised sections have been highlighted in red in the text.

Reviewer #1 (Evidence, reproducibility and clarity (Required):

The authors investigate the pathogenesis of congenital vascular malformations by overexpressing the *Pik3ca*H1047R mutation under the R26 locus in different cell populations and developmental stages using various Cre and CreERT2 lines, including endothelial-specific and different mesoderm precursor lines. The authors provide a thorough characterization of the vascular malformation phenotypes across models. Specifically, they claim that expressing *Pik3ca*H1047R in the cardiopharyngeal mesoderm (CPM) precursors results in vascular abnormalities localized to the head and neck region of the embryo. The study also includes scRNAseq data analyses, including from previously published data and new data generated by the authors. Trajectory inference analysis of a previous scRNA-seq dataset revealed that *Isl1*⁺ mesodermal cells can differentiate into *ETV2*⁺ cells, directly giving rise to *Prox1*⁺ lymphatic endothelial cell progenitors, bypassing the venous stage. Single-cell RNA sequencing of their CPM model and other in vitro datasets show that

Pik3caH1047R upregulates VEGF-A via HIF-1 α -mediated hypoxia signaling, findings further corroborated in human samples. Finally, preclinical studies in adult mice confirm that pharmacological inhibition of HIF-1 α and VEGF-A reduces the number and size of mutant vessels.

Major comments

1. While the study provides a nice characterization of Pik3caH1047R-derived vascular phenotypes induced by expressing this mutation in different cells, the main message of the study is unclear. What is the main question that the authors want to address with this manuscript?

Response:

Our main message is as follows:

1. Elucidation of pathogenesis based on developmental cellular origins:

This study focuses on using embryonic models to elucidate the mechanism by which the Pik3ca^{H1047R} mutation induces low-flow vascular malformations. Specifically, we demonstrate that expression of Pik3ca^{H1047R} in cells derived from the cardiopharyngeal mesoderm (CPM) induces vascular abnormalities that are confined to the head and neck region. Furthermore, vascular malformations originating from another cell type—for example, Pax3⁺ cells—are confined to the lower body. This suggests that the embryonic origin of endothelial cells may determine the anatomical location of vascular malformations, with important implications for clinical severity and treatment strategies.

2. Molecular signaling pathways and targeted therapeutic approaches:

Through single-cell RNA sequencing, we have identified hypoxia signaling—particularly via HIF-1 α and VEGF-A—as central to the pathogenesis of these malformations. Moreover, preclinical mouse model experiments demonstrate that pharmacological inhibition of HIF-1 α and VEGF-A significantly reduces lesion formation, supporting the potential of targeting these pathways as a novel therapeutic strategy.

In summary, our main message is that by elucidating the developmental and molecular mechanisms underlying Pik3ca^{H1047R}-driven low-flow vascular malformations—especially the pivotal role of hypoxia signaling via HIF-1 α /VEGF-A—we provide a strong rationale for novel therapeutic strategies aimed at these challenging conditions.

To further clarify these points, we have revised the manuscript by incorporating additional experiments and reorganizing the text into clearly defined sections.

2. The precursor type form where these lesions appear, that venous and lymphatic malformations emerge independently, when and where this phenotype appear?

Response:

In *Tie2-Cre; R26R-Pik3ca^{H1047R}* mutant embryos, no prominent phenotype was observed at E9.5 or E11.5. Vascular (venous) malformations are evident from E12.5, whereas lymphatic malformations become prominent from E13.5. We propose that the emergence of the lymphatic phenotype after E13.5 is due to the fact that lymphatic vessels, particularly in the upper body, begin forming a luminal structure mainly from E13.5 onward (Maruyama *et al*, 2022). For further details, please refer to the explanation provided in Question 6.

To address this, we have newly included **Supplemental Figure 2** and revised the Results section as follows:

Whereas clear phenotypes were evident at E12.5 and E13.5, no pronounced external abnormalities were observed at E9.5 or E11.5 (Supplemental Figure 2A–B). Similarly, histological examination revealed no significant differences in the short-axis diameter of the PECAM⁺ CV or in the number of Prox1⁺ LECs surrounding the CV between control and mutant embryos at E11.5 (Supplemental Figure 2C–F). We also assessed Tie2-Cre; R26R-Pik3ca^{H1047R} mutant embryos at E14.0 from five pregnant mice. Only two embryos were alive at this stage, and both showed severe edema and hemorrhaging, indicating they were nearly moribund. These observations suggest that the critical point for survival of these mutant embryos lies between E13.5 and E14.0 (Supplemental Figure 2G). (Page 5, lines 157–165)

3. The manuscript needs some work to make the sections more cohesive and to structure better the main findings and the rationale for choosing the models. Authors should explain better when and where the pathogenic phenotypes refer to blood and/or lymphatic malformations. From the quantifications provided in Figure 1, *Pik3ca^{H1047R}* leads to different phenotypes in blood and lymphatic vessels. These are larger diameters with no difference in the number of blood vessels (are you quantifying all *pecam1* positive? Vein, arteries, capillaries?), and an increase in the number of lymphatics vessels. Please clarify and discuss.

Response:

We interpreted this as a question regarding which vessels were quantified. The answer to this question is provided in Question 4.

4. Which vessel types are considered for the quantifications shown in Fig. 1I, M, Q? All *Pecam1*⁺ vessels, including lymphatic, vein, capillaries and arteries or which ones? Provide clarifications.

Response:

Vessel types were characterized based on anatomical and histological features. For the anatomical details, we referred to *The Atlas of Mouse Development* by M.H. Kaufman.

This aspect is described in the Methods section, as follows:

Veins and arteries were classified based on anatomical criteria. Vessels demonstrating continuity with a clearly identifiable vein (e.g., the anterior cardinal vein) in serial sections were defined as veins. In contrast, the aorta and pulmonary artery, each exhibiting a distinct wall structure indicative of a direct connection to the heart, were designated as arteries. Lymphatic vessels were identified based on the combined expression of Prox1, VEGFR3, and PECAM, along with the developmental stage, morphology, and anatomical location as described in our previous studies (Maruyama et al, 2019, 2022, 2021). PECAM⁺ vessels that lacked a definitive wall structure, did not express lymphatic markers, or did not exhibit clearly identifiable continuity necessary for classification as veins or capillaries were collectively designated as blood vessels or vasculatures. (Page 16, lines 530-539)

Regarding Figure 1I:

In the tongue and mandible, the facial vein—which branches from the anterior cardinal vein—is dilated, and its continuity with the venous system is confirmed. In contrast, **Figure 1J** shows the number of PECAM⁺ vasculatures; however, for smaller vessels, continuity is not always demonstrable, so these are designated as vasculatures according to the criteria.

Regarding Figures 1M and N:

In the liver, the dilated vessels are classified as veins because they exhibit continuity with the inferior vena cava. Even in the control group, the central veins tend to have relatively large diameters. Therefore, we compared the average area and quantified the number of abnormal central veins—defined as those contiguous with a vein and exceeding a specified area.

Regarding Figures 1Q and R:

Cerebral vessels are classified as veins due to their continuity with the common cardinal and jugular veins. However, as these vessels extend into the periphery, this continuity becomes less distinct, and they are consequently designated as blood vessels lacking Prox1 expression.

5. The authors propose that the CPM model results in localized head and neck vascular malformations. However, I am not convinced. The images supporting the neck defects are evident, but it is unclear whether there are phenotypes in the head.

Response:

Perhaps the discrepancy arises from a terminological issue. According to the *WHO Classification of Tumours*, commonly used in clinical settings, the term "Head and Neck" refers to the facial and cervical regions (including the oral cavity, larynx, pharynx, salivary glands, nasal cavity, etc.) and excludes the central nervous system. The inclusion of the brain in **Figure 10-R** may have led to some confusion. We included the brain because cerebral cavernous malformations are classified as venous malformations, and thus serve as an example of common sites for venous malformations in humans. To clarify this point, we have made slight revisions to the first part of the Introduction, as follows:

They frequently manifest in the head and neck region—here defined as the orofacial and cervical areas, excluding the brain. (Page2, lines 52-53)

6. Why are half of the experiments with the Tie2-Cre model conducted at E12.5 (e.g., validation of recombination, signaling, proliferation) and the others at E13.5? It becomes confusing for the reader why the authors start the results section with E13.5 and then study E12.5.

Response:

This is also related to the previous question (Question 4). We decided to include extensive anatomical information in a single figure. In **Supplemental Figure 1**, sagittal sections at E12.5 were used so that the pulmonary artery, aorta, and dilated common cardinal vein could be visualized within one sample. This allowed us to demonstrate that the $Pik3ca^{H1047R}$ mutation does not affect arteries by contrasting them with the dilated veins. At E13.5, in addition to the dilation observed at E12.5, the common cardinal vein becomes markedly dilated and compresses the surrounding structures. Capturing both veins and arteries simultaneously would require multiple images, which could potentially confuse the reader. Moreover, lymphatic and other organ phenotypes (e.g., in the liver) are more prominent at E13.5. Therefore, we selectively employed both E12.5 and E13.5 stages to suit our specific objectives.

7. The quantifications provided do not clarify what the "n" represents or how many embryos or litters were analyzed.

Response:

Thank you for your feedback. We have now incorporated the sample size (n) directly into the graphs and figure legends.

8. Blasio et al. (2018), Hare et al (2015) reported that $Pik3ca^{H1047R}$ with Tie2-Cre embryos die before E10.5. How do the authors explain the increase in survival here? Were embryos at E13.5 alive? What was the Mendelian ratio observed by the authors? Please provide this information and discuss this point.

Response:

Two types of *Tie2-Cre* lines are widely used worldwide. The mouse line employed by Blasio et al. (2018) differs from that used in our study (their manuscript did not specify whether the background was B6 or a mixed strain). In contrast, although Hare et al. (2015) used the same mouse line as we did, they maintained a C57BL/6 background. We selected a mixed background of B6 and ICR, as we believe that a heterogeneous genetic background more accurately reflects the diversity of human pathology. We examined five pregnant females, which yielded approximately 30 embryos from five pregnant mice, of which only two survived until E14.0. Based on these observations, we consider E13.5 to be the appropriate survival limit (see Supplemental Figure 2G for additional details). In our breeding strategy,

mice in the *Tie2-Cre* or *Tie2-Cre; R26R-eYFP* line were maintained as heterozygotes for *Tie2-Cre* and homozygotes for *R26R-eYFP*, whereas those carrying the *R26R-Pik3ca^{H1047R}* allele were homozygous. This approach produced control(Cre (-)) and heterozygous offspring in an expected 1:1 ratio at all examined stages: E9.5 (mutant n = 4, control n = 4 from two pregnant females), E11.5 (mutant n = 8, control n = 8 from two pregnant females), E12.5 (mutant n = 4, control n = 4 from two pregnant females), and E13.5 (mutant n = 5, control n = 5 from two pregnant females), with no deviation from the anticipated Mendelian ratio.

Regarding this point, we have described it in the Results section as follows:

Whereas clear phenotypes were evident at E12.5 and E13.5, no pronounced external abnormalities were observed at E9.5 or E11.5 (Supplemental Figure 2A–B). Similarly, histological examination revealed no significant differences in the short-axis diameter of the PECAM⁺ CV or in the number of Prox1⁺ LECs surrounding the CV between control and mutant embryos at E11.5 (Supplemental Figure 2C–F). We also assessed Tie2-Cre; R26R-Pik3ca^{H1047R} mutant embryos at E14.0 from five pregnant mice. Only two embryos were alive at this stage, and both showed severe edema and hemorrhaging, indicating they were nearly moribund. These observations suggest that the critical point for survival of these mutant embryos lies between E13.5 and E14.0 (Supplemental Figure 2G). (Page 5, lines 157-165)

9. Please explain the rationale for using the *Cdh5-CreERT2*. It is likely due to the lethality observed with *Tie2Cre*, but this was not mentioned.

Response:

Thank you very much for your comment. As mentioned above, nearly all *Tie2-Cre;Pik3ca^{H1047R}* embryos fail to survive past E14.0.

The lethality observed with *Tie2-Cre* mice is described as follows:

We also assessed Tie2-Cre; R26R-Pik3ca^{H1047R} mutant embryos at E14.0 from five pregnant mice. Only two embryos were alive at this stage, and both showed severe edema and hemorrhaging, indicating they were nearly moribund. These observations suggest that the critical point for survival of these mutant embryos lies between E13.5 and E14.0 (Supplemental Figure 2G). (Page 5, lines 161-165)

The rationale for using *CDH5-CreERT2* mice is described as follows:

*To investigate whether the resulting human disease subtype (e.g., lesions confined to the head and neck region) is determined by the specific embryonic stage at which **Pik3ca^{H1047R}** is expressed, we crossed tamoxifen-inducible, pan-endothelial *CDH5-CreERT2* mice with *R26R-Pik3ca^{H1047R}* mice and analyzed the embryos at E16.5 or E17.5.* (Page 5, lines 169-172)

10. Why were tamoxifen injections done at various time points (E9.5, E12.5, E15.5)? Please clarify the reasoning behind administering tamoxifen at these specific times. Explaining the

rationale will help the reader follow the experimental design more easily. Additionally, including an initial diagram summarizing all the strategies to guide the reader from the beginning would be helpful.

Response:

Martinez-Corral et al. (Nat. Commun., 2020) focused on lymphatic malformations, arguing that the timing of tamoxifen administration during the embryonic period determines the anatomical features of these lesions. They stated, “The majority of lesions appeared as large isolated cysts that were localized mainly to the cervical, and less frequently to the sacral region of the skin (Figure 2)”. Although not stated definitively, their data suggest that early embryonic tamoxifen administration results in the formation of large-caliber lymphatic vessels with region-specific distribution in the cervical skin (Figure 2C, Supplemental Figure 2). This description likely reflects an intention to model human vascular malformations, implying that the anatomical characteristics of these malformations are influenced by the developmental stage at which the *Pik3ca*^{H1047R} somatic mutation occurs.

Inspired by these findings, we conducted experiments to determine whether altering the timing of tamoxifen administration would yield region-specific anatomical patterns in vascular malformation development. However, our results indicate that changing the timing of tamoxifen administration does not lead to an anatomical bias similar to that observed in human vascular malformations. Instead, we propose that the embryological cellular origin plays a more significant role in the formation of these human pathologies.

Regarding this section, we have slightly revised the introductory part of the Figure 2 explanation as follows:

*To investigate whether the resulting human disease subtype (e.g., lesions confined to the head and neck region) is determined by the specific embryonic stage at which *Pik3ca*^{H1047R} is expressed, we crossed tamoxifen-inducible, pan-endothelial *CDH5-CreERT2* mice with *R26R-Pik3ca*^{H1047R} mice and analyzed the embryos at E16.5 or E17.5. (Page 5, lines 169-172)*

Additionally, we have added a schematic diagram of the tamoxifen administration schedule at the beginning of **Figure 2** and **Supplemental Figure 3**.

11. Why do you use the *Isl1-Cre* constitutive line (instead of the *CreERT2*)? The former does not allow control of the timing of recombination (targeting specifically your population of interest) and loses the ability to trace the mutant cell behaviors over time. Is the constitutive expression of *Pik3ca*^{H1047R} in *Isl1*⁺ cells lethal at any embryonic time, or do the animals survive into adulthood? When you later use the *Isl1-CreERT2* line, why do you induce recombination specifically at E8.5? It would be helpful for the reader to have an explanation for this choice, along with a reference to your previous paper.

Response:

Thank you for your comments. We did attempt the same experiments using *Isl1-CreERT2* under various conditions. However, administering tamoxifen earlier than E8.5 invariably caused embryonic lethality, likely due to both *Pik3ca* activity and tamoxifen toxicity, leaving no embryos for analysis. In our previous study, repeated attempts from E6.5 to E16.5 resulted in only two surviving embryos (Maruyama et al., *eLife*, 2022, Supplemental Figure 3). We also failed to recover any live embryos with tamoxifen administration at E7.5.

Even reducing the tamoxifen dose to one-fifth did not succeed when given before E8.5. Although E8.5 administration was feasible, the observed phenotype remained mild, and no phenotype was detected at E9.5, E11.5, E12.5, or later stages. These findings align with our earlier observations that moving tamoxifen injection from E8.5 to E9.5 markedly diminishes the *Isl1*⁺ contribution to the endothelial lineage.

Furthermore, **Supplemental Figure 5 and 6** suggest that a decrease in *Isl1* mRNA, which occurs as early as E8.0–E8.25, triggers the shift toward endothelial differentiation. Considering these data and the mild phenotype at E8.5, earlier administration would be ideal for impacting *Isl1*⁺ cell fate. However, technical constraints prevented us from doing so, leading us to utilize the constitutive *Isl1-Cre* line instead.

This section was already included in the Discussion; however, for clarity, we have revised it as follows:

Given that Isl1 expression disappears at a very early stage and contributes to endothelial differentiation, experiments using Isl1-Cre or Isl1-CreERT2 mice cannot clearly distinguish between LMs, VMs, and capillary malformations, In other words, Isl1⁺ cells likely label a common progenitor population for multiple endothelial subtypes. Consequently, the diverse vascular malformations in the head and neck—including mixed venous-lymphatic and capillary malformations, as well as the macro- and microcystic subtypes of LMs—cannot be fully accounted for by this study alone. (Page 13, lines 419-425)

12. What is the purpose of using this battery of CreERT2 lines (for example, the *Myf5-CreERT2*)?

Response:

The head and neck mesoderm arises primarily from the cardiopharyngeal mesoderm and the cranial paraxial mesoderm. *Myf5-CreERT2* labels the cranial paraxial mesoderm in the facial region, which gives rise to facial skeletal muscles. Stone et al. (*Dev Cell*, 2019) reported that a subset of this lineage contributes to head and neck lymphatic vessels, whereas our study (Maruyama et al., *eLife*, 2022) found no such contribution—an ongoing point of debate. Nevertheless, expressing *Pik3ca*^{H1047R} in this lineage did not induce any vascular malformations.

Pax3-CreERT2 mice label *Pax3*⁺ paraxial mesoderm (including cranial paraxial mesoderm), which reportedly contributes to the common cardinal vein and subsequently forms trunk lymphatics (Stone & Stainier, 2019; Lupu et al., 2022). When *Pik3ca*^{H1047R}

was expressed in *Pax3*⁺ cells, we observed abnormal vasculature in the lower trunk and around the vertebrae, consistent with that report.

Synthesizing these observations with our results from *Isl1-Cre*, *Isl1-CreERT2*, and *Mef2c-AHF-Cre* lines, we propose that *Pik3ca*^{H1047R} mutations within the cardiopharyngeal mesoderm underlie the clinically significant vascular malformations seen in the head and neck region.

We have also incorporated the following explanation into the main text.

Regarding the *Pax3-CreERT2*:

The head and neck mesoderm arises primarily from the cardiopharyngeal mesoderm and the cranial paraxial mesoderm. In Pax3-CreERT2; R26R-Pik3ca^{H1047R} embryos, Pax3⁺ paraxial mesoderm (including cranial paraxial mesoderm) is labeled; this lineage reportedly contributes to the common cardinal vein and subsequently forms trunk lymphatics (Lupu et al, 2022), (Page 8, lines 247-250)

Regarding the *Myf5-CreERT2*;

In Myf5-CreERT2; R26R-tdTomato mice—which label the cranial paraxial mesoderm, particularly muscle satellite cells—crossed with R26R-Pik3ca^{H1047R}, tamoxifen was administered to pregnant mice at E9.5. (Page 8, lines 255-257)

13. I find the scRNAseq data in Fig S4 and S5 results very interesting, although I am unsure how they fit with the rest of the story. In principle, a subset of *Isl1*⁺ cardiopharyngeal mesoderm (CPM) derivatives into lymphatic endothelial cells was already demonstrated in a previous publication from the group. What is the novelty and purpose here?

Response:

This also addresses Question 11. Our aim in using the *Isl1*⁺ lineage was to determine the extent of analysis possible with this experimental system. Through reanalysis, we found that the downregulation of *Isl1* triggers a switch toward endothelial cell differentiation, with this cell fate decision occurring at a very early embryonic stage. Consequently, our single-cell analysis supports the conclusion that, regardless of the *Isl1-CreERT2* line used or the timing of tamoxifen administration, it is challenging to precisely recapitulate the fine clinical phenotypes observed in humans (e.g., lymphatic or venous malformations) with this experimental system. We believe that this single-cell analysis provides a theoretical basis for the notion that our *Isl1-Cre*-based developmental model can only generate a mixed phenotype of vascular and lymphatic malformations.

This section is explained in a similar manner in the revised Discussion for Question 11 as follows:

Given that Isl1 expression disappears at a very early stage and contributes to endothelial differentiation, experiments using Isl1-Cre or Isl1-CreERT2 mice cannot clearly distinguish

between LMs, VMs, and capillary malformations, In other words, Isl1⁺ cells likely label a common progenitor population for multiple endothelial subtypes. Consequently, the diverse vascular malformations in the head and neck—including mixed venous-lymphatic and capillary malformations, as well as the macro- and microcystic subtypes of LMs—cannot be fully accounted for by this study alone. (Page 13, lines 419-425)

14. Why in Fig. 4 ECs were not subclustered for further analysis (as in Fig. S4,5)? This is a missed opportunity to understand the pathogenic phenotypes.

Response:

Thank you for your question. We performed sub-clustering analysis, particularly focusing on why no phenotype is observed in arteries, as we believed this approach could provide molecular-level insights. Accordingly, we conducted the analysis presented in **Figure 1 for Reviewer 1**.

Figure legends for Figure 1 for Reviewer 1. The number of endothelial cells was insufficient, making subclustering ineffective.

(Figure for Reviewer 1A, B) Left: UMAP plot showing color-coded clusters (0–3). Subcluster analysis of the Endothelium (Cluster 1) from Fig. 4B. Right: UMAP plot color-coded by condition. **(Figure for Reviewer 1C)** Heatmap showing the average gene expression of marker genes for each cluster by condition. After cluster annotation,

subclusters 0, 1, 2, and 3 were defined as Vein, Capillary, Artery, and Lymphatics, respectively. **(Figure for Reviewer 1D)** Cell type proportions. **(Figure for Reviewer 1E)** Number of differentially expressed genes (DEGs) in each subcluster of the *PIK3CA*^{H1047R} group relative to Control. **(Figure for Reviewer 1F)** Comparison of enrichment analysis between EC subclusters from scRNA-seq. The bar graph shows the top 20 significantly altered Hallmark gene sets in EC subclusters from scRNA-seq using ssGSEA (escape R package). Red bars represent significantly upregulated Hallmark gene sets in mutants (FDR < 0.1), and gray bars indicate non-significant sets. **(Figure for Reviewer 1G)** Volcano plot showing hypoxia-related genes (red: mutant upregulated, blue: control upregulated). Right: Glycolysis-related genes with the same color scheme.

Initially, we performed sub-clustering on endothelial cells; however, this resulted in a considerably reduced number of cells per sub-cluster, especially in control group **(Figure for Reviewer 1A, B)**. In the control group, there were only approximately 149 endothelial cells in total, and dividing these into four clusters led to very few cells per cluster, thereby introducing statistical instability. Although arterial endothelial cells were relatively well defined by their high expression of *Hey1* and *Hey2* and lower levels of *Nr2f2* and *Aplnr*, the boundaries between venous, capillary, and lymphatic endothelial cells were less distinct. In particular, defining lymphatic endothelial cells solely by *Prox1* expression yielded a very small population; even after incorporating additional lymphatic markers such as *Flt4* and *Lyve1*, it remained challenging to clearly separate the venous, capillary, and lymphatic populations **(Figure for Reviewer 1C)**. Consequently, the proportion of lymphatic endothelial cells was markedly low, and discrepancies with the histological findings further reduced our confidence in this dataset **(Figure for Reviewer 1D, E)**. Moreover, the number of differentially expressed genes (DEGs) increased with the number of cells, and the results of the enrichment analysis as well as the volcano plot were nearly identical to those shown in Figure 4 **(Figure for Reviewer 1F, G)**. In other words, the subclustering process itself had limitations, resulting in the overall outcome being dominated by the most abundant venous cluster.

It is possible that these limitations in sub-clustering are due to the relatively small number of endothelial cells. Nonetheless, a major strength of our single-cell analysis is its ability to compare various cell types derived from *Isl1*⁺ lineages, not just endothelial cells. Therefore, the relative scarcity of endothelial cells represents a limitation of this experimental system. For these reasons, we decided to omit this figure from the final version of the manuscript.

This point is described in the Discussion section as follows:

Additionally, we performed endothelial subclustering to explore potential differences in gene expression among arterial, venous, capillary, and lymphatic endothelium. However, in the

control embryos, the number of endothelial cells was too low to yield reliable data (data not shown). (Page 13, lines 434-437)

15. Hypoxia and glycolysis signatures are not specific to mutant ECs. Do the authors have an explanation for this? It is well known that PI3K overactivation increases glycolysis; please acknowledge this.

Response:

Thank you for your important comment. We have now incorporated a discussion, along with relevant references, on the section addressing that PI3K overactivation increases glycolysis into the Discussion section as follows:

It is well known that overactivation of PI3K enhances glycolysis(Hu *et al*, 2016) . *In our study, the elevated expression of glycolytic enzymes, including Ldha, suggests a shift toward aerobic glycolysis, consistent with the Warburg effect.* (Page 13, lines447-450)

16. Do you have an explanation for the expression of VEGFA by lymphatic mutant cells?

Response:

VEGF-A acts on VEGFR2 expressed on LECs, thereby promoting their proliferation and migration(Hong *et al*, 2004; Dellinger & Brekken, 2011) .To clarify this point, we have revised the text accordingly and added additional references as follows:

We focused on Vegf-a, a key regulator of ECs proliferation and a downstream target of Hif-1a. Vegf-a likely drives both cell-autonomous and non-cell-autonomous effects on blood ECs , as well as LECs(Hong *et al*, 2004; Dellinger & Brekken, 2011). (Page 13, lines 445-447)

17. Likewise, why mesenchymal cells traced from the Islt1-Cre decreased upon expression of Pik3caH1047R?

Response:

When comparing the mesenchyme cluster with other mesoderm-derived cells, we observed a marked downregulation of signaling pathways—notably those involved in inhibiting EMT, such as TGF- β , Wnt/ β catenin, and MYC target genes (**Supplemental Figure 7B**). Many of these pathways are associated with decreased epithelial-to-mesenchymal transition(Xu *et al*, 2009; Singh *et al*, 2012; Larue & Bellacosa, 2005; Yu *et al*, 2015), which could explain the reduction in the number of mesenchymal cells. However, PI3K activation is generally considered to promote EMT, which is at odds with previous studies.

On the other hand, several investigations—including those using ES cells—suggest that PI3K activation could suppress TGF- β signaling via SMAD2/3(Yu *et al*, 2015) , and in some undifferentiated cell contexts, it may also inhibit the Wnt/ β -catenin pathway via Smad2/3(Singh *et al*, 2012) . These multifaceted roles of PI3K could be particularly important during embryonic development(Larue & Bellacosa, 2005).

Understanding how mesenchymal cell changes under PI3K activation affect endothelial cells is an important issue that requires further study. Accordingly, we have added these points to the Discussion section as follows:

In our data, the mesenchymal cell population was decreased, and within this cluster, pathways typically promoting epithelial mesenchymal transition (EMT) (e.g., TGF- β , Wnt, and MYC target genes) were downregulated (Supplemental Figure 7B). Although PI3K activation is generally thought to enhance EMT, several studies in undifferentiated cells have reported that PI3K can suppress these signals via SMAD2/3 (Singh et al, 2012; Yu et al, 2015). Elucidating how these changes in the mesenchyme contribute to vascular malformation pathogenesis remains an important avenue for future research. (Page 13, lines 437-444)

18. Authors need to characterize the preclinical model before conducting any preclinical study. No controls are provided, including wild-type mice and phenotypes, before starting the treatment (day 4).

Response:

Thank you very much for your comment. We have now added new images illustrating skin under three conditions: untreated skin at Day 7, skin from Cre-negative animals that received tamoxifen, and skin from Cre-positive animals examined 4 days after tamoxifen administration. Additionally, we have included the corresponding statistical data for these skin samples (**Figure 6C–E**).

19. Why did the authors not use their developmental model of head and neck malformation model for preclinical studies? This would be much more coherent with the first part of the manuscript. Also, how many animals were treated and quantified for the different conditions?

Response:

We have now indicated the number of animals (n) used under each condition directly on the graphs for clarity. As for why we did not use the *Isl1-Cre* model, we observed that—similar to the *Tie2-Cre* line—all *Isl1-Cre* mutant embryos died between E13.5 and E14.0 (indeed, none survived beyond E14.0; see our newly added **Figure 3N**). Consequently, we could not perform any postnatal treatment experiments. Moreover, as previously noted, the *Isl-CreERT2* line has an extremely narrow developmental window for vascular malformation formation, making it less suitable as a general model.

Although we considered potential in utero or maternal interventions (e.g., direct uterine injection or placental transfer), these approaches demand extensive technical optimization and remain an area for future investigation. From a clinical standpoint, postnatal therapy meets a more immediate need: while vascular malformations are congenital, they often enlarge over time (Ryu et al, 2023), becoming more apparent and more likely to require treatment.

In this study, because embryonic *Pik3ca*^{H1047R} expression was lethal before birth, we generated and treated postnatal cutaneous vascular malformations instead. Although this model does not strictly recapitulate the embryonic disease state, previous studies assessing drug efficacy have similarly employed postnatal tamoxifen-inducible mouse models (Martinez-Corral *et al*, 2020), lending validity to this approach. Moreover, because lesions typically become evident later in life rather than in utero, this method more closely aligns with clinical reality and may be more readily translated into practice.

Minor Comments

References in the introduction need to be revised. Specifically, how authors reached the stats on head and neck vascular malformations needs to be clarified. For instance, one of the cited papers refers to all types of vascular malformation, while the other focuses exclusively on lymphatic malformations with PIK3CA mutations. Moreover, in the latter, the groups are divided into orofacial and neck and body categories. How do authors substrate the information from the neck and head here?

Response:

We have clarified our definition of the “head and neck” region early in the Introduction and separated the discussion on anatomical localization from that on PIK3CA genetics. Additionally, we removed the percentage data of localization to avoid potential confusion with the genetic aspects.

In Japan, lymphatic and other vascular malformations of the head and neck typically require complex, multidisciplinary management. Consequently, these conditions are officially designated as “intractable diseases,” and the government provides financial assistance for their treatment. Although most of the information is available only in Japanese, we refer reviewers to the following websites for details on head and neck vascular malformations:

<https://www.nanbyou.or.jp/entry/4893>

<https://www.nanbyou.or.jp/entry/4631>

<https://www.nanbyou.or.jp/entry/4758>.

(Please read with English translator, e.g., Google chrome translator)

We are not aware of a comparable system in other countries. However, it is well recognized that vascular malformations frequently occur in the head and neck region (Nair, 2018; Alsuwailem *et al*, 2020; Sadick *et al*, 2017), as evidenced by over 250 PubMed hits when searching for “vascular malformation” and “head and neck.

Incorporating this comment, we have revised the early part of the Introduction as follows:

They frequently manifest in the head and neck region—here defined as the orofacial and cervical areas, excluding the brain (Zenner *et al*, 2019; Lee & Chung, 2018; Nair, 2018; Alsuwailem *et al*, 2020). (Page 2, lines 52-53)

Also, in line 79, I need clarification on ref 24 about fibrosis.

Response:

Thank you very much for pointing out the error. We have corrected the placement of the reference accordingly.

Include references: Studies in mice have shown that p110 α is essential for normal blood and lymphatic vessel development. Please clarify and correct.

Response:

Thank you very much. We have now added the references (Graupera *et al*, 2008; Gupta *et al*, 2007; Stanczuk *et al*, 2015).

Please define PIP2 and PIP3

Response:

Thank you very much for your comment. We have now added the following definitions to the Introduction:

PIP2: Phosphatidylinositol 4,5-bisphosphate

PIP3: Phosphatidylinositol 3,4,5-trisphosphate

Why is Prox1 showing positivity in erythrocytes in Figure 1?

Response:

We used paraffin-embedded sections to preserve tissue morphology. Although we applied a reagent to suppress autofluorescence, some spillover from excitation around 488 nm was unavoidable. Moreover, in the mutant mice, blood remained within the abnormal vessels rather than being completely flushed out, which further increased the autofluorescence. Despite our efforts to mitigate this, some residual autofluorescence persisted. Consequently, we also employed DAB-based staining to confirm the specificity of Prox1 labeling in other Figures.

Regarding Figure 1, I suggest organizing the quantifications in the same order to facilitate phenotype comparisons. For example, I, J vs. Q, R. What is the difference between M and N?

Response:

To facilitate the comparison between **Figures 1I, J and 1Q, R**, we have swapped **Figures 1Q and R**. Regarding **Figures 1M and N**, these panels represent the average cross-sectional area of an enlarged malformed vessel and the number of vessels exceeding a defined size, respectively. Although some central veins appeared slightly enlarged in the control group, the liver exhibits both a significant dilation of malformed vessels and an increased number of such vessels.

Add the reference of the Bulk RNseq data.

Response:

We have added the following references: (Jauhiainen *et al*, 2023)

Mark in the Fig. 4F that the volcano plots are from cluster one of the scRNASeq (this is explained in text and legend, but when you go to the figure, it isn't very clear).

Response:

We have added the label “Cluster 1: Volcano Plot (genes associated with hypoxia/glycolysis)” to

Figure 4F.

Please label Figure 6D/E with the proper labels.

Response:

We have provided appropriate labels for **Figure 6.**

In Fig. 6, it is mentioned that vacuoles are from the tamoxifen injection, how do you know? Do you also see them if you add oil alone (without tamoxifen) or tamoxifen in a WT background?

Response:

In **Figure 6C**, we have included both the image at Day 4 and the condition of Cre(-) animals 7 days after tamoxifen injection.

****Referees cross-commenting****

I completely agree with referee #2 regarding the preclinical studies. Bevacizumab, does not neutralize murine VEGFA. This is a major issue.

Response:

As noted in the Reviewer #2 section, there appears to be some effect on mouse vasculature (Lin *et al*, 2022). However, given the ongoing debate regarding this issue, we performed additional experiments using a neutralizing antibody against mouse VEGF-A (clone 2G11). This antibody has been shown to suppress the proliferation of mouse vascular endothelial cells *in vivo*, for example (Mashima *et al*, 2021; Wuest & Carr, 2010). Our results demonstrate that it more sharply suppresses the proliferation of malformed vasculatures (both blood and lymphatic vessels) than bevacizumab. Based on these additional experiments, we revised the figures and updated them as **Figure 6.**

Reviewer #1 (Significance (Required)):

This study addresses a timely and relevant question: the origins, onset and progression of congenital vascular malformations, a field with limited understanding. The work is novel in its approach, employing complex embryonic models that aim to mimic the disease in its native context. By focusing on the effects of Pik3caH1047R mutations in cardiopharyngeal mesoderm-derived endothelial cells, it sheds light on how these mutations drive phenotypic outcomes through specific pathways, such as HIF-1 α and VEGF-A signaling, while also

identifying potential therapeutic targets. A strong aspect of the study is the use of embryonic models, which enables the investigation of disease onset in a context that closely resembles the *in vivo* environment. This is particularly valuable for congenital disorders, where native developmental cues are an integral aspect of disease progression. The study also integrates advanced techniques, including single-cell RNA sequencing, to dissect the cellular and molecular responses induced by the *Pik3caH1047R* mutation. Moreover, from a translational perspective, it provides novel therapeutic strategies for these diseases. Limitations of the study are (1) unclarity of the main question authors try to address, and main conclusions dereived thereof; (2) the different parts of the manuscripts are not well connected, not clear the rationale; (3) scRNAseq analysis is underdeveloped; (4) characterization of the preclinical model is not provided.

Audience:

The findings presented here interest specialized audiences within developmental biology, vascular biology, and congenital disease research fields, and clinicians by providing new therapies to treat vascular anomalies. Moreover, the study's integration of single-cell and *in vivo* models could inspire further research in other contexts where understanding clonal behavior and signaling pathways is critical.

Reviewer #2 (Evidence, reproducibility and clarity (Required)):

This paper focuses on vascular malformations driven by PI3K mutation, with particular interest on the vascular defects localized at head and neck anatomical sites. The authors exploit the H1047R mutant which has been largely demonstrated to induce both vascular and lymphatic malformation. To limit the effect of H1047R to tissues originated from cardiopharinegal mesoderm, *PI3caH1047R* mice were crossed with mice expressing Cre under the control of the promoter of *Isl1*, a transcription factor that contributes to the development of cardiopharinegal mesoderm-derived tissues. By comparing the embryo phenotype of this model with that observed by inducing at different times of development the expression of *PI3caH1047R*, the authors conclude that *Isl-Cre; PI3caH1047R; R26R-eYFP* model recapitulates better the anatomical features of human vascular malformations and in particular those localized at head and neck. In my opinion the new proposed model represents a significant progress to study human vascular malformations. Furthermore, scRNA seq analysis has allowed to propose a mechanism focused on the role of HIF and VEGFA. The authors provides partial evidences that HIF and VEGFA inhibitors halt the development of vascular malformation in *VeCAdCre; Pik3caH1047* mice. This experiment is characterized by a conceptual mistake because bevacizumab does not recognize murine VEGFA (see for instance [10.1073/pnas.0611492104](https://doi.org/10.1073/pnas.0611492104); [10.1167/iovs.07-1175](https://doi.org/10.1167/iovs.07-1175)). This error dampens my enthusiasm

CRITICISM

1. Fig 1A. E13.5 corresponds to the early phase of vascular remodelling. Which is the phenotype at earliest stages (e.g. 9.5 or 10.5)

Response:

Thank you very much for your comment. We have created new **Supplemental Figure 2**, which demonstrates that no obvious phenotype is observed in mutant embryos at E9.5 and E11.5, and that the survival limit of these mutant embryos is around E13.5 to E14.0.

In response to Reviewer 1's question, previous study(Hare *et al*, 2015) have shown that on a B6 background, this mouse model exhibits an earlier onset of phenotype, resulting in early lethality. However we selected a mixed background of B6 and ICR, as we believe that a heterogeneous genetic background more accurately reflects the diversity of human pathology. We examined five pregnant females, which yielded approximately 30 embryos, of which only two survived until E14.0. Based on these observations, we consider E13.5 to E14.0 to be the appropriate survival limit (see **Supplemental Figure 2G** for additional details).

We have described this in the Results section as follows:

Whereas clear phenotypes were evident at E12.5 and E13.5, no pronounced external abnormalities were observed at E9.5 or E11.5 (Supplemental Figure 2A–B). Similarly, histological examination revealed no significant differences in the short-axis diameter of the PECAM⁺ CV or in the number of Prox1⁺ LECs surrounding the CV between control and mutant embryos at E11.5 (Supplemental Figure 2C–F). We also assessed Tie2-Cre; R26R-Pik3ca^{H1047R} mutant embryos at E14.0 from five pregnant mice. Only two embryos were alive at this stage, and both showed severe edema and hemorrhaging, indicating they were nearly moribund. These observations suggest that the critical point for survival of these mutant embryos lies between E13.5 and E14.0 (Supplemental Figure 2G). (Page 5, lines 157-165)

2. Fig 1,2,3. The analysis of VEGFR2 expression is required. This request is important for the paradigmatic and non-overlapping role of this receptor in early and late vascular development. Furthermore ,these data better clarify the mechanism suggested by the experiments reported in fig 5 (VEGFA and HIF expression)

Response:

Thank you very much for your comment. For each mouse presented in **Figures 1, 2, and 3**, we performed VEGFR2 immunostaining on serial sections corresponding to each figure and created a new **Supplemental Figure 9**. VEGFR2 was broadly expressed in both vascular and lymphatic endothelial cells in control and mutant embryos.

We have described this in the Results section as follows:

Furthermore, to verify whether VEGF-A can act via VEGFR2, we performed VEGFR2 immunostaining on several mouse models: *Tie2-Cre; R26R-Pik3ca^{H1047R}* embryos (E13.5, corresponding to Figure 1), *CDH5-CreERT2; R26R-Pik3ca^{H1047R}* embryos (tamoxifen administered at E9.5 and analyzed at E16.5, corresponding to Figure 2), and *Isl1-Cre; R26R-Pik3ca^{H1047R}* embryos (E11.5 and E13.5, corresponding to Figure 3). In all cases, both control and mutant embryos exhibited widespread VEGFR2 expression in blood and lymphatic vessels at early and late developmental stages (**Supplemental Figure 9A-R**). These findings suggest that *Pik3ca^{H1047R}* may act in an autocrine manner, at least in part via the VEGF-A/VEGFR2 axis in endothelial cells, potentially explaining the observed phenotype. (Page 11, lines352-361)

3. As done in Fig 1,2 and 3, data quantification by morphometric analysis is also required for results reported in supplemental figure 3

Response:

Thank you for your comment. We have now added additional statistics and graphs for clarity, which are presented as **Supplemental Figure 4**.

4. Lines 166-174. I suppose that the reported observations were done at E16.5. What happens later? It's crucial to sustain the statement at lines 187-190

Response:

At E9.5 and E12.5, we reduced the tamoxifen dose to one-fifth of the standard dose. After collecting embryos from approximately 10 pregnant females, we were only able to obtain three embryos at these stages. When tamoxifen was administered at E15.5, three embryos were obtained from two litters. In most cases, miscarriages occurred by E16.5, making further observation difficult. We focused on the time point around E16.5 because it is generally believed that the basic distribution of the lymphatic system throughout the body is established around this stage (Srinivasan *et al*, 2007; Maruyama *et al*, 2022).

A similar experiment has been reported using *T-CreERT2* to induce mosaic expression of *Pik3ca^{H1047R}* in the mesoderm, which resulted in subcutaneous venous malformations in mice at P1–P5 (Castillo *et al*, 2016). However, that study did not report whether the mice survived normally after birth. In fact, regarding the survival rate, the authors stated, “Our observations on the lethality and vascular defects in *MosMes-Pik3caH1047R (T-CreERT2;R26R-Pik3ca^{H1047R})* embryos are similar to the previously reported phenotypes of ubiquitous or EC-specific expression of *Pik3caH1047R* in the developing embryo (Hare *et al*, 2015),” suggesting a high mortality rate when *Pik3ca^{H1047R}* is expressed using *Tie2-Cre*. Moreover, according to Hare *et al.*, analysis of 250 *Tie2-Cre; R26R-Pik3ca^{H1047R}* embryos revealed that all were lethal by E11.5. Thus, considering our results in conjunction with those from previous studies, it appears that expression of *Pik3ca^{H1047R}* in the mesoderm or endothelial cells during embryonic development results in the death of most embryos before birth.

We have supplemented the Results section with the following details:

Since the standard tamoxifen dose (125 mg/kg body weight) leads to miscarriage or embryonic death within 1–2 days, we diluted it to one-fifth of the original concentration. (Pages 5-6, lines 175-177)

5. scRNAseq was performed at E13.5 (Fig 4). It's mandatory to perform the same analysis at E16.5, which corresponds to the phenotypic analysis shown in fig 3. This experiment is required to understand how hypoxia and glycolysis genes changes along the development of the vascular malformation.

Response:

Thank you very much for your comment. First, regarding the experiments using *Isl1-Cre*, we would like to clarify that the survival aspect was not adequately addressed. Our *Isl1-Cre* embryos die between E13.5 and E14.0, which makes it practically impossible to perform single-cell analysis beyond this stage (please refer to the newly added **Figure 4N**). Similarly, for experiments using *CDH5-CreERT2*, the limited number of embryos obtained renders further analysis extremely challenging. Additionally, we have supplemented the Results section with the following description:

*These *Isl1-Cre; R26R-Pik3ca^{H1047R}* mutant embryos likely died from facial hemorrhaging between E13.5 and E14.0 (**Figure 3N**). (Page 7, lines 236-237)*

Further analysis at later embryonic stages proved challenging. Consequently, we aimed to investigate the effects of *Pik3ca^{H1047R}* on endothelial cells by comparing gene expression at E10.5 with that at E13.5. We performed single-cell RNA sequencing on E10.5 embryos from both the control (*Isl1-Cre; R26R-eYFP*) and mutant (*Isl1-Cre; R26R-eYFP; R26R-Pik3ca^{H1047R}*) embryos. Unfortunately, the quality of both datasets was insufficient for reliable analysis. In the control sample, only 40.3% of reads were assigned to cell-associated barcodes—substantially below the ideal threshold of >70%—with an estimated 790 cells and a median of 598 genes per cell. Similarly, in the mutant sample, only 37.0% of reads were associated with cells, despite an estimated cell count of 7,326 and a median of only 526 genes per cell. These metrics indicate that both datasets were severely compromised by high levels of ambient RNA or by a significant number of cells with low RNA content, precluding robust downstream analysis. This may be due to the fact that immature cells are particularly susceptible to damage incurred during FACS sorting and transportation to the analysis facility. Moreover, the relatively low number of control endothelial cells at E13.5 led us to conclude that performing similar experiments at earlier stages would be difficult. Despite our best efforts, we acknowledge this as a limitation of the present study.

6. Lines 326-343. In this section the authors provide pharmacological evidences that HIF and VEGFa are involved in vascular malformation caused by H1047R . However , I'm surprised

of efficacy of bevacizumab, which neutralizes human but not murine VEGFA. Genentech has developed B20 mAb that specifically neutralizes murine VEGFA. So the data shown require a. clarification by the authors and the experiments must be done with the appropriate reagent. Furthermore, which is the pharmacokinetics of these compounds topically applied?

Response:

Thank you very much for your comment. There are reports that bevacizumab exerts an in vivo inhibitory effect on neovascularization mediated by mouse Vegf-A (Lin *et al*, 2022). However, given the contentious nature of this issue, we conducted additional experiments. Due to the requirement for an MTA to obtain B20 mAb from Genentech—and considering the time constraints during revision—we opted to use a neutralizing antibody against mouse VEGF-A (clone 2G11) instead. This antibody has been shown to suppress the proliferation of mouse vascular endothelial cells in vivo (Mashima *et al*, 2021; Wuest & Carr, 2010) .

The dosing regimen for 2G11 was determined based on previous studies (Surve *et al*, 2024; Churchill *et al*, 2022). Moreover, an example of effective local administration is provided in (Nagao *et al*, 2017). Since this product is an antibody drug, it is metabolized and does not function as a prodrug. Although the precise half-life of 2G11 is unknown, rat IgG2a antibodies generally have a circulating half-life of approximately 7–10 days in rats. However, when administered to mice, the half-life is often significantly reduced due to interspecies differences in neonatal Fc receptor (FcRn) binding affinity, with estimates in murine models typically around 2–4 days (Abdiche *et al*, 2015; Medesan *et al*, 1998) . However, in our model the injection is subcutaneous—almost equivalent to an intradermal injection (Figure 6B, C). Because this method is expected to provide a more sustained, slow-release effect (similar to the tuberculin reaction), the half-life should be longer than that achieved with intravenous administration. Consequently, we believe that sufficient efficacy is maintained in this model.

Regarding LW-6:

LW-6 is a small molecule that, due to its hydrophobic nature, is believed to freely cross cell membranes. Once inside the cell, it facilitates the degradation of HIF-1 α , leading to reduced expression of its downstream targets (Lee *et al*, 2010). Although its half-life is estimated to be around 30 minutes, the active metabolites may exert sustained secondary effects (Lee *et al*, 2021). When administered intravenously, peak blood concentrations are reached within 5 minutes, making C_{max} a critical parameter due to the rapid onset of action. In our experiments, we based the dosing regimen on previous studies (Lee *et al*, 2010; Song *et al*, 2016; Xu *et al*, 2022, 2024). While those studies administered doses comparable to or twice as high as ours via intravenous, intraperitoneal, or oral routes, our experimental design—in which a single dose was administered on Day 4 and samples were collected on Day 7—necessitated a single-dose protocol.

Regarding Rapamycin:

Several studies have demonstrated that local administration yields anti-inflammatory effects (Takayama *et al*, 2014; Tyler *et al*, 2011). Similar outcomes have been observed in vascular malformations (Boscolo *et al*, 2015; Martinez-Corral *et al*, 2020). Although the half-life of rapamycin is estimated to be approximately 6 hours following intravenous administration, it may be even shorter (Comas *et al*, 2012; Popovich *et al*, 2014).

In light of these comments, we have revised Figure 6. Furthermore, the Results section pertaining to Figure 6 has been updated as follows:

Hif-1 α and Vegf-A inhibitors suppress the progression of vascular malformations.

We next examined whether administering Hif-1 α and Vegf-A inhibitors could effectively treat vascular malformations. Tamoxifen was administered to 3–4-week-old CDH5-CreERT2;R26R-Pik3ca^{H1047R} mice to induce mutations in the dorsal skin. Anti-VEGF-A, a Vegf-A neutralizing antibody; LW6, a Hif-1 α inhibitor; and rapamycin, an mTOR inhibitor, were topically applied, and their effects were analyzed (Figure 6A). Both anti-VEGF-A and LW6 reduced the visible swelling in the dorsal skin, whereas the difference between the drug-treated and control groups was less pronounced with rapamycin (Figure 6B). In tamoxifen-treated Cre(–) mice, inflammatory cell infiltration and fibrosis were observed from the dermis to the subcutaneous tissue; however, there were no changes in the number of PECAM⁺ vasculatures or VEGFR3⁺ lymphatic vessels, including their enlarged forms, compared to the untreated control (Figure 6C–E). In contrast, tamoxifen administration to CDH5-CreERT2;R26R-Pik3ca^{H1047R} mice resulted in an increase in these vascular structures by day 4 (Figure 6C–E). At day 7, comparing mice with or without treatment using anti-VEGF-A, LW6, or rapamycin, the number of PECAM⁺ vasculatures was reduced in the treated groups; however, in the rapamycin group, the number of enlarged PECAM⁺ vasculatures did not differ from that in the untreated group (Figure 6F–M). Similarly, for VEGFR3⁺ lymphatic vessels, both anti-VEGF-A and LW6 induced a reduction, whereas rapamycin did not produce a statistically significant decrease (Figure 6N–U). (Page 11, lines 363–381)

****Referees cross-commenting****

The issues raised by referee #1 related to the phenotype analysis are right. In my opinion the Isl model here proposed well mimic human pathology evenf the vascular damage at. head is not so evident

Response:

Perhaps the discrepancy arises from a terminological issue. According to the *WHO Classification of Tumours*, commonly used in clinical settings, the term "Head and Neck" refers to the facial and cervical regions (including the oral cavity, larynx, pharynx, salivary glands, nasal cavity, etc.) and excludes the central nervous system. The inclusion of the brain in **Figure 10-R** may have led to some confusion. We included the brain because cerebral

cavernous malformations are classified as venous malformations, and thus serve as an example of common sites for venous malformations in humans.

To clarify this point, we have made slight revisions to the first part of the Introduction, as follows:

They frequently manifest in the head and neck region—here defined as the orofacial and cervical areas, excluding the brain. (Page2, lines 52-53)

Reviewer #2 (Significance (Required)):

General assessment

STRENGTH : a new mouse model seems to well recapitulate human vascular malformation.

Possible key molecules have been identified

WEAKNESS. The pharmacological approach to support the role of VEGFA e HIF is not appropriate

References for the review:

Abdiche YN, Yeung YA, Chaparro-Riggers J, Barman I, Strop P, Chin SM, Pham A, Bolton G, McDonough D, Lindquist K, *et al* (2015) The neonatal Fc receptor (FcRn) binds independently to both sites of the IgG homodimer with identical affinity. *mAbs* 7: 331–343

Alsuwailem A, Myer CM & Chaudry G (2020) Vascular anomalies of the head and neck. *Semin Pediatr Surg* 29: 150968

Boscolo E, Limaye N, Huang L, Kang K-T, Soblet J, Uebelhoer M, Mendola A, Natynki M, Seront E, Dupont S, *et al* (2015) Rapamycin improves TIE2-mutated venous malformation in murine model and human subjects. *J Clin Investig* 125: 3491–3504

Castillo SD, Tzouanacou E, Zaw-Thin M, Berenjano IM, Parker VER, Chivite I, Milà-Guasch M, Pearce W, Solomon I, Angulo-Urarte A, *et al* (2016) Somatic activating mutations in *Pik3ca* cause sporadic venous malformations in mice and humans. *Sci Transl Med* 8: 332ra43

Churchill MJ, Bois H du, Heim TA, Mudianto T, Steele MM, Nolz JC & Lund AW (2022) Infection-induced lymphatic zipper restricts fluid transport and viral dissemination from skin. *J Exp Med* 219: e20211830

Comas M, Toshkov I, Kuropatwinski KK, Chernova OB, Polinsky A, Blagosklonny MV, Gudkov AV & Antoch MP (2012) New nanoformulation of rapamycin Rapatar extends lifespan in homozygous *p53*^{-/-} mice by delaying carcinogenesis. *Aging (Albany NY)* 4: 715–722

- Dellinger MT & Brekken RA (2011) Phosphorylation of Akt and ERK1/2 Is Required for VEGF-A/VEGFR2-Induced Proliferation and Migration of Lymphatic Endothelium. *PLoS ONE* 6: e28947
- Graupera M, Guillermet-Guibert J, Foukas LC, Phng L-K, Cain RJ, Salpekar A, Pearce W, Meek S, Millan J, Cutillas PR, *et al* (2008) Angiogenesis selectively requires the p110 α isoform of PI3K to control endothelial cell migration. *Nature* 453: 662–666
- Gupta S, Ramjaun AR, Haiko P, Wang Y, Warne PH, Nicke B, Nye E, Stamp G, Alitalo K & Downward J (2007) Binding of Ras to Phosphoinositide 3-Kinase p110 α Is Required for Ras- Driven Tumorigenesis in Mice. *Cell* 129: 957–968
- Hare LM, Schwarz Q, Wiszniak S, Gurung R, Montgomery KG, Mitchell CA & Phillips WA (2015) Heterozygous expression of the oncogenic Pik3ca H1047R mutation during murine development results in fatal embryonic and extraembryonic defects. *Dev Biol* 404: 14–26
- Hong Y, Lange-Asschenfeldt B, Velasco P, Hirakawa S, Kunstfeld R, Brown LF, Bohlen P, Senger DR & Detmar M (2004) VEGF-A promotes tissue repair-associated lymphatic vessel formation via VEGFR-2 and the α 1 β 1 and α 2 β 1 integrins. *FASEB J* 18: 1111–1113
- Hu H, Juvekar A, Lyssiotis CA, Lien EC, Albeck JG, Oh D, Varma G, Hung YP, Ullas S, Lauring J, *et al* (2016) Phosphoinositide 3-Kinase Regulates Glycolysis through Mobilization of Aldolase from the Actin Cytoskeleton. *Cell* 164: 433–446
- Jauhiainen S, Ilmonen H, Vuola P, Rasinkangas H, Pulkkinen HH, Keränen S, Kiema M, Liikkanen JJ, Laham-Karam N, Laidinen S, *et al* (2023) ErbB signaling is a potential therapeutic target for vascular lesions with fibrous component. *eLife* 12: e82543
- Larue L & Bellacosa A (2005) Epithelial–mesenchymal transition in development and cancer: role of phosphatidylinositol 3' kinase/AKT pathways. *Oncogene* 24: 7443–7454
- Lee JW & Chung HY (2018) Vascular anomalies of the head and neck: current overview. *Arch Craniofacial Surg* 19: 243–247
- Lee K, Kang JE, Park S-K, Jin Y, Chung K-S, Kim H-M, Lee K, Kang MR, Lee MK, Song KB, *et al* (2010) LW6, a novel HIF-1 inhibitor, promotes proteasomal degradation of HIF-1 α via upregulation of VHL in a colon cancer cell line. *Biochem Pharmacol* 80: 982–989
- Lee K, Lee J-Y, Lee K, Jung C-R, Kim M-J, Kim J-A, Yoo D-G, Shin E-J & Oh S-J (2021) Metabolite Profiling and Characterization of LW6, a Novel HIF-1 α Inhibitor, as an Antitumor Drug Candidate in Mice. *Molecules* 26: 1951
- Lin Y, Dong M, Liu Z, Xu M, Huang Z, Liu H, Gao Y & Zhou W (2022) A strategy of vascular-targeted therapy for liver fibrosis. *Hepatology* 76: 660–675
- Lupu I-E, Kirschnick N, Weischer S, Martinez-Corral I, Forrow A, Lahmann I, Riley PR, Zobel T, Makinen T, Kiefer F, *et al* (2022) Direct specification of lymphatic endothelium from non-venous angioblasts. *Biorxiv*: 2022.05.11.491403

- Martinez-Corral I, Zhang Y, Petkova M, Ortsäter H, Sjöberg S, Castillo SD, Brouillard P, Libbrecht L, Saur D, Graupera M, *et al* (2020) Blockade of VEGF-C signaling inhibits lymphatic malformations driven by oncogenic PIK3CA mutation. *Nat Commun* 11: 2869
- Maruyama K, Miyagawa-Tomita S, Haneda Y, Kida M, Matsuzaki F, Imanaka-Yoshida K & Kurihara H (2022) The cardiopharyngeal mesoderm contributes to lymphatic vessel development in mouse. *Elife* 11
- Maruyama K, Miyagawa-Tomita S, Mizukami K, Matsuzaki F & Kurihara H (2019) Isl1-expressing non-venous cell lineage contributes to cardiac lymphatic vessel development. *Dev Biol* 452: 134–143
- Maruyama K, Naemura K, Arima Y, Uchijima Y, Nagao H, Yoshihara K, Singh MK, Uemura A, Matsuzaki F, Yoshida Y, *et al* (2021) Semaphorin3E-PlexinD1 signaling in coronary artery and lymphatic vessel development with clinical implications in myocardial recovery. *Iscience*: 102305
- Mashima T, Wakatsuki T, Kawata N, Jang M-K, Nagamori A, Yoshida H, Nakamura K, Migita T, Seimiya H & Yamaguchi K (2021) Neutralization of the induced VEGF-A potentiates the therapeutic effect of an anti-VEGFR2 antibody on gastric cancer in vivo. *Sci Rep* 11: 15125
- Medesan C, Cianga P, Mummert M, Stanescu D, Ghetie V & Ward ES (1998) Comparative studies of rat IgG to further delineate the Fc : FcRn interaction site. *Eur J Immunol* 28: 2092–2100
- Nagao M, Hamilton JL, Kc R, Berendsen AD, Duan X, Cheong CW, Li X, Im H-J & Olsen BR (2017) Vascular Endothelial Growth Factor in Cartilage Development and Osteoarthritis. *Sci Rep* 7: 13027
- Nair SC (2018) Vascular Anomalies of the Head and Neck Region. *J Maxillofac Oral Surg* 17: 1–12
- Popovich IG, Anisimov VN, Zabezhinski MA, Semenchenko AV, Tyndyk ML, Yurova MN & Blagosklonny MV (2014) Lifespan extension and cancer prevention in HER-2/neu transgenic mice treated with low intermittent doses of rapamycin. *Cancer Biol Ther* 15: 586–592
- Ryu JY, Chang YJ, Lee JS, Choi KY, Yang JD, Lee S-J, Lee J, Huh S, Kim JY & Chung HY (2023) A nationwide cohort study on incidence and mortality associated with extracranial vascular malformations. *Sci Rep* 13: 13950
- Sadick M, Wohlgemuth WA, Huelse R, Lange B, Henzler T, Schoenberg SO & Sadick H (2017) Interdisciplinary Management of Head and Neck Vascular Anomalies: Clinical Presentation, Diagnostic Findings and Minimalinvasive Therapies. *Eur J Radiol Open* 4: 63–68
- Singh AM, Reynolds D, Cliff T, Ohtsuka S, Mattheyses AL, Sun Y, Menendez L, Kulik M & Dalton S (2012) Signaling Network Crosstalk in Human Pluripotent Cells: A Smad2/3-

Regulated Switch that Controls the Balance between Self-Renewal and Differentiation. *Cell Stem Cell* 10: 312–326

Song JG, Lee YS, Park J-A, Lee E-H, Lim S-J, Yang SJ, Zhao M, Lee K & Han H-K (2016) Discovery of LW6 as a new potent inhibitor of breast cancer resistance protein. *Cancer Chemother Pharmacol* 78: 735–744

Srinivasan RS, Dillard ME, Lagutin OV, Lin F-J, Tsai S, Tsai M-J, Samokhvalov IM & Oliver G (2007) Lineage tracing demonstrates the venous origin of the mammalian lymphatic vasculature. *Gene Dev* 21: 2422–2432

Stanczuk L, Martinez-Corral I, Ulvmar MH, Zhang Y, Laviña B, Fruttiger M, Adams RH, Saur D, Betsholtz C, Ortega S, *et al* (2015) cKit Lineage Hemogenic Endothelium-Derived Cells Contribute to Mesenteric Lymphatic Vessels. *Cell Reports* 10: 1708–1721

Stone OA & Stainier DYR (2019) Paraxial Mesoderm Is the Major Source of Lymphatic Endothelium. *Dev Cell* 50: 247–255.e3

Surve CR, Duran CL, Ye X, Chen X, Lin Y, Harney AS, Wang Y, Sharma VP, Stanley ER, Cox D, *et al* (2024) Signaling events at TMEM doorways provide potential targets for inhibiting breast cancer dissemination. *bioRxiv*: 2024.01.08.574676

Takayama K, Kawakami Y, Kobayashi M, Greco N, Cummins JH, Matsushita T, Kuroda R, Kurosaka M, Fu FH & Huard J (2014) Local intra-articular injection of rapamycin delays articular cartilage degeneration in a murine model of osteoarthritis. *Arthritis Res Ther* 16: 482

Tyler B, Wadsworth S, Recinos V, Mehta V, Vellimana A, Li K, Rosenblatt J, Do H, Gallia GL, Siu I-M, *et al* (2011) Local delivery of rapamycin: a toxicity and efficacy study in an experimental malignant glioma model in rats. *Neuro-Oncol* 13: 700–709

Wuest TR & Carr DJJ (2010) VEGF-A expression by HSV-1–infected cells drives corneal lymphangiogenesis. *J Exp Med* 207: 101–115

Xu H, Chen Y, Li Z, Zhang H, Liu J & Han J (2022) The hypoxia-inducible factor 1 inhibitor LW6 mediates the HIF-1 α /PD-L1 axis and suppresses tumor growth of hepatocellular carcinoma in vitro and in vivo. *Eur J Pharmacol* 930: 175154

Xu J, Lamouille S & Derynck R (2009) TGF- β -induced epithelial to mesenchymal transition. *Cell Res* 19: 156–172

Xu Q, Liu H, Ye Y, Wuren T & Ge R (2024) Effects of different hypoxia exposure on myeloid-derived suppressor cells in mice. *Exp Mol Pathol* 140: 104932

Yu JSL, Ramasamy TS, Murphy N, Holt MK, Czapiewski R, Wei S-K & Cui W (2015) PI3K/mTORC2 regulates TGF- β /Activin signalling by modulating Smad2/3 activity via linker phosphorylation. *Nat Commun* 6: 7212

Full Revision

Zenner K, Cheng CV, Jensen DM, Timms AE, Shivaram G, Bly R, Ganti S, Whitlock KB, Dobyys WB, Perkins J, *et al* (2019) Genotype correlates with clinical severity in PIK3CA-associated lymphatic malformations. *Jci Insight* 4

11th Mar 2025

Dear Dr. Maruyama,

Thank you for submitting your revised study. We have now received the feedback from referees #1 and #2 who evaluated your revised manuscript (and had reviewed your initial submission at Review Commons).

As you will see from the reports below, while referee #2 is satisfied with the revisions, referee #1 still raises some minor concerns. We would therefore like to invite you to revise the manuscript according to this referee's recommendations, and to further address the following editorial issues:

1/ Manuscript text:

- Please remove the red font text, and only keep in track changes any new modification.

- Please remove "data not shown" (p.13). As per journal policy, all data discussed in the manuscript should be shown in the main or EV figures.

- "Methods and Protocols" should be renamed "Methods":

o All Materials and Methods need to be described in the main text using our 'Structured Methods' format. According to this format, the Methods section includes a Reagents and Tools Table (listing key reagents, experimental models, software and relevant equipment and including their sources and relevant identifiers) followed by a Methods and Protocols section describing the methods, ideally using a step-by-step protocol format. The aim is to facilitate adoption of the methodologies across labs. Please download and fill our Reagents and Tools Table template (.docx), which you can find in our author guidelines:

o Human samples: please include the full sentence that the experiments conformed to the principles set out in the WMA Declaration of Helsinki and the Department of Health and Human Services Belmont Report.

- Data availability: Please note that the specific URL for GSE279129 dataset is not provided in the data availability statement. Please note that the data must be publicly available at time of publication. If practically possible and compatible with the individual consent agreement, we have to make sure that the authors deposit the human clinical datasets to public databases at the time of publication, however, authors must ensure that privacy of individuals is preserved. Kindly remove "There is no restriction on data availability. Source data are provided in this paper."

- Acknowledgements: the funding information provided in this section should match the information provided in the submission system (currently, funding information as listed in the Acknowledgments are missing in the submission system).

- Author contributions: CRediT has replaced the traditional author contributions section because it offers a systematic machine readable author contributions format that allows for more effective research assessment. Please remove the Authors Contributions from the manuscript and use the free text boxes beneath each contributing author's name in our system to add specific details on the author's contribution. More information is available in our guide to authors.

- Please rename "Competing financial interests" to "Disclosure and competing interests".

2/ Figures:

- Supplementary figures should also be uploaded as individual, high resolution figure files, the legends should be moved to the manuscript text, after the main figure legends and under the heading "Expanded View Figure Legends". The suppl. figures should be renamed Figure EV1 - 9.

- Supplemental Table 1 should be renamed Table EV1 and the legend in the file should also be corrected; the datasets with Supplemental Data 1 - 4 should be renamed Dataset EV1 - EV4. A legend should be added to each dataset, in a separate tab/worksheet.

- Please make sure that all figures are referenced in the text, in chronological order (currently, a callout is missing for Fig 3K; the callout for the table with primers should be corrected to Table EV1).

- Please note that (partial) figure re-use must be mentioned in the figure legend (partial re-use Fig 2X and Fig 2X', Suppl. Fig. 1B and Suppl. Fig. 9a').

- Please address the queries from our copy editors:

1. Please note that information related to n is missing in the legends of figures 4F

2. Please note that the error bars are not defined in the legends of figures 1E, F, I, J, M, N, Q, R; 2 AD, AE; 3K, L, M; 6D, E, L, M, T, U

3/ At EMBO Press we ask authors to provide source data for the main figures. Our source data coordinator will contact you to discuss which figure panels we would need source data for and will also provide you with helpful tips on how to upload and organize the files.

4/ Please provide a complete author checklist, which you can download from our author guidelines

(<https://www.embopress.org/page/journal/17574684/authorguide#submissionofrevisions>). Please insert information in the checklist that is also reflected in the manuscript. The completed author checklist will also be part of the RPF.

5/ The paper explained: EMBO Molecular Medicine articles are accompanied by a summary of the articles to emphasize the major findings in the paper and their medical implications for the non-specialist reader. Please provide a draft summary of your article highlighting

6/ Every published paper includes a 'Synopsis' to further enhance discoverability. Synopses are displayed on the journal webpage and are freely accessible to all readers. They include a short stand first (maximum of 300 characters, including space) as well as 2-5 one-sentences bullet points that summarizes the paper. Please write the bullet points to summarize the key NEW findings. They should be designed to be complementary to the abstract - i.e. not repeat the same text. We encourage inclusion of key acronyms and quantitative information (maximum of 30 words / bullet point). Please use the passive voice.

Please also suggest a visual abstract to illustrate your article as a PNG file 550 px wide x 300-600 px high. A cropped portion of this image will serve as thumbnail for the table of content on our webpage.

7/ As part of the EMBO Publications transparent editorial process initiative (see our Editorial at <http://embomolmed.embopress.org/content/2/9/329>), EMBO Molecular Medicine will publish online a Review Process File (RPF) to accompany accepted manuscripts.

This file will be published in conjunction with your paper and will include the anonymous referee reports, your point-by-point response and all pertinent correspondence relating to the manuscript. Let us know whether you agree with the publication of the RPF and as here, if you want to remove or not any figures from it prior to publication.

I look forward to receiving your revised manuscript.

Yours sincerely,

Lise Roth

***** Reviewer's comments *****

Referee #1 (Comments on Novelty/Model System for Author):

Overall this is an interesting study which provides important answers for the understanding of PIK3CA-related vascular malformations.

Referee #1 (Remarks for Author):

While the manuscript has improved with some the clarifications, there are still some aspects which need some attention

In Fig. 6C, please provide histological characterization of the same day (either 7 or 4). Same applies to quantification in Fig. 6D and E. The same time point should be considered. To the best of knowledge, this is the first time that this model is reported and used for preclinical work. So, it is important that authors do a good job for the community. If the model has been published, please cite the original publication.

In Fig. 6B provide an image of non-drug treated CreP, and Tamoxifen treated mouse

Clarify in the figure legend what treatment (-) and (+) means in Fig. 6F. I am assuming is non-drug treated CreP and Tamoxifen treated mouse. But it is not clear.

Provide the color scale of Fig. 4C

Explain in the figure legend what the red circle is in Fig. 4G.

Add the "n" in Supplementary Fig. 1L-Q for consistency with other figures.

It is unclear why authors propose Etv2, Kdr, Flt4 and Pecam1 as markers of immature ECs in Supplementary Figure 6C,D (line 295-296). And then later they use Flt4 and Pecam1 in Supplementary Figure 6E-J as markers of mature ECs (line 302). Also, I do not understand the rationale of sub clustering the Cluster 4, as it clearly expresses much less of those markers. Please explain better the rationale.

Referee #2 (Remarks for Author):

The authors provided exhaustive answers to my comments. About my comment on scRNA (point 5) I've well understood the authors' reply and I agree with it. However I suggest to add in the discussion section a paragraph indicating this limit

Rev_Com_number: N/a

New_manu_number: EMM-2025-21443-T

Corr_author: Maruyama

Title: Embryological cellular origins and hypoxia-mediated mechanisms in PIK3CA-Driven refractory vascular malformations

***** Reviewer's comments *****

Referee #1 (Comments on Novelty/Model System for Author):

Overall this is an interesting study which provides important answers for the understanding of PIK3CA-related vascular malformations.

Referee #1 (Remarks for Author):

While the manuscript has improved with some the clarifications, there are still some aspects which need some attention

1. In Fig. 6C, please provide histological characterization of the same day (either 7 or 4). Same applies to quantification in Fig. 6D and E. The same time point should be considered. To the best of knowledge, this is the first time that this model is reported and used for preclinical work. So, it is important that authors do a good job for the community. If the model has been published, please cite the original publication.

Response:

First of all, thank you very much for taking the time to thoroughly review our work.

Although our study differs in that we used the ear and back skin, our experimental procedure closely follows that of (Martinez-Corral *et al*, 2020), who locally administered tamoxifen to the ear. Initially, we tried the same approach, but because the ear skin is very thin and easily torn, it was challenging to administer or apply tamoxifen reliably. Consequently, we chose to use the dorsal skin instead.

To clarify the figures, we reorganized our layout to distinguish between day 4 and day 7, and we provided more detailed explanations in the figure legends and the Results section regarding what each group represents. Although our main conclusions remain essentially unchanged, the statistically significant effect of rapamycin observed at day 7 has now disappeared.

We did not emphasize this strongly in the main text, but the cited study showed rapamycin to be effective. However, current clinical findings do not support the efficacy of topical rapamycin. The discrepancy may stem from the relatively high doses administered in mice. As a result, rapamycin effectively served as a kind of control in our experiments, where it did not appear to exert a strong effect.

In Fig. 6B provide an image of non-drug treated CreP, and Tamoxifen treated mouse

Response;

We are not entirely certain we understand the reviewer's request. However, in Figure 6B:

- The red circle indicates the *CreLoxP+* *Tamoxifen+* *Drug-* condition (*CDH5-CreERT2; R26R-Pik3ca^{H1047R}* with tamoxifen but without drug).
- The green circle indicates the *CreLoxP+* *Tamoxifen-* *Drug-* condition (*CDH5-CreERT2; R26R-Pik3ca^{H1047R}* without tamoxifen or drug).
- The blue circle indicates the *CreLoxP+* *Tamoxifen+* *Drug+* condition (*CDH5-CreERT2; R26R-Pik3ca^{H1047R}* with both tamoxifen and drug).

Administering tamoxifen clearly induces a visible swelling (vascular malformation) in the red circle region. By comparing the left and right sides, the size difference becomes evident. We shaved the hair in areas farther from the region of interest—where neither tamoxifen nor drug was administered—to serve as an additional control. The layout of our figure closely follows Figure 5 in (Martinez-Corral *et al*, 2020).

Clarify in the figure legend what treatment (-) and (+) means in Fig. 6F. I am assuming is non-drug treated CreP and Tamoxifen treated mouse. But it is not clear.

Response:

We have addressed the above comments. However, because the figures and explanations were unclear, we have comprehensively revised Figure 6, the Results section, and the figure legends to improve clarity.

Provide the color scale of Fig. 4C

Response:

We have added two different color bars.

Explain in the figure legend what the red circle is in Fig. 4G.

Response:

Thank you for your feedback. We have added the following statement:

“Genes that were upregulated in the *Pik3ca^{H1047R}* mutant endothelial cell clusters are circled in red.”

(Page 34, lines 1085–1086)

Add the "n" in Supplementary Fig. 1L-Q for consistency with other figures.

Response:

We believe it is already included, but please let us know if there is anything else you would like us to address.

It is unclear why authors propose Etv2, Kdr, Flt4 and Pecam1 as markers of immature ECs in Supplementary Figure 6C,D (line 295-296). And then later they use Flt4 and Pecam1 in Supplementary Figure 6E-J as markers of mature ECs (line 302). Also, I do not understand the rationale of sub clustering the Cluster 4, as it clearly expresses much less of those markers. Please explain better the rationale.

Response:

We included Cluster 4 because, although it may appear less distinct, we observed ETV2 and Kdr expression in the UMAP analysis (Figure EV 6D). To clarify that ETV2 is an upstream regulator in endothelial cells, followed by the expression of markers such as PECAM, we have cited additional references and revised our text as follows:

We focused on clusters 4, 11, 12, and 3, which contained immature ECs, as indicated by markers such as Etv2, Kdr (Figure EV 6C, D, Dataset EV 2). Subclustering analysis of these clusters revealed seven subclusters, labeled 0 through 6 (Figure EV 6E-H, Dataset EV 2). RNA velocity and trajectory analyses revealed early expression of Isl1 and Wnt5a, which decreased as EC differentiation progressed. This was followed by the expression of key endothelial differentiation markers, Etv2 and Kdr, and subsequently by markers characteristic of a later stage of endothelial differentiation (Val & Black, 2009; Morita et al, 2015), such as Flt4 and Pecam1 (Figure EV 6E-J). (Page 9, lines 287-294)

Referee #2 (Remarks for Author):

The authors provided exhaustive answers to my comments. About my comment on scRNA (point 5) I've well understood the authors' reply and I agree with it. However I suggest to add in the discussion section a paragraph indicating this limit

Response:

Thank you for your valuable feedback. We have addressed it in the Discussion section as follows.

Furthermore, future studies using both earlier and later embryonic stages will be essential to fully elucidate how *Pik3ca*^{H1047R} influences cellular differentiation and the progression of vascular malformations. (P13, lines 431-433)

References:

- Martinez-Corral I, Zhang Y, Petkova M, Ortsäter H, Sjöberg S, Castillo SD, Brouillard P, Libbrecht L, Saur D, Graupera M, *et al* (2020) Blockade of VEGF-C signaling inhibits lymphatic malformations driven by oncogenic PIK3CA mutation. *Nat Commun* 11: 2869
- Morita R, Suzuki M, Kasahara H, Shimizu N, Shichita T, Sekiya T, Kimura A, Sasaki K, Yasukawa H & Yoshimura A (2015) ETS transcription factor ETV2 directly converts human fibroblasts into functional endothelial cells. *Proc National Acad Sci* 112: 160–165
- Val SD & Black BL (2009) Transcriptional Control of Endothelial Cell Development. *Dev Cell* 16: 180–195

21st Mar 2025

Dear Dr. Maruyama,

Thank you for submitting your revised files. I am now ready to accept your manuscript once the following editorial queries are addressed:

- Please remove the red font and only keep in track changes mode any new modification.
- Thank you for providing a Reagent and Tools table, please remove the pages with examples and only keep data relevant to your study.
- Data availability: We understand that NGS was performed for identification of human variants. Please note that if practically possible and compatible with the individual consent agreement, human clinical datasets must be deposited to public databases at the time of publication (i.e. EGA). Authors must also ensure that patients' anonymity is preserved.
- Figure re-use: please see the folder attached, that shows the figure panels that were partially re-used. Please correct the figures or clarify in the figure legends.
- Checklist:
 - o please check your answer to "If collected and within the bounds of privacy constraints report on age, sex and gender or ethnicity for all study participants." as I haven't found this information in the manuscript.
 - o Please fill in all subsections of "Experimental study design and statistics"
 - o Please fill in "Data availability/human clinical and genomic datasets" (see comment above).
- The paper explained: please replace "the authors" (results, second line) by "we".
- Synopsis: please remove the text from the manuscript file and upload it individually. I cropped a small portion from your graphical abstract to serve as thumbnail on our electronic table of content (attached), please let me know if you agree or provide an alternative image (115x70 pixels).

I look forward to receiving your revised manuscript at your earliest convenience.

Yours sincerely,

Lise Roth

***** Reviewer's comments *****

Rev_Com_number: N/a
New_manu_number: EMM-2025-21443-V2
Corr_author: Maruyama
Title: Embryological cellular origins and hypoxia-mediated mechanisms in PIK3CA-Driven refractory vascular malformations

The authors addressed the remaining editorial issues.

27th Mar 2025

Dear Dr. Maruyama,

Thank you for sending your revised files. I am pleased to inform you that your manuscript is accepted for publication and is now being sent to our publisher to be included in the next available issue of EMBO Molecular Medicine!

Yours sincerely,

Lise Roth
